# STEERING MASKED DISCRETE DIFFUSION MODELS VIA DISCRETE DENOISING POSTERIOR PREDICTION

**Jarrid Rector-Brooks**[1,2]**, Mohsin Hasan**[1,2]**, Zhangzhi Peng**[3]**, Zachary Quinn**[3]**,
Chenghao Liu**[2,4]**, Sarthak Mittal**[1,2]**, Nouha Dziri**[5]**, Michael Bronstein**[6,7]**,
Yoshua Bengio**[1,2]**, Pranam Chatterjee**[3]**, Alexander Tong**[1,2][*]**Avishek Joey Bose**[2,6][*]
[1]Université de Montréal, [2]Mila, [3]Duke University, [4]McGill University,
[5]Allen Institute for AI, [6]Oxford University, [7]Aithyra

## ABSTRACT

Generative modeling of discrete data underlies important applications spanning text-based agents like ChatGPT to the design of the very building blocks of life in protein sequences. However, application domains need to exert control over the generated data by steering the generative process—typically via RLHF—to satisfy a specified property, reward, or affinity metric. In this paper, we study the problem of steering Masked Diffusion Models (MDMs), a recent class of discrete diffusion models that offer a compelling alternative to traditional autoregressive models. We introduce DISCRETE DENOISING POSTERIOR PREDICTION (DDPP), a novel framework that casts the task of steering pre-trained MDMs as a problem of probabilistic inference by learning to sample from a target Bayesian posterior. Our DDPP framework leads to a family of three novel objectives that are all simulation-free, and thus scalable while applying to general non-differentiable reward functions. Empirically, we instantiate DDPP by steering MDMs to perform class-conditional pixel-level image modeling, RLHF-based alignment of MDMs using text-based rewards, and finetuning protein language models to generate more diverse secondary structures and shorter proteins. We substantiate our designs via wet-lab validation, where we observe transient expression of reward-optimized protein sequences.

## 1 INTRODUCTION

The success of diffusion models in continuous spaces, leading to state-of-the-art foundation models for image (Stability AI, 2023; Midjourney, 2023) and video synthesis (Villegas et al., 2022; Brooks et al., 2024), has spurred several attempts to translate these approaches for the generative modeling of discrete structures. The most performant approaches squarely fall under the scalable framework of absorbing state discrete diffusion (Austin et al., 2021), with new simplified training recipes that result in Masked Diffusion Models (MDMs) (Sahoo et al., 2024; Shi et al., 2024; Gat et al., 2024; Zhao et al., 2024a). Indeed, recent MDMs now rival autoregressive models of a similar scale to GPT-2 (Radford et al., 2019) for language modeling, with the potential for further progress through scaling. Furthermore, MDM style models are not constrained to generating data sequentially—unlike autoregressive models—which invites a more straightforward application to domains without a natural causal ordering, e.g. molecule generation (Vignac et al., 2022), discrete modeling of images (Salimans et al., 2017), and modeling protein sequences (Lin et al., 2022; Wang et al., 2024).

Critical to the successful deployment of discrete generative models in practical applications—beyond simply producing high-quality samples—is the ability to steer the generated samples to optimize a pre-specified downstream metric. For instance, in Language Modeling (LM) it is desirable to bias the model's generations to be sanitized from harmful responses (Zou et al., 2023; Perez et al., 2022), or aiming to generate protein sequences that are highly likely to be successfully synthesized and expressed in real wet lab settings (Verkuil et al., 2022; Dauparas et al., 2022). Put succinctly, highly performant discrete generative models are required to be aligned in a manner that fine-tuning against downstream reward models has the intended effect of *controllable generation*, wherein the model post fine-tuning selects high-scoring samples from the universe of possible high-fidelity generations.

---

[*]Equal advising

The standard approach for incorporating steerability into discrete generative models, which are autoregressive, using pre-defined reward models is often framed as a fine-tuning task using reinforcement learning from human feedback (RLHF) (Christiano et al., 2017; Rafailov et al., 2024). However, applying RLHF frameworks to diffusion models is far more challenging. Unlike autoregressive models, diffusion models do not allow for straightforward computation of a sample's exact likelihood without costly simulations. Although fine-tuning diffusion models that bypass exact likelihood computation can yield simulation-free algorithms that resemble RLHF (Wallace et al., 2024; Uehara et al., 2024a), these methods effectively optimize a loose lower bound to the true RLHF objective, leading to unstable training and suboptimal fine-tuning performance. Consequently, steering diffusion models in continuous spaces is primarily done through inference techniques that leverage the gradient of a conditional model in the form of guidance (Dhariwal and Nichol, 2021; Ho and Salimans, 2022). Unfortunately, discrete settings do not allow for principled definitions of guidance due to the lack of a conventional gradient operator. As a result, at present, there exists no scalable and rigorous method to steer and align Masked Diffusion Models to optimize desired reward models.

**Main contributions**. In this paper, we cast the problem of steering a Masked Diffusion Model as a task of probabilistic inference in sampling from a target Bayesian posterior. More precisely, we construct the target Bayesian posterior as being proportional to the product distribution of a base pre-trained MDM model modulated by a pre-specified reward model. Importantly, this sampling viewpoint is fully compatible with classical RLHF for autoregressive models (Uehara et al., 2024a; Zhao et al., 2024b), but enjoys broader applicability as for the first time it can be applied to discrete diffusion models. Under this sampling perspective, our key insight is that the challenging sampling problem can be solved by learning an amortized sampler, which when taken as an MDM, can be viewed as *finetuning* the pre-trained MDM model by learning to approximate the (reward-induced) Bayesian posterior.

We introduce DISCRETE DENOISING POSTERIOR PREDICTION (DDPP), a novel framework that exploits the denoising posterior parametrization inherent to current MDMs to define a series of simpler matching problems across varying corruption (masking) levels of the target Bayesian posterior. In particular, DDPP designs a forward process that corrupts the Bayesian posterior through a forward masking process and frames the finetuning task as learning another MDM to approximate the corresponding reverse process. As a result, each matching problem in the reverse process requires the construction of a "denoising" Bayesian posterior that is conditioned on a partially masked sample which we demonstrate is simply proportional to the pre-trained model's own denoising posterior and the (terminal) reward of the fully unmasked sample. Crucially, each matching problem in DDPP can be defined on a particular noise level without running the entire forward (corruption) process. Consequently, this makes DDPP a *simulation-free* method which is a key ingredient needed to finetune large pre-trained MDMs. We test the empirical caliber of DDPP by steering MDMs across a multitude of domains ranging from images to protein sequences and steering MDM-based language models. We observe DDPP fine-tuned MDMs lead to competitive performance on images, transient expression of reward-optimized protein sequences (with high secondary structure diversity and $\beta$-sheet composition) in a wet-lab setting, and natural textual responses that are steered to human sentiments.

## 2 BACKGROUND AND PRELIMINARIES

**Notation and convention**. A discrete data sample $x$ is identified by its realization over a vocabulary set $\mathcal{X} = \{1, \ldots, d-1\}$, over $d-1$ possible categories. Of particular interest is the setting of masked diffusion models that include an additional $d$-th category of a masked token $\mathbf{m}$ to the vocabulary $\mathcal{X}$ which serves as an absorbing state for the diffusion process. A discrete token is represented by the one-hot vector $e^i \in \Delta^d$ in the $d$-dimensional probability simplex and corresponds to placing a 1 on the $i$-th index and 0 on all the other $d-1$ indices. In this paper, by convention, we set $e^{\mathbf{m}} = e^d$ as the one hot vector associated with the masked state $\mathbf{m}$. A categorical distribution over $d$-categories, $\text{Cat}(x; p)$, is constructed by placing a Dirac $\delta$ with weight $p^i$, with the constraint $\sum_i p^i = 1$ and the density of a discrete sample is written as $p(X = x) = \sum_{i=0}^{d} p^i \delta(x - e^i)$, where $X$ is the discrete random variable.

A sequence $\mathbf{x} = (x^1, \ldots, x^n)$ of $n$ tokens is defined over the product space $\mathcal{X}^n = \{1, \ldots, \mathbf{m}\}^n$, and its corresponding probability mass function is given by $p(X = \mathbf{x}) = \prod_i^n \sum_{j=0}^{d} p^j \delta(x^i - e^j)$. To reduce notational clutter, we make use of the shorthand $\delta(y)$ to denote a Dirac measure on a discrete sample $y$ and interchangeably write $p(X = \mathbf{x}) = p(\mathbf{x})$ to denote the probability mass function. A dataset of sequences is designated as samples from the data distribution $p_{\text{data}}$ to be learned by a discrete diffusion model $q_\theta$, with parameters $\theta$. Discrete diffusion models, like their continuous

counterparts, are stochastic processes that evolve with time $t \in [0,1]$ such that $t = 0$ corresponds to $p_{\text{data}} := p_0$ and $t = 1$ corresponds to the terminal distribution, $p_1$ of the process. As a discretization of time, we divide $[0,1]$ into $T$ intervals, and let $t(i) = i/T$. For brevity, we drop $i$ and simply write $t$ to denote the corresponding discrete timestep $t(i)$. The notation $0:t$ designates a collection of objects, e.g. densities $p(\mathbf{x}_{0:t})$, starting from time $t$ to and including time $t = 0$. A trajectory of sequences is denoted as $\tau(\mathbf{x}_{0:1}) = \mathbf{x}_1 \to \cdots \to \mathbf{x}_t \to \mathbf{x}_{t-1} \to \cdots \to \mathbf{x}_0$. Finally, we use subscripts to denote the time index—i.e. $p_t$—and reserve superscripts to designate indices over a set such as a specific sample $\mathbf{x}^i$ among a collection of samples or dimensions within a vector, e.g. dimension $x^i$ in a sequence.

**Problem Setting**. We are interested in the task of *probabilistic inference* of sampling from an unnormalized target distribution $\pi_0(\mathbf{x}_0)$ defined over a discrete space consisting of $n$ tokens $\mathbf{x} \in \mathcal{X}^n$,

$$\pi_0(\mathbf{x}_0) = \frac{p_0^{\text{pre}}(\mathbf{x}_0) R(\mathbf{x}_0)}{\mathcal{Z}_{\pi_0}}, \quad R(\mathbf{x}_0) = \frac{\exp(-\mathcal{E}(\mathbf{x}_0))}{\mathcal{Z}_R}. \tag{1}$$

A key aspect of the considered setting is that $\pi_0(\mathbf{x}_0)$ is defined as the product distribution of a pre-trained masked discrete diffusion model $p_0^{\text{pre}}(\mathbf{x}_0)$ and a distribution induced by a (potentially differentiable) reward model $R : \mathcal{X}^n \to \mathbb{R}$. The problem definition in Eq. 1 is an instance of Bayesian posterior sampling where the pre-trained MDM is the prior and reward acts as the likelihood or observation model which modulates samples with a high score. For instance, in scientific domains, the reward model can be provided as a Boltzmann distribution with a known energy function $\mathcal{E}(\mathbf{x}_0)$, or a human preference model as in RLHF (Ouyang et al., 2022; Rafailov et al., 2024). Importantly, this setting does not afford us *any* ground truth samples from $\pi_0(\mathbf{x}_0)$ in the form of a dataset which prevents classically training another generative model. Instead, we are able to evaluate the reward model—and in special cases its gradient $\nabla R$—but not the normalizing constant, i.e. the partition function $\mathcal{Z}_{\pi_0}$. Samples from the posterior $\pi_0$ thus lie in the intersection of the modes of both the pretrained MDM and the reward model. As a result, learning an amortized sampler, $q_\theta(\mathbf{x}_0)$, for $\pi(\mathbf{x}_0)$ is rationally equivalent to finetuning the pretrained MDM $p_0^{\text{pre}}(\mathbf{x}_0)$ using the reward $R(\mathbf{x}_0)$ in an analogous manner to RLHF (Uehara et al., 2024a) and is the main focus and contribution of this paper and outlined in §3.2.

## 2.1 SIMPLIFIED MASKED DISCRETE DIFFUSION

We are interested in developing a discrete diffusion model directly on discrete data—i.e. without embeddings or continuous reparameterizations—whose approach mirrors the construction of diffusion models for continuous spaces. Consequently, we require the specification of a forward process that converts discrete data $\mathbf{x}_0 \sim p_0$ at time $t = 0$ to an unstructured prior, $p_1$ at the terminal time $t = 1$. The specification of a forward process via the transition kernel $p_t(\mathbf{x}_t | \mathbf{x}_0)$ implies a unique time reversal of this forward process, termed the "reverse process", such that simulating from this reverse process results in samples from the desired target data distribution $p_0(\mathbf{x}_0)$.

We restrict our attention to the recent performant "simplified masked" forward process (Sahoo et al., 2024; Shi et al., 2024; Gat et al., 2024; Zhao et al., 2024a) which hits a terminal distribution of all mask tokens in a sequence $p_1 = [\delta(\mathbf{m})]^n$. Given a non-masked token in a sequence, $x_0^i \in \mathbf{x}$ the simplified masked forward process increases the likelihood of transition to the mask state as time increases. Moreover, the masked forward process is simplified by design since the transition probabilities of a token unmasking ($x_{t+1}^i \neq \mathbf{m}$ when $x_t^i = \mathbf{m}$) is set to zero—i.e. the token remains a masked token for the remainder of the trajectory. The design of the simplified forward process is also independent across each dimension of the sequence, conditioned on $\mathbf{x}_0$, which allows us to model the transitions of each discrete token in a sequence separately. In totality, the forward process for a sequence $\mathbf{x}_0$ can be summarized using the following expression for the transition kernel $p_t(x_t^i | x_0^i)$:

$$p_t(\mathbf{x}_t | \mathbf{x}_0) = \prod_{i=1}^n p_t(x_t^i | x_0^i) = \prod_{i=1}^n \text{Cat}(x_t^i; \alpha_t \delta(x_0^i) + (1 - \alpha_t)\delta(\mathbf{m})), \tag{2}$$

where $\alpha_t$ is an invertible reparameterization of time such that $\alpha_0 = 1$ and $\alpha_1 = 0$. Effectively, $\alpha_t$ corresponds to the noise schedule which corrupts the discrete data to $p_1$. The corresponding marginal density induced by the forward process at time $t$ can written as $p_t(\mathbf{x}_t) = \sum_{\mathbf{x}_0} p_t(\mathbf{x}_t | \mathbf{x}_0) p_0(\mathbf{x}_0)$.

The reverse process which denoises a sample from $t \to t-1$, and is the time reversal of the simplified masked forward process, also factorizes over each dimension of a sequence $\mathbf{x}$. The probability $p_t(x_{t-1}^i | x_t^i, x_0^i)$ of a reverse transition is given by the following posterior conditioned on $x_0^i$,

$$p_t(x_{t-1}^i | x_t^i, x_0^i) = \begin{cases} \text{Cat}(x_{t-1}^i; \delta(x_t^i)) & x_t^i \neq \mathbf{m} \\ \text{Cat}\left(x_{t-1}^i; \frac{(1 - \alpha_{t-1})\delta(\mathbf{m}) + (\alpha_{t-1} - \alpha_t)\delta(x_0^i)}{1 - \alpha_t}\right) & x_t^i = \mathbf{m}. \end{cases} \tag{3}$$

Under the reverse process once a token transitions out of the masked state for a time $t > 0$ it remains in this state for the remainder of the trajectory. The analytical form of the posterior suggests a natural mean parametrization for a denoiser in a discrete diffusion model, $\mu_\theta : \mathcal{X}^n \times \mathbb{R} \to (\Delta^d)^n$, which predicts the clean sample at $t = 0$ by denoising a noisy $x_t^i$,

$$q_{t,\theta}(x_{t-1}^i | x_t^i, \mu_\theta(x_t^i, t)) = \begin{cases} \text{Cat}(x_{t-1}^i; \delta(x_t^i)) & x_t^i \neq \mathbf{m} \\ \text{Cat}\left(x_{t-1}^i; \frac{(1-\alpha_{t-1})\delta(\mathbf{m}) + (\alpha_{t-1} - \alpha_t)\mu_\theta(x_t^i, t)}{1 - \alpha_t}\right) & x_t^i = \mathbf{m}. \end{cases} \quad (4)$$

Interestingly, the mean parametrization $\mu_\theta$ used in the posterior is equivalent to predicting the concrete score (Meng et al., 2022) which is the discrete equivalent of the Stein score found in conventional continuous diffusion models (Zheng et al., 2024). As the number of steps $t \to \infty$, training yields a valid *evidence lower bound* (ELBO) to the marginal log-likelihood of the data distribution $\log p(\mathbf{x}_0)$,

$$\log p(\mathbf{x}_0) \geq - \int_0^1 \frac{d\alpha_t}{dt} \cdot \frac{1}{1 - \alpha_t} \mathbb{E}_{\mathbf{x}_t \sim p_t(\mathbf{x}_t | \mathbf{x}_0)} \left[ \sum_{i=1}^n (x_0^i)^T \log \mu_\theta(x_t^i, t) \right] dt. \quad (5)$$

Thus, when given access to samples $\mathbf{x}_0 \sim p_0$ training an MDM can be seen as optimizing a weighted cross-entropy loss and is analogous to fitting a (mean-field) variational posterior distribution $q_{t,\theta}(\mathbf{x}_0 | \mathbf{x}_t) = \text{Cat}(\mathbf{x}_0; \mu_\theta(\mathbf{x}_t, t))$ that matches the first moments of $p_t(\mathbf{x}_0 | \mathbf{x}_t)$ and also minimizes the forward KL divergence $\mathbb{D}_{\text{KL}}(p_t(\mathbf{x}_0 | \mathbf{x}_t) p_t(\mathbf{x}_t) || q_{t,\theta}(\mathbf{x}_0 | \mathbf{x}_t) p_t(\mathbf{x}_t))$ (Eijkelboom et al., 2024).

## 3    POSTERIOR SAMPLING VIA DISCRETE DENOISING POSTERIOR PREDICTION

Given access to a pretrained masked discrete diffusion model $p_0^{\text{pre}}(\mathbf{x}_0)$ we wish to sample from the reward-induced Bayesian posterior distribution $\pi_0(\mathbf{x}_0) \propto p_0^{\text{pre}}(\mathbf{x}_0) R(\mathbf{x}_0)$. We solve this sampling problem by first defining a time-dependent forward masking process that progressively adds noise to $\pi_0$ yielding the noisy reward-induced posterior $\pi_t(\mathbf{x}_t) = \sum_{\mathbf{x}_0} \pi_t(\mathbf{x}_t | \mathbf{x}_0) \pi_0(\mathbf{x}_0)$, where we set $\pi_t(\mathbf{x}_t | \mathbf{x}_0) = p_t(\mathbf{x}_t | \mathbf{x}_0)$ as it is the same masking process for the pre-trained MDM. Unfortunately, since $p_0^{\text{pre}}(\mathbf{x}_0)$ is an MDM it does not easily provide an exact likelihood. Undeterred we seek to approximate the reverse process $\pi_t(\mathbf{x}_{t-1} | \mathbf{x}_t)$ tied to the masking forward process by using another parametrized model $q_{t,\theta}(\mathbf{x}_0 | \mathbf{x}_t) = \text{Cat}(\mathbf{x}_0; \mu_\theta(\mathbf{x}_t, t))$ which we take to be another MDM.

**Matching sub-trajectories**. To approximate the reverse process using an MDM we require matching the denoising trajectory $\tau(\mathbf{x}_{0:t})$ of the reward-induced posterior $\pi_t(\mathbf{x}_0, \ldots, \mathbf{x}_{t-1} | \mathbf{x}_t)$ across all masking levels. Assisted in this endeavor, we recall the fact that since $p_0^{\text{pre}}(\mathbf{x}_0)$ is also an MDM, we have direct access to the pre-trained model's denoiser. Thus, we can compute any transition density starting from $p_t^{\text{pre}}(\mathbf{x}_{t-1} | \mathbf{x}_t, \mu^{\text{pre}}(\mathbf{x}_t, t))$ to the posterior over the endpoint $p_t^{\text{pre}}(\mathbf{x}_0 | \mathbf{x}_t)$, conditioned on a partially masked sample $\mathbf{x}_t$. We form the sub-trajectory matching problem as an instantiation of a detailed balance constraint starting from a partially masked sequence $\mathbf{x}_t$ of a clean starting point $\mathbf{x}_0$:

$$q_\theta(\mathbf{x}_0, \ldots, \mathbf{x}_{t-1} | \mathbf{x}_t, \hat{\mathbf{x}}_0) p_t(\mathbf{x}_t) = \pi_t(\mathbf{x}_0, \ldots, \mathbf{x}_{t-1} | \mathbf{x}_t) p_t(\mathbf{x}_t). \quad (6)$$

Setting $\hat{\mathbf{x}}_0 = \mu_\theta(\mathbf{x}_t, t)$ as the MDM's denoised sample, then $\pi_t(\mathbf{x}_0, \ldots, \mathbf{x}_{t-1} | \mathbf{x}_t)$ is defined as,

$$\pi_t(\mathbf{x}_0, \ldots, \mathbf{x}_{t-1} | \mathbf{x}_t) = \frac{p_t^{\text{pre}}(\mathbf{x}_0, \ldots, \mathbf{x}_{t-1} | \mathbf{x}_t) R(\mathbf{x}_0)}{\mathcal{Z}_{\pi_t}(\mathbf{x}_t)} = \frac{\prod_{j=1}^t p_t^{\text{pre}}(\mathbf{x}_{j-1} | \mathbf{x}_j, \hat{\mathbf{x}}_0^{\text{pre}}) R(\mathbf{x}_0)}{\mathcal{Z}_{\pi_t}(\mathbf{x}_t)}.$$

The detailed balance constraint over sub-trajectories suggests a natural discrete denoising posterior predictive (DDPP) objective that minimizes the mean squared error of a log-ratio between the denoising sub-trajectories of the amortized MDM sampler and the reward-induced target posterior,

$$\mathcal{L}_\tau^{\text{PP}} = \mathbb{E}_{t,\mathbf{x}_t} \left[ \mathbb{E}_{\tau(\mathbf{x}_{0:t})} [\| \log q_\theta(\mathbf{x}_{0:t-1} | \mathbf{x}_t, \hat{\mathbf{x}}_0)) - \log p_t^{\text{pre}}(\mathbf{x}_{0:t-1} | \mathbf{x}_t) + \kappa \|_2^2] \right], \quad (7)$$

where reward and the log partition function are captured in the constant $\kappa = \log \mathcal{Z}_{\pi_t}(\mathbf{x}_t) - \log R(\mathbf{x}_0)$. Interestingly, we can form an equivalent expression for the sub-trajectory loss $\mathcal{L}_\tau^{\text{PP}}$ above by sampling two intermediate points $\mathbf{x}_s, \mathbf{x}_{s-\gamma}$ in the sub-trajectory $\tau(\mathbf{x}_{0:t})$, such that $0 < s - \gamma < s < t$:

$$\mathcal{L}_\tau^{\text{PP}} = \mathbb{E}_{t,\mathbf{x}_t, \tau(\mathbf{x}_{0:t})} \left[ \| t \mathbb{E}_{s, \mathbf{x}_s, \mathbf{x}_{s-\gamma}} [\log q_\theta(\mathbf{x}_{s-\gamma} | \mathbf{x}_s, \hat{\mathbf{x}}_0) - \log p_t^{\text{pre}}(\mathbf{x}_{s-\gamma}, | \mathbf{x}_s, \hat{\mathbf{x}}_0^{\text{pre}}) + \kappa ] \|_2^2 \right]. \quad (8)$$

The proof for this equivalence is presented in §C.3. Note that we sample $s, s - \gamma \sim \mathcal{U}[0, t], \mathcal{U}[0, s]$ uniformly, and when $\gamma = 1/T$ we sample $\mathbf{x}_{s-1}$ which is simple to do since the $\tau(\mathbf{x}_{0:t})$ already contains this information. Crucially, unlike Eq. 7 the reformulation of the sub-trajectory loss in Eq. 8 is effectively a simulation-free version of Relative Trajectory Balance (RTB) (Venkatraman et al., 2024). If the approximation $q_{t,\theta}$ matches the denoising reward-induced target posterior over all sub-trajectories then the reverse process of $q_{t,\theta}$ can be simulated to draw samples that follow $\pi_0(\mathbf{x}_0)$.

Consequently, we term the $q_{t,\theta}$ that minimizes the DDPP objective in Eq. 8 as the *finetuned MDM* which solves the probabilistic inference task of sampling from $\pi_0(\mathbf{x}_0)$.

In contrast to learning MDMs in typical generative modeling setups, the DDPP objective requires the computation of the intractable log partition function $\log \mathcal{Z}_{\pi_t}(\mathbf{x}_t)$ evaluated at $\mathbf{x}_t$ which is a component of the term $\kappa$. This observation motivates the design of three concrete learning objectives for posterior matching, which as a collection we term the DISCRETE DENOISING POSTERIOR PREDICTION framework. Specifically, finetuning $q_{t,\theta}$ under a DDPP framework can be done in the following algorithms: 1.) DDPP-IS which uses a Monte Carlo based importance sampling estimate to approximate $\log \mathcal{Z}_{\pi_t}$ in Eq. 8, 2.) DDPP-LB that constructs a lower bound to DDPP-IS that is cheaper to evaluate by parameterizing $\log \mathcal{Z}_{\pi_t}$, and 3.) DDPP-KL which uses a discrete gradient estimator to bypass computing $\log \mathcal{Z}_{\pi_t}$ at the cost of requiring a differentiable reward—i.e. $\nabla R$.

### 3.1 ESTIMATING THE LOG PARTITION FUNCTION

Inspecting the posterior predictive objective in Eq. 8 we remark that it is a simulation-free stochastic regression objective which does not require a differentiable reward as the loss computes $R(\mathbf{x}_0)$ and not a gradient of the reward. Consequently, this makes the posterior predictive objective both a scalable and efficient objective for fine-tuning large pre-trained MDMs as long the reward model is easy to evaluate. Moreover, the posterior predictive objective is also an *off-policy* objective as it can be evaluated using any partially masked samples $\mathbf{x}_t \sim p(\mathbf{x}_t|\mathbf{x}_0)$. Practically, this means that fine-tuning can be performed using a replay buffer of samples from a biased dataset, e.g. the original training set for $p_0^{\text{pre}}$, or even partially masked sequences that arrive from a different model altogether. Despite its simple form the posterior predictive objective requires the computation of the log partition function of a partially masked sequence $\log \mathcal{Z}_{\pi_t}$ which does not have a closed-form expression and must be estimated.

**Monte Carlo Estimate of** $\log \mathcal{Z}_{\pi_t}$ **with DDPP-IS**. A numerical estimate of the log normalization constant can be obtained by utilizing the trick of using the pre-trained model's denoising posterior $p^{\text{pre}}(\mathbf{x}_0|\mathbf{x}_t)$. Specifically, given $\mathbf{x}_t \sim p_t(\mathbf{x}_t)$ we obtain a Monte Carlo estimate of $\log \mathcal{Z}_{\pi_t}(\mathbf{x}_t)$ that uses $M$ additional samples from $\mathbf{x}_0 \sim p_t^{\text{pre}}(\mathbf{x}_0|\mathbf{x}_t)$ to estimate the log partition function,

$$\log \hat{\mathcal{Z}}_{\pi_t}(\mathbf{x}_t) = \log \left( \sum_{\mathbf{x}_0,\dots\mathbf{x}_{t-1}} p_t^{\text{pre}}(\mathbf{x}_0,\dots,\mathbf{x}_{t-1}|\mathbf{x}_t) R(\mathbf{x}_0) \right) \approx \log \left( \mathbb{E}_{\mathbf{x}_0' \sim p_t^{\text{pre}}(\mathbf{x}_0|\mathbf{x}_t)}[R(\mathbf{x}_0')] \right).$$

Where in the second equality in the first line we used the fact that we can approximately jump to the endpoint of the reverse process directly by using the pretrained model's denoiser to sample $\mathbf{x}_0$. Conveniently, this MC estimate solely requires obtaining a denoised sample from the pre-trained MDM which can be efficiently done as each sample requires a single step as due to the denoising posterior parametrization of an MDM (Eq. 4). We can further improve the estimation of this log normalization constant by leveraging importance sampling (IS) with a proposal distribution $w(\mathbf{x}_0)$:

$$\log \hat{\mathcal{Z}}_{\pi_t}^{\text{IS}}(\mathbf{x}_t) = \log \left( \mathbb{E}_{\mathbf{x}_0' \sim w(\mathbf{x}_0)} \left[ \frac{p_t^{\text{pre}}(\mathbf{x}_0|\mathbf{x}_t) R(\mathbf{x}_0')}{w(\mathbf{x}_0')} \right] \right) = \log \left( \frac{1}{M} \sum_{j=1}^{M} \left[ \frac{p_t^{\text{pre}}(\mathbf{x}_0|\mathbf{x}_t) R(\mathbf{x}_0^j)}{w(\mathbf{x}_0^j)} \right] \right).$$

For the IS estimator above it is easy to verify that the optimal proposal distribution for variance reduction is proportional to the denoising reward-induced target posterior $w^*(\mathbf{x}_0) \propto \pi_t(\mathbf{x}_0|\mathbf{x}_t)$. Fortunately, this is precisely the distribution that is approximated by $q_{t,\theta}$ using the posterior predictive objective which motivates the reuse of the finetuned model as a suitable proposal, i.e. $w(\mathbf{x}_0) = q_{t,\theta}(\mathbf{x}_0|\mathbf{x}_t)$.

**Learning** $\log \mathcal{Z}_{\pi_t}$ **with DDPP-LB**. An alternative to using an MC-based estimate for $\log \mathcal{Z}_{\pi_t}$ is to parameterize the log partition function itself $\log \hat{\mathcal{Z}}_{\pi_t,\theta}^{\text{LB}}$ jointly with the $q_{t,\theta}$ and optimize both using the same posterior predictive objective as first defined in Eq. 11. Operationally, this amounts to including another prediction head for the finetuned MDM model and is cheaper to compute than using an MC-based estimate as we do not require $M$ evaluations of the pre-trained model as in $\log \hat{\mathcal{Z}}_{\pi_t}^{\text{IS}}(\mathbf{x}_t)$.

At first glance, it remains unclear whether a parameterized $\log \hat{\mathcal{Z}}_{\pi_t,\theta}^{\text{LB}}$ is a sensible strategy. However, in the particular case where we choose the proposal distribution to be on-policy by using finetuned MDM $w(\mathbf{x}_0) = q_{t,\theta}(\mathbf{x}_0|\mathbf{x}_t)$, we can show that the learned log partition function estimate is a lower bound to the importance sampling estimate. This is formalized in the following proposition below.

**Proposition 1.** *Let* $\log \hat{\mathcal{Z}}_{\pi_t}^{IS}$ *and* $\log \hat{\mathcal{Z}}_{\pi_t,\theta}^{LB}$ *be the $M$-sample importance sampling estimate using the proposal* $q_{t,\theta}(\mathbf{x}_0|\mathbf{x}_t)$ *and learned approximation to the log partition function respectively. Given*

---

**Algorithm 1** Single-step DDPP-IS and DDPP-LB

---

**Input**: Reward $R(\mathbf{x}_0)$, base MDM $p_0^{\text{pre}}(\mathbf{x}_0|\mathbf{x}_t)$, sampling policy $r(\mathbf{x}_0)$, fine-tuning MDM $q_\theta(\mathbf{x}_0|\mathbf{x}_t)$

1: **while** Training **do**
2:    $t, \mathbf{x}_0 \sim \mathcal{U}[0, 1], r(\mathbf{x}_0)$                              ▷ *Sample time and clean data on or off-policy*
3:    $\mathbf{x}_t \sim p_t(\mathbf{x}_t|\mathbf{x}_0)$                           ▷ *Construct partially masked sample given clean data*
4:    **if** Importance Sample $\log \mathcal{Z}(\mathbf{x}_t)$ **then**           ▷ *Log Partition Function Estimation Strategy*
5:       $\log \hat{\mathcal{Z}}_{\pi_t}(\mathbf{x}_t) := \log \hat{\mathcal{Z}}_{\pi_t}^{\text{IS}}(\mathbf{x}_t) = \log \left( \frac{1}{M} \sum_{j=1}^{M} \left[ \frac{p_t^{\text{pre}}(\mathbf{x}_0^j|\mathbf{x}_t) R(\mathbf{x}_0^j)}{w(\mathbf{x}_0^j)} \right] \right)$
6:    **else**
7:       $\log \hat{\mathcal{Z}}_{\pi_t}(\mathbf{x}_t) := \log \hat{\mathcal{Z}}_{\pi_t,\theta}^{\text{LB}}(\mathbf{x}_t)$
8:    $\mathcal{L}^{\text{PP}} = \left\| \left| \log q_{t,\theta}(\mathbf{x}_0|\mathbf{x}_t) - \log p_t^{\text{pre}}(\mathbf{x}_0|\mathbf{x}_t) - \log R(\mathbf{x}_0) + \log \hat{\mathcal{Z}}_{\pi_t}(\mathbf{x}_t) \right| \right\|_2^2$
9:    $\theta \leftarrow \text{Update}(\theta, \nabla_\theta \mathcal{L}^{\text{PP}})$
10: **Return** $q_\theta$

---

*a partially masked sample $\mathbf{x}_t \sim p_t(\mathbf{x}_t)$ the optimal learned approximation is a lower bound to the importance sampling estimate with a fixed proposal $q_{t,\theta}(\mathbf{x}_0|\mathbf{x}_t)$ and the following inequality holds:*

$$\log \hat{\mathcal{Z}}_{\pi_t,\theta}^{LB}(\mathbf{x}_t) \leq \log \hat{\mathcal{Z}}_{\pi_t}^{IS}(\mathbf{x}_t). \tag{9}$$

The proof for Eq. 1 is provided in §C.1. We highlight that the lower bound becomes equality at the optimal proposal $q_{t,\theta}(\mathbf{x}_0|\mathbf{x}_t) \propto \pi_t(\mathbf{x}_0|\mathbf{x}_t)$. Learning $\log \hat{\mathcal{Z}}_{\pi_t,\theta}^{\text{LB}}$ has the benefit of amortization as the same network can be reused for all partially masked samples $\mathbf{x}_t \sim p_t(\mathbf{x}_t)$, across all levels of masking. In addition, over the course of training, the learned estimate $\log \hat{\mathcal{Z}}_{\pi_t,\theta}^{\text{LB}}$ becomes a better estimate for the true log partition function. In practice, it suffices to take a single gradient step to optimize $\log \hat{\mathcal{Z}}_{\pi_t,\theta}^{\text{LB}}$ rather than optimizing till convergence. As a result, no additional overhead needs to be incurred, and the learned estimate is averaged over a batch of noisy samples $\mathcal{B} = \{\mathbf{x}_t^i\}_{i=1}^N$.

### 3.2 SINGLE-STEP POSTERIOR SAMPLING WITH ENDPOINT PREDICTION

The sub-trajectory matching objective used by DDPP-IS and DDPP-LB can be simplified to a faster single-step objective at the cost of paying a discretization error by not using finer-grained trajectory information. Specifically, we note that for MDMs the denoising posterior over endpoints $q_{t,\theta}(\mathbf{x}_0|\mathbf{x}_t) \approx \text{Cat}(\mathbf{x}_0; \mu_\theta(\mathbf{x}_t, t))$ can be approximately computed *without unrolling the sub-trajectory*. This fact also holds for the pre-trained MDM as the model parametrization implies $p_t^{\text{pre}}(\mathbf{x}_0|\mathbf{x}_t) \approx \text{Cat}(\mathbf{x}_0; \mu(\mathbf{x}_t, t))$. For the single-step objective we assume the parameterized denoisers exactly match the posteriors. Leveraging this enables us to express the denoising reward-induced target posterior using a simple expression that directly uses the pre-trained model's denoising posterior $p_t^{\text{pre}}(\mathbf{x}_0|\mathbf{x}_t)$ as follows:

$$\pi_t(\mathbf{x}_0|\mathbf{x}_t) = \frac{p_t(\mathbf{x}_t)}{p_t(\mathbf{x}_t)} \cdot \frac{p_t(\mathbf{x}_t|\mathbf{x}_0) p^{\text{pre}}(\mathbf{x}_0) R(\mathbf{x}_0)}{\sum_{\mathbf{x}_0'} p_t(\mathbf{x}_t|\mathbf{x}_0') p^{\text{pre}}(\mathbf{x}_0') R(\mathbf{x}_0')} = \frac{p_t^{\text{pre}}(\mathbf{x}_0|\mathbf{x}_t) R(\mathbf{x}_0)}{\mathcal{Z}_{\pi_t}(\mathbf{x}_t)}. \tag{10}$$

The choice of parameterizing $q_{t,\theta}(\mathbf{x}_0|\mathbf{x}_t)$ as another MDM offers a prescriptive strategy for sampling from the desired target $\pi_0$ by learning to match the denoising reward-induced posterior at the predicted endpoint $\pi_t(\mathbf{x}_0|\mathbf{x}_t)$. This simplifies the expression of DDPP defined over trajectories in Eq. 8 to a single point, namely the predicted endpoint $\mathbf{x}_0$ of each MDM. This objective is presented below:

$$\mathcal{L}^{\text{PP}} = \mathbb{E}_{t,\mathbf{x}_0,\mathbf{x}_t} \left[ \left\| \left| \log q_{t,\theta}(\mathbf{x}_0|\mathbf{x}_t) - \underbrace{\log p_t^{\text{pre}}(\mathbf{x}_0|\mathbf{x}_t) - \log R(\mathbf{x}_0) + \log \mathcal{Z}_{\pi_t}(\mathbf{x}_t)}_{\log \pi_t(\mathbf{x}_0|\mathbf{x}_t)} \right| \right\|_2^2 \right]. \tag{11}$$

As done previously, we can employ any estimation strategy to compute the log partition function Eq. 11. We note in many cases, such as when the sequence length of the trajectory is small to moderate, the single-step objective may be an attractive alternative to the sub-trajectory variants of DDPP. Algorithm 1 provides a detailed description of the single-step version of DDPP.

### 3.3 DDPP-KL: POSTERIOR PREDICTION VIA REVERSE KL MINIMIZATION

The single-step posterior prediction objective as defined using the loss function $\mathcal{L}^{\text{PP}}$ in Eq. 11 requires the estimation of $\log \mathcal{Z}_{\pi_t,\theta}^{\text{LB}}$ which introduces a source of variance in loss estimates that may sub-

optimally influence learning dynamics of the fine-tuned model. In settings where the reward model is differentiable, we can bypass computing $\log \mathcal{Z}_{\pi_t,\theta}^{\text{LB}}$ altogether by learning to match the denoising reward-induced posterior under the *reverse* KL divergence. Note that the *forward* KL divergence is inapplicable here as we do not have samples from $\pi_0$—i.e. a dataset. To see this, we define a variational posterior matching problem using the reverse KL divergence that takes the following form:

$$\mathcal{L}_t^{\text{KL}} := \mathbb{D}_{\text{KL}}(q_{t,\theta}(\mathbf{x}_0|\mathbf{x}_t)p_t(\mathbf{x}_t)||\pi_t(\mathbf{x}_0|\mathbf{x}_t)p_t(\mathbf{x}_t)). \tag{12}$$

Unlike conventional generative modeling using the reverse KL divergence which solely matches distributions at $t = 0$ the problem definition in Eq. 12 defines a series of reverse KL minimization problems through time. In this manner, the reverse KL matches distributions annealed through time and can be used to derive a stochastic regression objective for fine-tuning,

$$\mathcal{L}^{\text{KL}} = \mathbb{E}_{t,\mathbf{x}_0,\mathbf{x}_t} \left[ \log q_{t,\theta}(\mathbf{x}_0|\mathbf{x}_t) - \log p_t^{\text{pre}}(\mathbf{x}_0|\mathbf{x}_t) - \log R(\mathbf{x}_0) \right] + C. \tag{13}$$

The expectation in Eq. 13, like DDPP-IS and DDPP-LB is taken uniformly with respect to time $t \sim \mathcal{U}[0,1]$. However, unlike the previous estimators, clean data needed to compute $\mathcal{L}^{\text{KL}}$ is drawn purely on-policy by simulating the fine-tuning model $\mathbf{x}_0 \sim q_{t,\theta}(\mathbf{x}_0)$, which then also allows us to craft a noisy sample using the masking forward process $\mathbf{x}_t \sim p_t(\mathbf{x}_t|\mathbf{x}_0)$. Additionally, in Eq. 13 the constant $C = \mathbb{E}_{t,\mathbf{x}_0,\mathbf{x}_t}[\log \mathcal{Z}_{\pi_t}(\mathbf{x}_t)]$ does not depend on the $\theta$—and as a result is also independent of the sample $\mathbf{x}_0 \sim q_{t,\theta}(\mathbf{x}_0)$. This results in the constant $C$ being zero when computing the gradient of the loss $\nabla_\theta \mathcal{L}^{\text{KL}}$ and as a result we can safely disregard computing $\log \mathcal{Z}_{\pi_t}$ entirely.

As samples $\mathbf{x}_0$ are procured on-policy to compute the gradient of the loss $\nabla_\theta \mathcal{L}^{\text{KL}}$ we require backpropagating through the stochastic sampling of $\mathbf{x}_0$ which comes from simulating the fine-tuning MDM $q_{t,\theta}(\mathbf{x}_0)$. Fortunately, we can make use of modern discrete gradient estimators which provide a biased but low variance gradient estimate enabling us to compute $\mathcal{L}^{\text{KL}}$. Specifically, we opt to use the scalable 2nd order REINMAX estimator (Liu et al., 2024) which estimates the discrete gradient up to second-order terms in a Taylor approximation of the actual gradient. We note that unlike DDPP-IS and DDPP-LB this new loss that minimizes the reverse KL divergence $\mathcal{L}^{\text{KL}}$ requires the reward model $R$ to be differentiable and as a result is less broadly applicable than computing $\mathcal{L}^{\text{PP}}$. However, in practice, learning can be faster as we make use of the information afforded to us by the gradient $\nabla R$ as well as the fact that the objective does not need to estimate the log partition function.

In appendix §C.2 we provide the exact algorithm Alg. 2 to compute the reverse KL objective. We further show how using a gradient estimator like REINMAX can be used to derive efficient gradient estimation for a more general class of problems of sampling from $\pi_0(\mathbf{x}_0) = R(\mathbf{x}_0)/\mathcal{Z}_{\pi_0}$, as well as the main fine-tuning setting for matching the denoising reward-induced posterior as defined in Eq. 10.

## 4 EXPERIMENTS

We investigate the application of DDPP to a variety of discrete generative modeling settings. We provide the full experimental details in §D and present our main experimental results next.

Table 1: Overview of posterior sampling methods

| Method | Model calls / inf. step | Model calls / train step | Sim. Free |
|---|---|---|---|
| SVDD | $N$ | — | ✓ |
| Discrete guidance | 1 | — | ✓ |
| RTB | 1 | $T$ | ✗ |
| DDPP-KL | 1 | 1 | ✓ |
| DDPP-IS | 1 | $M$ | ✓ |
| DDPP-LB | 1 | 1 | ✓ |

**Baselines**. Throughout our experiments, we rely on four principal baselines in: sampling from the pre-trained MDM model, Best-of-N sampling (Stiennon et al., 2020), Relative Trajectory Balance (RTB) (Venkatraman et al., 2024), and SVDD (Li et al., 2024) which is a concurrent inference time technique for steering diffusion models. Best-of-N represents a computationally expensive baseline but is guaranteed to produce samples from $\pi_0$, as such we use this as an upper bound on performance in terms of reward obtained as $N \to \infty$ (Beirami et al., 2024). RTB is a GFlowNet (Bengio et al., 2023; Madan et al., 2022; Lahlou et al., 2023) that requires simulating the entire diffusion trajectory. For image settings with differentiable reward, we also include discrete guidance as a baseline (Nisonoff et al., 2024). In Table 1 we illustrate the computational differences between DDPP and baselines.

### 4.1 SYNTHETIC EXPERIMENTS

We consider a synthetic task of learning to sample from a target distribution on a 2D discrete grid and finetuning an MDM on binarized MNIST. This synthetic setting tests all DDPP variations with chosen baselines, presenting qualitative results in Figure 6, Figure 4 and quantitative results in Table 2.

**Grid Experiment**. We define a prior density $p_0^{\text{pre}}$ over the discrete 2-dimensional, $128 \times 128$ grid, as showcased in Figure 6(a) where the probability mass corresponding to each point $\mathbf{x}_0$ is on if the color is yellow. The goal is to sample from the product distribution as outlined in Equation 1, which in this case



| (a) Prior Density | (b) Target Density | (c) DDPP-IS | (d) DDPP-LB | (e) DDPP-KL |

Figure 1: Samples generated by fine-tuning a masked diffusion model to sample from the lower half of its prior distribution. Samples $\mathbf{x}_0$ in this setting are 2-dimensional, with a vocabulary size of 128.

is defined to drop the modes in $p_0^{\text{pre}}$ which are at the top half of the grid, as visualized in Figure 6(b). These results show that all three variants of DDPP effectively learn to sample from this target.

**MNIST**. We finetune MDMs to generate even MNIST digits. As observed in Table 2 we find that all three variants of DDPP match or outperform the base pre-trained model and RTB in all metrics, with DDPP-KL being the best. In comparison to the concurrent work of SVDD, we find that it outperforms DDPP in average $\log R$ but is worse in sample-based metrics such as class conditional FLD (Jiralerspong et al., 2023) which measures the overall quality, diversity and generalizability of generated samples and class conditional BPD. We further report generated samples in Figure 4 located in §D.3.

## 4.2 PIXEL-LEVEL IMAGE MODELLING

We fine-tune MDMs on order-agnostic image data, discretizing pixels in $64 \times 64$ downsampled CelebA images (Liu et al., 2018) to a vocabulary of 256 tokens. As there are no publicly available pre-trained MDM models we train our own MDM by modeling the raw pixel space and achieve 1.85 bits-per-dim (BPD) on CelebA. Our full experimental setup is outlined in §D.3. For fine-tuning, we consider steering a pre-trained MDM using DDPP-LB as it is the

Table 2: Fine-tuning to produce only even digits on binarized MNIST. We report the mean performance over 3 runs for the $\log R$, FLD, and BPD metrics.

| Algorithm ↓ Metric → | $\log R(\mathbf{x}_0)$ ↑ | FLD ↓ | BPD ↓ |
|---|---|---|---|
| Base Model | -26.90 ± — | 33.89 ± — | 0.130 ± — |
| SVDD | **-0.03 ± 0.01** | 34.19 ± 0.95 | — |
| Guidance (scale 1) | -25.24 ± 0.26 | 34.67 ± 0.67 | 0.171 ± 0.001 |
| Guidance (scale 5) | -23.21 ± 0.21 | 37.33 ± 0.87 | 0.174 ± 0.001 |
| Guidance (scale 100) | -9.32 ± 0.24 | 72.19 ± 0.43 | 0.147 ± 0.001 |
| RTB | -18.66 ± 2.45 | 45.97 ± 0.89 | **0.128 ± 0.000** |
| DDPP-IS (*ours*) | -5.14 ± 1.24 | 33.11 ± 0.71 | 0.130 ± 0.000 |
| DDPP-LB (*ours*) | -5.68 ± 0.34 | 33.76 ± 0.90 | **0.128 ± 0.000** |
| DDPP-KL (*ours*) | -3.13 ± 0.06 | **31.75 ± 0.51** | 0.129 ± 0.000 |

most computationally cheap method with a class-conditional reward based on an auxiliary classifier. Specifically, we steer the generative model to generate human faces with blond hair. For quantitative metrics, we report the mean log reward obtained, and BPD in Figure 3 as well as selected generated samples. Our quantitative results show that our proposed variant DDPP-LB significantly outperforms all other baselines in obtaining the highest reward. We also observe DDPP obtains BPD values that are within the range of the base model while being worse than RTB. We further find visual samples produced by DDPP to have the highest fidelity faces with blond hair, matching our fine-tuning goal.

## 4.3 PROTEIN SEQUENCE MODELLING

Table 3: *In-silico* results for protein generation tasks. We report the mean result for a metric with standard deviation across three seeds. DDPP-LB performs well across designability metrics (pLDDT and pTM) while simultaneously performing best on task specific metrics ($\beta$-sheet % and TM-Score).

| | High $\beta$-sheet-content protein generation | | | | Protein shrinking | | | | |
|---|---|---|---|---|---|---|---|---|---|
| | $\beta$-sheet % ↑ | pLDDT ↑ | pTM ↑ | $\log R(x_0)$ ↑ | SS-KL ↓ | TM-Score ↑ | pLDDT ↑ | pTM ↑ | $\log R(x_0)$ ↑ |
| Base Model | 0.111 ± 0.121 | 0.724 ± 0.144 | 0.584 ± 0.226 | 2.070 ± 0.749 | 3.040 ± 3.043 | 0.245 ± 0.058 | 0.724 ± 0.144 | 0.584 ± 0.226 | 0.490 ± 0.116 |
| Best-of-10 | 0.280 ± 0.093 | 0.812 ± 0.033 | 0.786 ± 0.035 | 3.212 ± 0.371 | 1.621 ± 2.804 | 0.345 ± 0.049 | 0.786 ± 0.023 | 0.737 ± 0.097 | 0.690 ± 0.098 |
| SVDD | 0.114 ± 0.148 | 0.484 ± 0.134 | 0.349 ± 0.174 | 1.669 ± 0.907 | 3.353 ± 2.913 | 0.337 ± 0.042 | 0.492 ± 0.131 | 0.368 ± 0.171 | 0.673 ± 0.083 |
| RTB | 0.319 ± 0.218 | 0.806 ± 0.059 | 0.767 ± 0.101 | 3.386 ± 1.061 | 2.193 ± 2.724 | 0.290 ± 0.056 | **0.797 ± 0.056** | 0.747 ± 0.093 | 0.581 ± 0.112 |
| DDPP-LB | **0.436 ± 0.037** | **0.897 ± 0.027** | **0.806 ± 0.029** | **3.703 ± 0.186** | **0.640 ± 1.793** | **0.361 ± 0.047** | 0.768 ± 0.048 | **0.747 ± 0.063** | **0.722 ± 0.094** |

**Task description**. We next apply DDPP to generate high-quality protein sequences by fine-tuning discrete diffusion protein language models (DPLM) (Wang et al., 2024). Specifically, we address two experimentally relevant tasks where vanilla DPLMs underperform. We outline exact reward functions and experimental setup in §D.2. First, we fine-tune DPLM to generate soluble protein sequences with high $\beta$-sheet content. The second task, protein shrinking, involves miniaturizing known proteins by generating shorter sequences that preserve key structural features, using the TM-align score as the reward metric (Devkota et al., 2024). We evaluate performance by measuring designability metrics (ESMFold pLDDT and pTM) as well as task-specific metrics ($\beta$-sheet percent and TM-Score). We also provide wet-lab validation for our best designs in the designable $\beta$-sheet task. We provide a

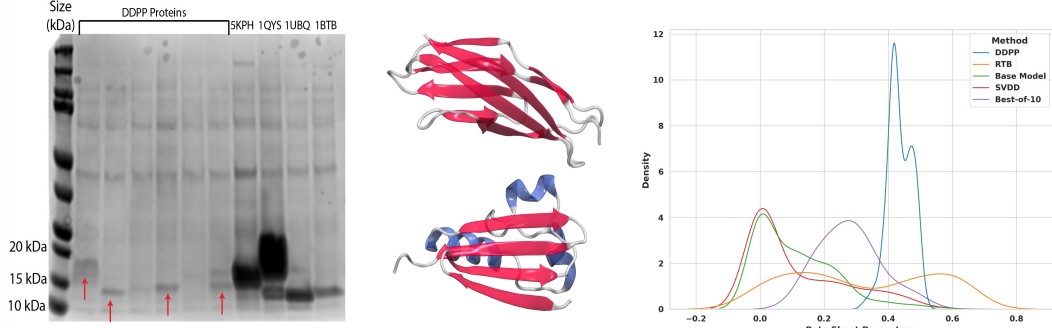

Figure 2: **Left:** SDS-PAGE of elution fractions from histidine tag purification of DDPP-designed protein constructs and positive controls following Coomassie blue staining. All DDPP-designed constructs are between 7.8-8.3 kDa. Predicted molecular weights of positive controls 5KPH, 1QYS, 1UBQ, and 1BTB are 10 kDa, 12 kDa, 8.5 kDa, and 10 kDa, respectively. Recombinant protein bands for MDM-designed sequences are indicated with red arrows and relevant ladder references are labeled with their molecular weight. **Middle:** Folded structures generated by DDPP $\beta$-sheet fine-tuning. **Right:** Distribution of $\beta$-sheets generated by each method.

deeper description of evaluation metrics and experimental setup in §D.2. Finally, as ESMFold is itself expensive to query and, in particular, non-differentiable we test our fastest method—DDPP-LB.

**Main results**. In-silico validation shows that DDPP-LB outperforms all baselines for the designable $\beta$-sheet task, generating better sequences across all metrics. In particular DDPP achieves a significantly higher $\beta$-sheet percentage than baseline methods while maintaining high designability as measured by ESMFold (namely, high pLDDT and pTM). We further observe that for the miniaturization task, DDPP-LB outperforms all baselines in shrinking ribonuclease proteins, removing 34 residues while maintaining high structural similarity (lowest SS-KL of 0.64 and highest TM-Score of 0.361), and high structural quality with high pTM and competitive pLDDT. This demonstrates DDPP-LB's effectiveness in generating compact yet structurally faithful proteins.

**Experimental validation**. We selected 6 designs from DDPP-finetuned DPLM for wet-lab validation, based on AlphaFold2 pLDDT/pTM scores. Sequences and structures were clustered using MMseqs and Foldseek (van Kempen et al., 2022; Steinegger and Söding, 2017), with two representative sequences selected from each cluster. 4 positive controls consisting of two previously validated de novo designed proteins (PDB: 5KPH, 1QYS) and two other stable proteins, ubiquitin and Barstar (PDB: 1UBQ, 1BTB) were included as a comparison. We expressed the designed proteins, including the controls in E. coli, and purified them using histidine-tag purification, after which we assessed expression level and purity via SDS-PAGE, followed by Coomassie staining. Our results demonstrate strong overexpression and efficient purification of the two previously validated de novo controls and moderate overexpression of ubiquitin and barstar controls (Figure 2). Purified protein can also be observed for four out of the six DDPP-derived constructs, though with comparatively lower yields than the positive controls (Figure 2). One potential cause of these relatively low yields may be the sizeable accumulation of DDPP-derived proteins in the insoluble fraction of the cell lysate. As such, it is likely that further optimization of the expression and purification methods (e.g., longer induction time or lower induction temperatures) may lead to significant improvements to overall soluble yields.

## 4.4 TEXT

**Task description**. We consider two text tasks: (i) toxic story generation using the Tinystories dataset (Eldan and Li, 2023), and (ii) product review generation using Amazon data (Hou et al., 2024). For both tasks, we start by fine-tuning a pre-trained MDM model (Sahoo et al., 2024) in a supervised fine-tuning manner on both datasets before running online fine-tuning. As reward models, we use RoBERTa (Liu, 2019) fine-tuned for toxicity classification, and BERT (Devlin, 2018), fine-tuned for Amazon review sentiment analysis, respectively. Our experiments aim to demonstrate our method's ability to induce behaviors that are uncommon in the base pre-trained model, specifically in generating toxic content in product reviews. Full experimental details are provided in Appendix §D.4.

**Main results**. In Table 4 we report the average log reward as well as perplexity (Gen PPL) of the generated samples as measured by GPT-2 (Radford et al., 2019). We find that DDPP-LB is the most effective variant of DDPP and achieves significantly higher log reward compared to SVDD and RTB for both tasks. We further observe that all methods achieve comparable Gen PPL suggesting

| Algorithm ↓ Metric → | $\log R(\mathbf{x}_0) \uparrow$ | BPD ↓ |
|---|---|---|
| Base | -57.31 ± — | 2.67 ± — |
| SVDD | -13.27 ± 12.38 | — |
| Guidance (scale 1) | -91.10 ± 1.63 | 3.20 ± 0.01 |
| Guidance (scale 5) | -62.75 ± 0.25 | 5.39 ± 0.01 |
| Guidance (scale 100) | -41.51 ± 0.02 | 5.15 ± 0.00 |
| RTB | -60.28 ± 1.74 | **2.04 ± 0.00** |
| DDPP-LB (*ours*) | **-6.94 ± 1.39** | 2.62 ± 0.15 |

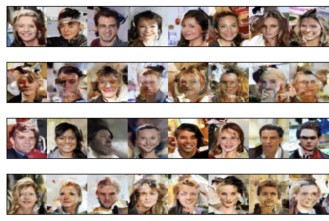

Figure 3: **Left:** Results for discrete image modeling over raw pixel values on CelebA ($64 \times 64$). We report the mean performance of DDPP and baselines separated into inference-based (top) and amortized (bottom) over 3 runs for the $\log R$ and class-BPD metrics. **Right:** Generated samples from Base, SVDD, RTB, and DDPP-LB.

that generated responses are fluent; however, samples from DDPP-LB adheres better to the task specification. We refer to §D.4.1 and §D.4.2 for generated samples from DDPP.

Table 4: Text experiments with log reward and Gen PPL results averaged over 3. As Best of 10 draws samples directly from $p_0^{\text{pre}}(\mathbf{x}_0)$ we instead bold the fine-tuning method whose Gen PPL is lowest.

| Dataset → | Tinystories | | Amazon reviews | |
|---|---|---|---|---|
| Algorithm ↓ Metric → | $\log R(\mathbf{x}_0) \uparrow$ | Gen PPL ↓ | $\log R(\mathbf{x}_0) \uparrow$ | Gen PPL ↓ |
| Best of 10* | 93.25 ± 0.17 | 15.94 ± 0.03 | -103.05 ± 0.25 | 124.45 ± 1.02 |
| SVDD | 146.95 ± 1.08 | 20.35 ± 0.03 | -27.48 ± 10.91 | 165.86 ± 1.22 |
| RTB | 107.83 ± 3.08 | **18.53 ± 0.55** | -35.22 ± 16.03 | 160.54 ± 12.19 |
| DDPP-IS (*ours*) | 163.45 ± 7.06 | 20.15 ± 0.30 | 105.16 ± 2.41 | **152.85 ± 1.64** |
| DDPP-LB (*ours*) | **205.76 ± 3.88** | 19.60 ± 0.69 | **152.08 ± 34.01** | 167.25 ± 27.33 |

## 5 RELATED WORKS

**Discrete diffusion**. The prevailing paradigms for diffusion over discrete spaces can be broadly categorized into 1.) continuous diffusion in a latent or reparametrized space by first transforming the initial discrete data (Li et al., 2022; Chen et al., 2022; Davis et al., 2024; Cheng et al., 2024), and 2.) defining diffusion using discrete analogs of score approximation (Meng et al., 2022; Lou et al., 2023). The latter approach can also be described using the theoretical framework of Continuous-time Markov Chains (CTMC) (Austin et al., 2021; Campbell et al., 2022; 2024). Closest to our setting we consider a specific instantiation of discrete diffusion that simplifies the CTMC framework by using a masked forward process (Sahoo et al., 2024; Shi et al., 2024; Zhao et al., 2024a; Gat et al., 2024).

**Finetuning as sampling**. The task of fine-tuning generative models under reward models can be viewed as a sampling problem and encompasses conventional RLHF (Uehara et al., 2024a; Black et al., 2023; Fan et al., 2024; Dong et al., 2023). A simple but expensive method to sample from the reward-induced Bayesian posterior distribution is best of $N$ sampling (Stiennon et al., 2020), which provably samples from the correct distribution as the number of samples from the base pre-trained model grows, $N \to \infty$ (Beirami et al., 2024; Gao et al., 2023; Ferbach et al., 2024). Alternatively, the sampling perspective has been explored in the discrete setting to fine-tune autoregressive models (Zhao et al., 2024a; Hu et al., 2023), and diffusion models (Uehara et al., 2024b; Venkatraman et al., 2024; Zhao et al., 2024a). Finally, inference time techniques represent the most prominent approach to conditional sampling (Ho and Salimans, 2022; Dhariwal and Nichol, 2021; Li et al., 2024; Nisonoff et al., 2024).

## 6 CONCLUSION

In this paper, we present DISCRETE DENOISING POSTERIOR PREDICTION a novel framework to steer Masked Discrete Diffusion Models by viewing it as a problem of sampling from a Bayesian posterior. We introduced three concrete training strategies to instantiate our framework in DDPP-IS, DDPP-LB, and DDPP-KL and apply them to modeling synthetic data, pixel-level image modeling, fine-tuning protein MDMs to increase secondary structure diversity, and steering MDMs on language to match human sentiment. We find that DDPP not only is able to optimize an amortized sampler to closely match the reward-induced Bayesian posterior but it has a good agreement in other sample quality metrics—without severely compromising generated sample quality. An interesting direction for future work is to understand how to balance optimization of DDPP-LB and strategies to selecting $\gamma$.

## 7 CONTRIBUTIONS STATEMENT

J.R. conceived of the initial simulation-free fine-tuning framework, while M.H. derived the final version of the objective that can be written as a log ratio of denoisers. J.R. drove the development of code for all experimental settings, M.H. and A.T. drove the completion of image experiments, J.R. and Z.P. drove the completion of protein and text experiments, and M.H. drove the completion of toy experiments. Z.P. led TM-score protein experiments. J.R., C.L., Z.Q., and P.C. conceived of the $\beta$-sheet protein task and wet lab experimental design. Z.Q. ran all wet lab experiments. N.D. guided text experiments. S.M. helped in setting up the baselines and brainstorming about the estimators. A.J.B. drove the writing of the paper with help from M.H.. M.B., Y.B., A.T., S.M. and A.J.B. guided the project from the machine learning side, while P.C. guided it from the wet lab side. A.J.B. with help from M.H. designed the discrete (Reinmax) gradient estimator version of the objective function. J.R., A.J.B., and A.T. were responsible for the overall organization of the project.

## 8 ACKNOWLEDGEMENTS

The authors would like to thank Kolya Malkin, Moksh Jain, Emily Jin, Katarina Petrovic, Scott Le Roux, Ahmed Elhag, Xingyue Huang, and Vignesh Ram Somnath for their useful comments on early versions of this manuscript. AJB is partially supported by an NSERC Post-doc fellowship. This research is partially supported by EPSRC Turing AI World-Leading Research Fellowship No. EP/X040062/1 and EPSRC AI Hub on Mathematical Foundations of Intelligence: An "Erlangen Programme" for AI No. EP/Y028872/1

The authors acknowledge funding from UNIQUE, CIFAR, NSERC, Intel, Samsung, and Dreamfold. The research was enabled in part by computational resources provided by the Digital Research Alliance of Canada (https://alliancecan.ca), Mila (https://mila.quebec), and NVIDIA.

## 9 REPRODUCIBILITY STATEMENT

We take the following steps to enhance the reproducibility of our work. In particular, all of our theoretical results include full proofs which are presented in §C. To assist in the reproducibility of our empirical findings we provide precise experimental details such as algorithmic descriptions of all variants of DDPP in Algorithm 1 and Algorithm 2. We further provide architectural choices, training details, and hyperparameters for all datasets and tasks in §D. Code is available at https://github.com/jarridrb/ddpp.

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

## A    BROADER IMPACT

Our proposed DISCRETE DENOISING POSTERIOR PREDICTION is a tailored approach to steering and fine-tuning Masked Diffusion Models. At present, MDMs are an emergent category of discrete generative models that have general-purpose modeling capabilities in a variety of domains including language modeling, sequence-based drug design, and discrete modeling of graphs. Consequently, we believe DDPP has potential use in various practical use cases. For instance, like current RLHF techniques applied to modern autoregressive LLMs, future scaled MDMs on text datasets might be tuned to promote harmful behavior and toxic content. Moreover, applying DISCRETE DENOISING POSTERIOR PREDICTION in drug design use cases has the potential to create in-silico sample of protein sequences that may have biologically potent negative externalities. We do, however, make the distinction that such a risk is speculative at this stage given the large complexities of translating in-silico designs to actual synthesized biomolecules. As a result, we encourage practitioners who seek to fine-tune MDMs using DDPP to exercise due caution when applying our proposed techniques to actual use cases.

**Ethical statement**. As part of qualitatively evaluating DDPP, this paper includes generated samples of text. We highlight that the set of examples may contain potentially disturbing, harmful, or upsetting examples, covering a variety of sensitive topics like discriminatory language, descriptions of harm, and misinformation, among other high-risk categories. Its primary purpose is to advance research in understanding the impact of DDPP from a more interpretable lens. It is not advised to train future MDMs on such generated samples in order to prevent further propagation of undesirable content and behaviors.

## B    ADDITIONAL RELATED WORK

**Sampling proportional to energy**.  Our approach can be closely linked to learning to sample proportional to a target probability, as in our setup we aim to approximate sampling proportional to the energy $p_t^{\text{pre}}(\cdot|\mathbf{x}_t)R(\cdot)$ for any point $\mathbf{x}_t$ at any time $t$. This has been an avenue of research for a number of works in continuous time (Bengio et al., 2021; 2023; Malkin et al., 2022; Lahlou et al., 2023; Akhound-Sadegh et al., 2024; Sendera et al., 2024; De Bortoli et al., 2024), in Bayesian posterior inference where the energy is defined by the product of likelihood and prior (Mittal et al., 2023), as well as posterior inference in settings where we even do not have access to energy function but only to a simulator (Radev et al., 2020; Wildberger et al., 2024; Geffner et al., 2023).

## C    THEORETICAL RESULTS

### C.1    PROOF OF PROPOSITION 1

Before proving proposition 1 we first prove a useful Lemma that states the optimal log partition function $\log \hat{\mathcal{Z}}_{\pi_t}(\mathbf{x}_t)$ which is the learning goal for a parameterized approach $\log \hat{\mathcal{Z}}_{\pi_t,\theta}(\mathbf{x}_t)$.

**Lemma 1.** *Given a sample* $\mathbf{x}_t \sim p_t(\mathbf{x}_t|\mathbf{x}_0)$ *and the denoising posterior distribution* $q_{t,\theta}(\mathbf{x}_0|\mathbf{x}_t)$, *a local minimizer for estimate for the log partition function* $\log \hat{\mathcal{Z}}_{\pi_t}$ *using* $N$ *samples from* $\mathbf{x}_0^i \sim q_{t,\theta}(\mathbf{x}_0|\mathbf{x}_t)$ *is given by:*

$$\log \mathcal{Z}_{\pi_t}^* = \frac{1}{N}\sum_{i=1}^{N}\log\left(\frac{p_t(\mathbf{x}_0^i|\mathbf{x}_t)R(\mathbf{x}_0^i)}{q_{t,\theta}(\mathbf{x}_0^i|\mathbf{x}_t)}\right). \tag{14}$$

*Proof.* By definition the log partition function is a constant, let that constant be $\log \mathcal{Z}_{\pi_t}(\mathbf{x}_t) = C$. Then the loss in Eq. 11 is a quadratic in $C$,

$$\mathcal{L} = \mathbb{E}_{\mathbf{x}_0 \sim r(\mathbf{x}_0)}\left[||\log q_{t,\theta}(\mathbf{x}_0|\mathbf{x}_t) + C - \log p_t(\mathbf{x}_0|\mathbf{x}_t) - \log R(\mathbf{x}_0)||_2^2\right] \tag{15}$$

For a batch of $N$ samples of $\mathbf{x}_0^i \sim q_{t,\theta}(\mathbf{x}_0|\mathbf{x}_t)$, we find a locally optimal constant $C$ (local minima) by taking the gradient of Eq. 15 and setting it 0. In more detail we have,

$$0 = \nabla_C \frac{1}{N} \sum_i^N (\log q_{t,\theta}(\mathbf{x}_0|\mathbf{x}_t) + C - \log p_t(\mathbf{x}_0|\mathbf{x}_t) - \log R(\mathbf{x}_0))^2 \tag{16}$$

$$0 = \frac{2}{N} \sum_i^N \left( \log q_{t,\theta}(\mathbf{x}_0^i|\mathbf{x}_t) + C - \log p_t(\mathbf{x}_0^i|\mathbf{x}_t) - \log R(\mathbf{x}_0^i) \right) \tag{17}$$

$$0 = 2C + \frac{2}{N} \sum_i^N \log q_{t,\theta}(\mathbf{x}_0^i|\mathbf{x}_t) - \log p_t(\mathbf{x}_0^i|\mathbf{x}_t) - \log R(\mathbf{x}_0^i) \tag{18}$$

$$0 = C + \frac{1}{N} \sum_i^N \log \left( \frac{q_{t,\theta}(\mathbf{x}_0^i|\mathbf{x}_t)}{p_t(\mathbf{x}_0^i|\mathbf{x}_t) R(\mathbf{x}_0^i)} \right) \tag{19}$$

$$C = \frac{1}{N} \sum_i^N \log \left( \frac{p_t(\mathbf{x}_0^i|\mathbf{x}_t) R(\mathbf{x}_0^i)}{q_{t,\theta}(\mathbf{x}_0^i|\mathbf{x}_t)} \right). \tag{20}$$

$\square$

Using Lemma 1 we now prove Proposition 1, stated again below for convenience.

**Proposition 1.** *Let $\log \hat{\mathcal{Z}}_{\pi_t}^{IS}$ and $\log \hat{\mathcal{Z}}_{\pi_t,\theta}^{LB}$ be the $M$-sample importance sampling estimate using the proposal $q_{t,\theta}(\mathbf{x}_0|\mathbf{x}_t)$ and learned approximation to the log partition function respectively. Given a partially masked sample $\mathbf{x}_t \sim p_t(\mathbf{x}_t)$ the optimal learned approximation is a lower bound to the importance sampling estimate with a fixed proposal $q_{t,\theta}(\mathbf{x}_0|\mathbf{x}_t)$ and the following inequality holds:*

$$\log \hat{\mathcal{Z}}_{\pi_t,\theta}^{LB}(\mathbf{x}_t) \le \log \hat{\mathcal{Z}}_{\pi_t}^{IS}(\mathbf{x}_t). \tag{9}$$

*Proof.* We optimize $\log \hat{\mathcal{Z}}_{\pi_t,\theta}^{\mathrm{LB}}(\mathbf{x}_t)$ using the loss defined in Eq. 11. Using Lemma 1 we know the analytic expression for the locally optimal estimate is given by $\log \mathcal{Z}_{\pi_t}^*(\mathbf{x}_t)$. Plugging this into the definition of the log partition function we get,

$$\log \hat{\mathcal{Z}}_{\pi_t,\theta}^{\mathrm{LB}}(\mathbf{x}_t) = \mathbb{E}_{\mathbf{x}_0 \sim q_{t,\theta}(\mathbf{x}_0|\mathbf{x}_t)} \left[ \log \left( \frac{p_t(\mathbf{x}_0|\mathbf{x}_t) R(\mathbf{x}_0)}{q_{t,\theta}(\mathbf{x}_0|\mathbf{x}_t)} \right) \right] \tag{21}$$

$$\le \log \mathbb{E}_{q_{t,\theta}(\mathbf{x}_0|\mathbf{x}_t)} \left[ \frac{p_t(\mathbf{x}_0|\mathbf{x}_t) R(\mathbf{x}_0)}{q_{t,\theta}(\mathbf{x}_0|\mathbf{x}_t)} \right] \tag{22}$$

$$= \log \hat{\mathcal{Z}}_{\pi_t}^{\mathrm{IS}}(\mathbf{x}_t) \tag{23}$$

The lower bound turns into equality at the optimal proposal $q_{t,\theta}(\mathbf{x}_0|\mathbf{x}_t) \propto p_t(\mathbf{x}_0|\mathbf{x}_t) R(\mathbf{x}_0)$.

$\square$

## C.2 ESTIMATING DDPP-KL WITH REINMAX

We first provide an algorithmic description below of training using DDPP-KL. We first highlight how the reverse KL objective can be applied to a more general setting beyond just fine-tuning before turning to the exact setting of the main paper.

---

**Algorithm 2** DDPP-KL

---

**Input**: Differentiable reward $R(\mathbf{x}_0)$, base MDM $p_0^{\mathrm{pre}}(\mathbf{x}_0|\mathbf{x}_t)$, fine-tuning MDM $q_\theta(\mathbf{x}_0|\mathbf{x}_t)$, Num samples $K$

1: **while** Training **do**
2: $\quad$ $t, \mathbf{x}_0 \sim \mathcal{U}[0,1], q(\mathbf{x}_0)$ $\qquad\qquad$ ▷ *Sample time and clean data on-policy from the fine-tuning MDM*
3: $\quad$ $\mathbf{x}_t \sim p_t(\mathbf{x}_t|\mathbf{x}_0)$ $\qquad\qquad\qquad$ ▷ *Construct a partially masked sample given clean data*
4: $\quad$ $\{\hat{\mathbf{x}}_0^i\}_{i=0}^K \sim q_{t,\theta}(\cdot|\mathbf{x}_t)$ $\qquad\qquad\qquad$ ▷ *Reparametrized Sampling of clean data*
5: $\quad$ $\mathcal{L}^{\mathrm{KL}} = \frac{1}{K} \sum_{i=1}^K \left( \log q_{t,\theta}(\hat{\mathbf{x}}_0^i|\mathbf{x}_t) - \log p_0^{\mathrm{pre}}(\hat{\mathbf{x}}_0^i|\mathbf{x}_t) - \log R(\hat{\mathbf{x}}_0^i) \right)$
6: $\quad$ $\nabla_\theta \mathcal{L}^{\mathrm{KL}} := \nabla_\theta^{\mathrm{Reinmax}} \left( \mathcal{L}^{\mathrm{KL}} \right)$ $\qquad\qquad$ ▷ *Use the Reinmax discrete gradient estimator*
7: $\quad$ $\theta \leftarrow \mathrm{Update}(\theta, \nabla_\theta \mathcal{L}^{\mathrm{KL}})$
8: **Return** $q_\theta$

---

**Non-finetuning Case**. In this appendix, we study the REINMAX gradient estimator for the general problem of sampling from the following distribution:

$$\pi_0(\mathbf{x}_0) \propto \frac{R(\mathbf{x}_0)}{\mathcal{Z}}. \tag{24}$$

**Gradient of $\mathcal{L}^{\mathbf{KL}}$.** We can decompose the gradient into the following terms due to the linearity of expectations:

$$\mathcal{L}_t^{\mathrm{KL}} = \mathbb{E}_{t,\mathbf{x}_0,\mathbf{x}_t} \left[ \log q_{t,\theta}(\mathbf{x}_0|\mathbf{x}_t) \right] - \mathbb{E}_{t,\mathbf{x}_0,\mathbf{x}_t} \left[ \log \pi_t(\mathbf{x}_0|\mathbf{x}_t) \right]$$
$$= \mathcal{L}_t^1 + \mathcal{L}_t^2. \tag{25}$$

We again highlight the fact that the expectation is taken using the following distributions $t, \mathbf{x}_0, \mathbf{x}_t \sim \mathcal{U}[0,1], q(\mathbf{x}_0), p_t(\mathbf{x}_t|\mathbf{x}_0)$. As a result, $\mathbf{x}_0$ is drawn on-policy and is a stochastic variable that needs gradient estimation since $q_\theta$ is the parameterized distribution. Furthermore, all terms that use *this sample* $\mathbf{x}_0$ inside the expectation are affected by this gradient computation.

Taking the gradient of each term respectively. The gradient of of $\mathcal{L}_t^1$ is:

$$\nabla_\theta \mathcal{L}_t^1 = \nabla_\theta \left( \mathbb{E}_{t,\mathbf{x}_0,\mathbf{x}_t} \left[ \log q_{t,\theta}(\mathbf{x}_0|\mathbf{x}_t) \right] \right)$$
$$\approx \mathbb{E}_{t,\mathbf{x}_0 \sim q_\theta(\mathbf{x}_0),\mathbf{x}_t \sim p_t(\mathbf{x}_t|\mathbf{x}_0)} \left[ \nabla^{\text{Rein-Max}} \circ \left( \log q_{t,\theta}(\mathbf{x}_0|\mathbf{x}_t) \right) \right]. \tag{26}$$

The gradient of of $\mathcal{L}_t^2$ is:

$$\nabla_\theta \mathcal{L}_t^2 = \nabla_\theta \left( \mathbb{E}_{t,\mathbf{x}_0,\mathbf{x}_t} \left[ \log \pi_t(\mathbf{x}_0|\mathbf{x}_t) \right] \right)$$
$$= \nabla_\theta \left( \mathbb{E}_{t,\mathbf{x}_0,\mathbf{x}_t} \left[ - \log p_t(\mathbf{x}_t|\mathbf{x}_0) - \log \pi_0(\mathbf{x}_0) + \log \pi_t(\mathbf{x}_t) \right] \right)$$
$$= \nabla_\theta \left( \mathbb{E}_{t,\mathbf{x}_0,\mathbf{x}_t} \left[ - \log p_t(\mathbf{x}_t|\mathbf{x}_0) - \log R(\mathbf{x}_0) + \log \left( \sum_{\mathbf{x}_0'} \pi_t(\mathbf{x}_t|\mathbf{x}_0')R(\mathbf{x}_0') \right) \right] \right). \tag{27}$$

To use the Reinmax gradient estimator we must compute $\partial f(z)/\partial z$, where $f$ is the function inside the expectation $\mathbb{E}_z[f(z)]$. We now make use of the following facts:

(F1) **Analytic expression of $\nabla_{x_0} \log p_t(x_t|x_0)$.** For simplicity of presentation, we focus on a single token $x_0^i$ in a sequence but the result remains true for the entire sequence $\mathbf{x}_0$. Recall in the discrete setting of masked diffusion models $p_t = \mathrm{Cat}(x_0; \bar{Q}_t x_t)$, which allows us to write:

$$\nabla_{x_0^i} \log p_t(x_t^i|x_0^i) = \frac{\nabla_{x_0^i} p_t(x_t^i|x_0^i)}{p_t(x_t^i|x_0^i)} \tag{28}$$

$$= \frac{\nabla_{x_0^i} \mathrm{Cat}(x_0^i; \bar{Q}_t x_t^i)}{\mathrm{Cat}(x_0^i; \bar{Q}_t x_t^i)} \tag{29}$$

$$= \frac{\nabla_{x_0^i} (x_0^{i,T} \bar{Q}_t x_t^i)}{x_0^{i,T} \bar{Q}_t x_t^i} \tag{30}$$

$$= \frac{\nabla_{x_0^i} (\alpha_t \langle x_t^i, x_0^i \rangle + (1-\alpha_t)\langle x_t^i, e_m \rangle)}{\alpha_t \langle x_t^i, x_0^i \rangle + (1-\alpha_t)\langle x_t^i, e_m \rangle} \tag{31}$$

$$= \frac{\alpha_t x_t^i}{\alpha_t \langle x_t^i, x_0^i \rangle + (1-\alpha_t)\langle x_t^i, e_m \rangle}. \tag{32}$$

(F2) **Differentiability of the reward $\nabla_{\mathbf{x}_0} R(\mathbf{x}_0)$.** If we assume the reward is differentiable we can exploit the same trick to write:

$$\nabla_{\mathbf{x}_0} \log R(\mathbf{x}_0) = \frac{\nabla_{\mathbf{x}_0} R(\mathbf{x}_0)}{R(\mathbf{x}_0)}. \tag{33}$$

Note that the final term in Eq. 27 does not depend on the realization of the sample $\mathbf{x}_0 \sim q(\mathbf{x}_0|\mathbf{x}_t)$ and thus its gradient in Rein-max is 0. This enables us to write the approximate gradient as:

$$\nabla_\theta \mathcal{L}_t^2 \approx \mathbb{E}_{t,\mathbf{x}_0,\mathbf{x}_t}[\nabla^{\text{Reinmax}} \circ (-\log p_t(\mathbf{x}_t|\mathbf{x}_0) - \log R(\mathbf{x}_0))]$$
$$= \mathbb{E}_{t,\mathbf{x}_0,\mathbf{x}_t} \left[ \left( -\sum_i^N \frac{\alpha_t x_t^i}{\alpha_t \langle x_t^i, x_0^i \rangle + (1-\alpha_t)\langle x_t^i, e_m \rangle} - \frac{\nabla^{\text{Reinmax}}_{\mathbf{x}_0} R(\mathbf{x}_0)}{R(\mathbf{x}_0)} \right) \right]. \tag{34}$$

The first term in the equation has a closed-form expression for the gradient but is still a stochastic gradient since it depends on $\mathbf{x}_0 \sim q_\theta(\mathbf{x}_0)$.

**Finetuning Case**. In the fine-tuning setting we aim to sample from the following Bayesian posterior:

$$\pi_0(\mathbf{x}_0) \propto \frac{p_0^{\text{pre}}(\mathbf{x}_0)R(\mathbf{x}_0)}{\mathcal{Z}} \tag{35}$$

For MDMs the likelihood under the model $p_0^{\text{pre}}(\mathbf{x}_0)$ is intractable to evaluate and leads to a modified objective for gradient estimation with REINMAX in $\mathcal{L}_t^{\text{KL}}$ in Eq. 25:

$$\begin{aligned}
\nabla_\theta \mathcal{L}_t^2 &= \nabla_\theta \left( \mathbb{E}_{t,\mathbf{x}_0,\mathbf{x}_t} \left[ \log \pi_t(\mathbf{x}_0|\mathbf{x}_t) \right] \right) \\
&= \nabla_\theta \left( \mathbb{E}_{t,\mathbf{x}_0,\mathbf{x}_t} \left[ -\log p_t^{\text{pre}}(\mathbf{x}_0|\mathbf{x}_t) - \log R(\mathbf{x}_0) + \log \mathcal{Z}_{\pi_t}(\mathbf{x}_t) \right] \right) \\
&= \nabla_\theta \left( \mathbb{E}_{t,\mathbf{x}_0,\mathbf{x}_t} \left[ -\log p_t^{\text{pre}}(\mathbf{x}_0|\mathbf{x}_t) - \log R(\mathbf{x}_0) + \log \left( \mathbb{E}_{\mathbf{x}_0' \sim p_t^{\text{pre}}(\mathbf{x}_0|\mathbf{x}_t)}[R(\mathbf{x}_0')] \right) \right] \right). \quad (36)
\end{aligned}$$

Note that in the equation above we can evaluate the log partition function using samples drawn from the denoising posterior of the pre-trained model $\mathbf{x}_0' \sim p_t^{\text{pre}}(\mathbf{x}_0|\mathbf{x}_t)$ and *not* the on-policy samples $\mathbf{x}_0 \sim q_\theta(\mathbf{x}_0)$. Thus this term is a constant when we compute the gradient. Thus we have,

$$\nabla_\theta \mathcal{L}_t^2 \approx \nabla^{\text{Reinmax}} \circ \left( \mathbb{E}_{t,\mathbf{x}_0,\mathbf{x}_t} \left[ -\log p_t^{\text{pre}}(\mathbf{x}_0|\mathbf{x}_t) - \log R(\mathbf{x}_0) \right] \right). \tag{37}$$

### C.3 EQUIVALENCE OF SUB-TRAJECTORY OBJECTIVES

In this appendix, we detail how to compute an efficient approximation of the loss function that is inspired by the KL divergence between sub-trajectories as found in the GFlowNet literature but adapted for MDMs.

Consider the trajectory of a sequence: $\tau(\mathbf{x}_{0:1}) := \mathbf{x}_1 \to \cdots \to \mathbf{x}_t \to \mathbf{x}_{t-1} \to \ldots \mathbf{x}_0$. We seek to minimize the joint distribution over the (sub-)trajectories conditioned on a partially masked sample $\mathbf{x}_t$:

$$q_\theta(\mathbf{x}_0, \ldots, \mathbf{x}_{t-1}|\mathbf{x}_t, \mu_\theta(\mathbf{x}_t, t))p_t(\mathbf{x}_t) = \pi_t(\mathbf{x}_0, \ldots, \mathbf{x}_{t-1}|\mathbf{x}_t)p(\mathbf{x}_t). \tag{38}$$

Here $\pi_t(\mathbf{x}_1, \ldots, \mathbf{x}_{t-1}|\mathbf{x}_t, \mathbf{x}_0)$ is defined as,

$$\pi_t(\mathbf{x}_0, \ldots, \mathbf{x}_{t-1}|\mathbf{x}_t, \mu_\theta(\mathbf{x}_t, t)) = \frac{p_t^{\text{pre}}(\mathbf{x}_0, \ldots, \mathbf{x}_{t-1}|\mathbf{x}_t)R(\mathbf{x}_0)}{\mathcal{Z}_{\pi_t}(\mathbf{x}_t)} \tag{39}$$

$$= \frac{\prod_{j=1}^t p_t^{\text{pre}}(\mathbf{x}_{j-1}|\mathbf{x}_j, \hat{\mathbf{x}}_0^{\text{pre}})R(\mathbf{x}_0)}{\mathcal{Z}_{\pi_t}(\mathbf{x}_t)} \tag{40}$$

We minimize the following KL divergence,

$$\mathbb{D}_{\text{KL}}(q_\theta(\mathbf{x}_0, \ldots, \mathbf{x}_{t-1}|\mathbf{x}_t, \hat{\mathbf{x}}_0)p_t(\mathbf{x}_t) || \pi_t(\mathbf{x}_0, \ldots, \mathbf{x}_{t-1}|\mathbf{x}_t)p(\mathbf{x}_t)). \tag{41}$$

Here we used the convention that $\hat{\mathbf{x}}_0 = \mu_\theta(\mathbf{x}_t, t)$ and $\hat{\mathbf{x}}_0^{\text{pre}} = \mu^{\text{pre}}(\mathbf{x}_t, t)$. The KL between path measures along the sub-trajectory shares the same optimum as the following loss objective:

$$\begin{aligned}
\mathcal{L}_\tau &= \mathbb{E}_{t,\mathbf{x}_t} \left[ \mathbb{E}_{\tau(\mathbf{x}_{0:t})} [\| \log q_\theta(\mathbf{x}_0, \ldots, \mathbf{x}_{t-1}|\mathbf{x}_t, \hat{\mathbf{x}}_0)) - \log p_t^{\text{pre}}(\mathbf{x}_0, \ldots, \mathbf{x}_{t-1}|\mathbf{x}_t) + \kappa \|_2^2 ] \right] \\
&= \mathbb{E}_{t,\mathbf{x}_t} \left[ \mathbb{E}_{\tau(\mathbf{x}_{0:t})} \left[ \left\| \sum_{j=1}^t \log q_\theta(\mathbf{x}_{j-1}|\mathbf{x}_j, \hat{\mathbf{x}}_0) - \log p_t^{\text{pre}}(\mathbf{x}_{j-1}, |\mathbf{x}_j, \hat{\mathbf{x}}_0^{\text{pre}}) + \kappa \right\|_2^2 \right] \right] \quad (42) \\
&= \mathbb{E}_{t,\mathbf{x}_t,\tau(\mathbf{x}_{0:t})} \left[ \left\| \sum_{s=1}^t \log q_\theta(\mathbf{x}_{s-\gamma}|\mathbf{x}_s, \hat{\mathbf{x}}_0) - \log p_t^{\text{pre}}(\mathbf{x}_{s-\gamma}, |\mathbf{x}_s, \hat{\mathbf{x}}_0^{\text{pre}}) + \kappa \right\|_2^2 \right] \\
&= \mathbb{E}_{t,\mathbf{x}_t,\tau(\mathbf{x}_{0:t})} \left[ \left\| t\mathbb{E}_{s,\mathbf{x}_s,\mathbf{x}_{s-\gamma}} \left[ \log q_\theta(\mathbf{x}_{s-\gamma}|\mathbf{x}_s, \hat{\mathbf{x}}_0) - \log p_t^{\text{pre}}(\mathbf{x}_{s-\gamma}, |\mathbf{x}_s, \hat{\mathbf{x}}_0^{\text{pre}}) + \kappa \right] \right\|_2^2 \right]. \quad (43)
\end{aligned}$$

In the last equation we define the constant $\kappa = (-\log R(\mathbf{x}_0) + \log \mathcal{Z}_{\pi_t}(\mathbf{x}_t))/t$ and use the fact that our notation convention uses $p(X = x) = p(x)$ for discrete random variables. Now we make the observation that for any $s < t$ we have effectively picked an endpoint over the trajectory. More precisely, $s \sim \mathcal{U}[0, t]$, which also allows us to sample $\mathbf{x}_s \sim p_t(\mathbf{x}_s|\mathbf{x}_0)$, in an analogous manner to how $\mathbf{x}_t$ is constructed.

# D  ADDITIONAL EXPERIMENTAL DETAILS

All experiments were performed on a shared heterogenous high-performance computing cluster. This cluster is primarily composed of GPU nodes with RTX8000, V100, A100, L40S, and H100 NVIDIA GPUs. We briefly note a trick used across a number of our experiments for DDPP-LB described as warming up $\log Z_t(x_t)$. We found that early iterations of training DDPP-LB could be somewhat unstable as the parameterized normalizing constant was not calibrated to a proper range given the pre-trained model and reward function. As such, we found that warming up $\log Z_t(x_t)$ for some number of steps at the beginning of training by only allowing gradient flow through the $\log Z_t(x_t)$ term helped stabilize training and improve overall performance. For the runs on which warming up $\log Z_t(x_t)$ was utilized, we resume normal training (i.e., allowing gradient flow through the fine-tuned denoiser *and* $\log Z_t(x_t)$) after the warmup period has concluded. For all experiments with DDPP-LB we used another separate, small DiT to parameterize the $\log Z_t(x_t)$ prediction.

## D.1  SYNTHETIC EXPERIMENTS

Two synthetic tasks were performed: (1) sampling from a posterior over a 2 dimensional grid, and (2) fine-tuning on binarized MNIST. In both cases a 90 million parameter MDM model was trained on samples from the prior distribution, with the same DiT architecture as in Sahoo et al. (2024).

### D.1.1  GRID EXPERIMENT

The space consists of discrete tokens $\mathbf{x}_0 \in \{0, \dots, 127\}^2$. A prior density $p_0^{\text{pre}}$ is defined over this space which assigns a uniform probability for tokens falling inside one of the 16 evenly spaced squares, and a near-zero probability outside this. This prior distribution is depicted in Figure 6(a). Pre-training was done using the Adam optimizer, with $\beta_1, \beta_2 = \{0.9, 0.999\}$, and a learning rate of $3e{-}4$.

The reward function $R(\mathbf{x}_0) = 0$ for $x_0^1 < 64$, and $R(\mathbf{x}_0) = 1$ for $x_0^1 \geq 64$. This results in a fine-tuning target $\propto R(\mathbf{x}_0)p^{\text{pre}}(\mathbf{x}_0)$ which selects out only the squares in the lower half of the grid. This product distribution is visualized in Figure 6(b).

For fine-tuning we train the model using our loss-functions with the Adam optimizer, using a learning rate of $4e-3$, $\beta_1, \beta_2 = \{0.9, 0.999\}$, and a weight decay of $0$ across all methods. DDPP-IS used 16 samples to estimate the partition function. Training is done using a replay buffer populated with points $\mathbf{x}_0$ sampled on policy from the model, as well as off-policy points from the prior distribution, added to the buffer every 100 training steps. A batch of 64 is used.

### D.1.2  MNIST

This task consisted of generating binarized MNIST digits $\mathbf{x}_0 \in \{0, 1\}^{28 \times 28}$. The prior $p^{\text{pre}}(\mathbf{x}_0)$ in this case is the MNIST data distribution. For pre-training, the Adam optimizer is used with a learning rate of $4e-3$, $\beta_1, \beta_2 = \{0.9, 0.999\}$ and a weight decay of $0$.

This MDM is fine-tuned to produce even digits. More precisely, the reward function is $R(\mathbf{x}_0) = p(\text{Even} \mid \mathbf{x}_0)^\beta = \left( \sum_{i=0,2,4,6,8} p(y = i \mid \mathbf{x}_0) \right)^\beta$, with $p(y = i \mid \mathbf{x}_0)$ being obtained from a pretrained MNIST classifier (LeNet 5 in this case). The inverse-temperature $\beta$ is set to 5 for all experiments.

For fine-tuning with our methods, we use Adam with a learning rate of $1e-5$ and $\beta_1, \beta_2 = \{0.9, 0.999\}$. Training is done with a batch-size of 64. Samples are drawn from a replay-buffer populated with only on-policy samples. Method specific hyperparameters include:

- DDPP-IS: the importance sampling estimate is done with 16 samples
- DDPP-LB: a learning rate of $1e-3$ is used for network layers estimating $\log \mathcal{Z}_{\pi_t}$
- DDPP-KL: The KL objective per $\mathbf{x}_t$ is computed using 8 samples

RTB is trained with a learning rate of $5e-5$, with weight decay $0.01$, on trajectories of length 32 with a batch size of $8$. For training, $30\%$ of the steps are detached. The smaller batch-size is chosen to fit the training on 80GB of GPU memory.

SVDD uses 10 particles in each inference step.

For all methods (including baselines), inference is done with 128 steps.

Additional information on computation of metrics is included in D.3.1.

### D.1.3 MNIST SAMPLES

Samples from our methods, as well as the pretrained model, are shown in Figure 4.

### D.2 PROTEIN SEQUENCES

Protein design involves the creation of novel protein sequences that adopt specific structures and perform desired functions. This is a critical field in synthetic biology and biotechnology, as it enables the rational engineering of proteins with enhanced stability, novel functionalities, or improved therapeutic properties. Advances in machine learning-based models, such as protein language models (pLMs), have enabled rapid exploration of protein sequence space, making de novo protein design more feasible and versatile. However, current pLMs struggle in generating realistic sequences which satisfy certain criteria, and we study using DDPP to finetune DPLM to generate high-scoring proteins given a reward function.

### D.2.1 IN-SILICO TASKS

In task 1, we fine-tune the DPLM model to generate designable protein sequences that optimize for several critical features, including high predicted template modeling (pTM) and predicted local distance difference test (pLDDT) scores from ESMFold, reduced exposed hydrophobic residues, high sequence entropy, and an increased proportion of $\beta$-sheet content (Hie et al., 2022). These optimizations are captured in the reward function $R$, given by:

$$\log R = w_{\text{pTM}} \cdot \text{pTM} + w_{\text{pLDDT}} \cdot \text{pLDDT} + w_{\text{Sheet}} \cdot \text{Sheet\%}$$
$$+ w_{\text{Entropy}} \cdot H(\mathbf{s}) - w_{\text{Hpho}} \cdot \text{Exposed\_Hpho\%}$$

Where the terms represent:

- pTM and pLDDT: Structural confidence scores from ESMFold, measuring global and local accuracy, respectively.
- Sheet%: The proportion of residues predicted to form $\beta$-sheets, determined by DSSP (Kabsch and Sander, 1983).
- $H(\mathbf{s})$: Sequence entropy, defined as:

$$H(\mathbf{s}) = -\sum_{i=1}^{L} \sum_{a} p_i(a) \log p_i(a),$$

where $L$ is the length of the sequence and $p_i(a)$ is the probability of amino acid $a$ at position $i$.
- Exposed_Hpho%: Percentage of hydrophobic residues exposed on the surface, calculated based on solvent-accessible surface area.

The weights for these features are set as follows:

$$w_{\text{pTM}} = 1, \quad w_{\text{pLDDT}} = 1, \quad w_{\text{Sheet}} = 4.5, \quad w_{\text{Entropy}} = 0.8, \quad w_{\text{Hpho}} = 0.25.$$

As the scale of the various reward terms are non-uniform we selected the reward weights to weight all rewards similarly besides the sheet percent reward which is weighted higher. For the $\beta$-sheet task we found that both RTB and DDPP faced issues with mode collapse. After investigating the protein structures generated by base DPLM we found that the base model is only capable of generating a small number of motifs (in particular, over 2k samples from the base model we found only two motifs with $\log R(x_0) \geq 3.5$), implying that the targeted product distribution indeed collapses around these structural motifs as we observe in the case of RTB and DDPP. As such, we conclude that DDPP (and RTB) achieve the goal of fine-tuning as they sample from the product distribution and reproduces samples with $\beta$-sheets at a much higher proportion than the base model.

In task 2, we focus on generating shorter sequences of known proteins that preserve essential structural characteristics, using the TM-align score as the reward function (Devkota et al., 2024). This task allows the exploration of mutational effects. Ribonuclease proteins (PDB IDs: 9RAT-A, 11BA-A) are selected for this task due to their well-characterized structure, function, and folding mechanisms.

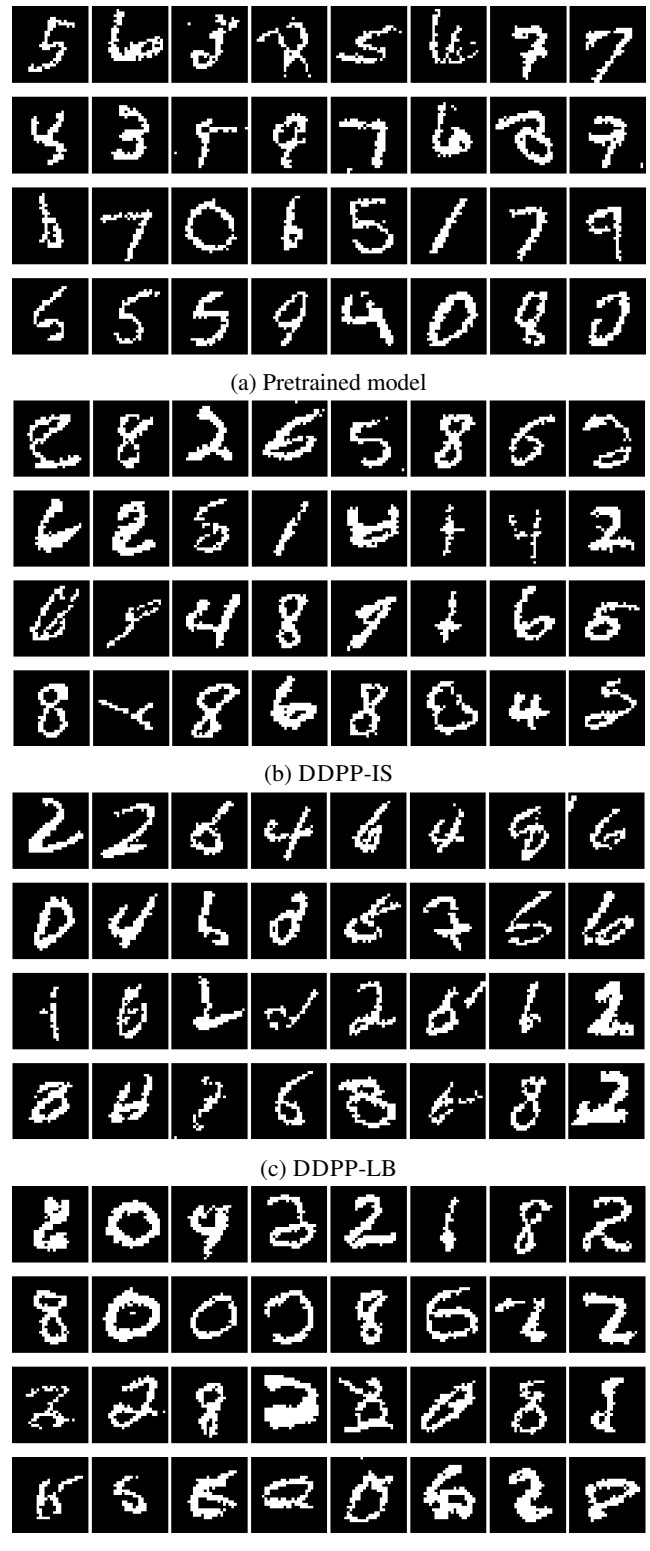

(a) Pretrained model

(b) DDPP-IS

(c) DDPP-LB

(d) DDPP-KL

Figure 4: Uncurated samples from the pretrained model, and after fine-tuning with our methods: DDPP-IS, DDPP-LB, DDPP-KL

The reward function $R$ is defined as:

$$R = w_{\text{tm\_score}} \cdot \text{TM-align}(\mathbf{s}, \mathbf{t}).$$

Where:

- $w_{\text{tm\_score}}$: The weight of the TM-Score reward, set to 2.
- $\mathbf{s}$: Predicted structure from ESMFold of the generated sequence.
- $\mathbf{t}$: Target protein structure.
- TM-align: A measure of structural similarity between $\mathbf{s}$ and $\mathbf{t}$, defined as:

$$\text{TM-align} = \max\left( \frac{1}{L_t} \sum_{i=1}^{L_{\text{ali}}} \frac{1}{1 + \left(\frac{d_i}{d_0}\right)^2} \right).$$

  where $L_t$ is the length of the target protein, $L_{\text{ali}}$ is the length of the aligned region, $d_i$ is the distance between the $i$-th pair of aligned residues, and $d_0$ is the distance scale based on $L_t$ (Zhang and Skolnick, 2005).

While not used in the reward function for either experimental setting, we also measure the KL divergence, reported as KL-SS in Table 3 between the secondary structure distribution given by DSSP for both the target and miniaturized protein.

Note that in these experiments, the number of recycles in ESMFold is set to 0 to reduce computational overhead. For both tasks we generate amino acid sequences of length 90. Evaluation is performed by sampling 200 proteins for each method across three seeds and reporting the mean and standard deviation of each metric accordingly. All methods ran 500 inference steps during evaluation. All protein experiments used a 150 million parameter DPLM base model[1] to begin fine-tuning from. All models used a log-linear noise schedule with $\sigma_{min} = 1\mathrm{e}{-4}$ and $\sigma_{max} = 20$ and used a linear learning rate warmup period of 2500 training steps.

DDPP was trained with no warmup period for $\log Z_t(x_t)$, a learning rate of $1\mathrm{e}{-5}$, a batch size of 16, a replay buffer of max length 10,000, and inserting new batches to the buffer sampled on-policy from the current model every 250 training steps. RTB was trained similarly, but with a smaller batch size to account for its greater memory requirements. RTB matches the setting of DDPP but with a batch size of 8 while doing 90 inference steps during training (a new batch of trajectories is simulated on-policy every training step). To allow RTB to fit in memory we detached 65% of trajectory timesteps when computing a backward pass on the RTB objective. SVDD was run on the base DPLM model with $n = 10$ particles. To control the concentration of our designated target distributions, we set the reward temperature $\beta = 0.125$ for the $\beta$-sheet task and $\beta = 0.001$ for the protein miniaturization task.

We report an extended version of Table 3 where we include results for both ribonuclease targets in Table 5. We observe that DDPP consistently achieves the highest TM-Score across the two templates while maintaining high structural quality with an average pLDDT of around 0.8.

### D.2.2 EXPERIMENTAL VALIDATION

Genes encoding for de novo protein sequences were obtained from Integrated DNA Technologies (IDT) and cloned into pET-24a(+) (Novagen) expression vectors with a C-terminal 6xHis tag using Gibson Assembly (New England Biolabs, NEB). Assembled plasmids were verified via Sanger sequencing, then transformed into chemically competent *Escherichia coli* BL21(DE3) cells (NEB). Starter cultures (3 mL Luria Bertani media, 50 μg/mL kanamycin) were inoculated from freshly prepared agar plates and grown at 37°C and shaken at 225 RPM overnight. Starter cultures were then diluted 1:100 into 50 mL LB medium supplemented with antibiotic. Cultures were then grown at 37°C and 225 RPM until an optical density (OD600) of 0.5-0.7 was reached. Protein expression was then induced with 1 mM isopropyl $\beta$-D-thiogalactopyranoside (IPTG) for 4 hours at 37°C. Cells were then collected by centrifugation (4,500xg) at 4°C and resuspended in lysis buffer (Tris-buffered saline (TBS), 25 mM imidazole). Cell suspensions were then lysed via sonication (10s pulses, 40% amplitude). The corresponding lysate was centrifuged at 12,000xg for 30 minutes, and the

---

[1] https://huggingface.co/airkingbd/dplm_150m

Table 5: Miniaturizing ribonuclease proteins 9RAT-A and 11BA-A (124 AAs) to 90 AAs while preserving structural fidelity (high TM-Score) and quality (high pLDDT and PTM).

| Template | | SS-KL $\downarrow$ | $\log R(\mathbf{x}_0) \uparrow$ | TM-Score $\uparrow$ | pLDDT $\uparrow$ | pTM $\uparrow$ |
|---|---|---|---|---|---|---|
| | Base Model | $2.944 \pm 2.936$ | $0.502 \pm 0.128$ | $0.251 \pm 0.064$ | $0.724 \pm 0.144$ | $0.584 \pm 0.226$ |
| | Best-of-10 | $\mathbf{0.640 \pm 1.872}$ | $0.725 \pm 0.098$ | $0.363 \pm 0.049$ | $0.789 \pm 0.018$ | $0.754 \pm 0.086$ |
| 9RAT-A | DDPP | $1.086 \pm 2.242$ | $\mathbf{0.735 \pm 0.122}$ | $\mathbf{0.368 \pm 0.061}$ | $0.793 \pm 0.044$ | $\mathbf{0.768 \pm 0.066}$ |
| | RTB | $1.808 \pm 2.597$ | $0.597 \pm 0.109$ | $0.299 \pm 0.055$ | $\mathbf{0.796 \pm 0.054}$ | $0.750 \pm 0.084$ |
| | SVDD | $3.465 \pm 2.835$ | $0.699 \pm 0.079$ | $0.350 \pm 0.039$ | $0.499 \pm 0.137$ | $0.383 \pm 0.178$ |
| | Base Model | $3.136 \pm 3.150$ | $0.478 \pm 0.101$ | $0.239 \pm 0.051$ | $0.724 \pm 0.144$ | $0.584 \pm 0.226$ |
| | Best-of-10 | $2.602 \pm 3.309$ | $0.654 \pm 0.089$ | $0.327 \pm 0.045$ | $0.782 \pm 0.027$ | $0.720 \pm 0.109$ |
| 11BA-A | DDPP | $\mathbf{0.194 \pm 1.009}$ | $\mathbf{0.709 \pm 0.048}$ | $\mathbf{0.354 \pm 0.024}$ | $0.743 \pm 0.036$ | $0.727 \pm 0.054$ |
| | RTB | $2.579 \pm 2.799$ | $0.564 \pm 0.111$ | $0.282 \pm 0.056$ | $\mathbf{0.797 \pm 0.058}$ | $\mathbf{0.744 \pm 0.101}$ |
| | SVDD | $3.240 \pm 2.992$ | $0.647 \pm 0.079$ | $0.324 \pm 0.040$ | $0.486 \pm 0.124$ | $0.354 \pm 0.162$ |

supernatant was loaded into a HisPur Ni-NTA His-spin column (ThermoScientific) and purified as recommended. Expression of purified proteins in both the soluble and insoluble fraction, as well as his-tag purification fractions, was assessed using SDS-polyacrylamide gel electrophoresis.

### D.3 DISCRETE IMAGE MODELLING

To setup the finetuning task we first pre-train large masked diffusion models on the original dataset. This uses a standard masked diffusion loss as explored in previous work (Shi et al., 2024; Sahoo et al., 2024).

**CelebA Pretraining**. We train a 241 million parameter model based on the variational diffusion model (VDM) architecture (Kingma et al., 2023) and the setup of Shi et al. (2024). We adapted the U-Net plus self-attention architectures from Kingma et al. (2023) as used in CIFAR-10 in their experiments, with a few notable additions. We replace the Fourier feature inputs with an input embedding layer which embeds 257 (256 pixel values + <MASK>) tokens into the embedding dimension. We double the number of residual blocks from 32 to 64 per encoder / decoder, and double the embedding dimension from 128 to 256. We use an Adam optimizer with learning rate $1e-3$, $\beta_1$=0.9 and $\beta_2$=0.999. We train our model for 450k steps with batch size 128 on a cluster of 16 NVIDIA L40S GPUs. We resize all CelebA images to 64x64 with bilinear interpolation. Samples from this model can be seen in Figure 5.

Separately, we train a 7M parameter classifier to classify hair color on CelebA. We use this as our energy function with a temperature setting of 0.1 for all finetuning experiments.

**CelebA Finetuning**. With the problem setup, we next finetune our pretrained model to sample images with blond hair. We train each model for up to 12 A100 hours. We use an early stopping criteria based on a validation set using an approximate bits-per-dimension calculation using the ELBO. We find that the original needs at least $1\,000$ inference steps for good performance therefore we evaluate all models in this setting. For our model we use $1\,000$ warmup steps for $\log Z$, a learning rate of $1e-4$, we resample two batches every 500 gradient steps of the model and add them to the replay buffer.

In contrast to our model, RTB requires a full trajectory for each gradient step. For CelebA, this means a rollout of $1\,000$ inference steps taking approximately 2 minutes for a batch size of 2 on an A100 with this model. Because of memory constraints we detach 99% of inference steps and use a batch size of 2 to fit in 80GB of GPU memory with a global batch size of 8 trajectories per gradient step.

### D.3.1 METRICS

The metrics used to evaluate image fine-tuning include mean log reward, feature-likelihood divergence (FLD), and bits per dimension (BPD).

**FLD**. For FLD, we draw $K$ samples from the model, and $K$ samples from the test set restricted to the target class. The FLD is computed using the DINOV2 feature space (from the ViT-B14 model) between these two sets of samples (Oquab et al., 2024). For MNIST, $K = 5k$.

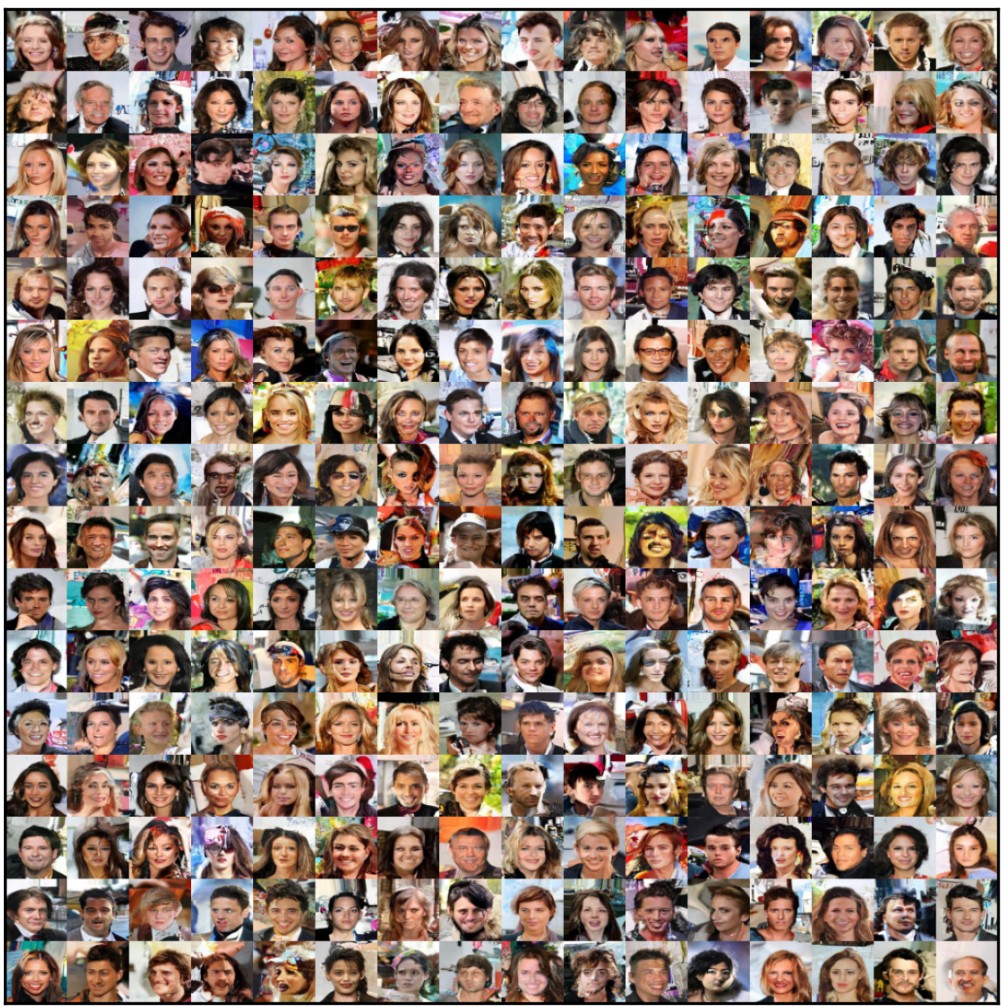

Figure 5: Uncurated pre-trained CelebA model samples using a discrete generative model.

**BPD**. An upper bound on the log-likelihood is computed using the MDM ELBO loss (on the fine-tuned model), and this is normalized (by the number of pixels and $\log 2$) to obtain the reported BPD metric. For baselines other than discrete guidance, this metric is computed on the test set restricted to the target-class. For discrete guidance, BPD is computed by evaluating the MDM ELBO of the base model on samples generated using guidance (due to lacking an analogous ELBO for guidance-based sampling).

## D.4 TEXT EXPERIMENTS

All text experiments begin by starting from the pretrained MDLM[2] consisting of 170 million parameters. We then do supervised fine-tuning to produce a model capable of producing output of the desired format before proceeding with online fine-tuning. For all text experiments we train using the Adam optimizer with $\beta_1, \beta_2 = \{0.9, 0.999\}$ and weight decay of 0. For both tasks SVDD was run with $n = 10$ particles. Evaluation was done by training each method for one day across three seeds and generating 1000 samples from the best checkpoint according to the mean reward generated during training. We report both the mean reward of the 1000 samples across three seeds and their standard deviations, as well as the generative perplexity according to a GPT2-Large model with 812 million parameters [3].

### D.4.1 TINYSTORIES

To obtain a base model we performed supervised fine-tuning from the base MDLM model on the Tinystories dataset. As all methods were prompted with the text "Once upon a time" during training, we restricted the SFT dataset to only datapoints whose stories started with the text "Once upon a time,", resulting in a corpus of 977,921 examples. SFT was done using Adam, $\beta_1, \beta_2 = \{0.9, 0.999\}$, and a learning rate of $4e{-}3$ over 60,000 training steps using 4 NVIDIA A100 GPUs. All models were trained for up to 24 GPU hours on NVIDIA L40S GPUs. Fine-tuning checkpoints were selected based upon the iteration with best average reward when sampling a new training batch from the model. All methods employ a learning rate schedule with a linear warmup for the first 2,500 training steps and keep the noise schedule provided by the pre-trained MDLM model – a log-linear schedule with $\sigma_{min} = 1e{-}4$ and $\sigma_{max} = 20$. All evaluations were performed by taking 1000 samples for each method across three seeds. We provide a set of curated samples in Table 6.

The reward function for this task was selected to be a pre-trained classifier[4] (Kluge Corrêa, 2024). The classifier is a RoBERTa model with 125 million parameters which was fine-tuned on a curated subset of various toxicity/harmlessness datasets. The reward $R(x_0)$ is then set to the likelihood of a sequence being toxic under the pre-trained classifier so that $R(x_0) = p(a = 1|x_0)$ where $p(a = 1|x_0)$ denotes the likelihood of the sequence $x_0$ possessing a toxic sentiment. We select this task as it allows a demonstration of how our method can recover rare behavior under the pre-trained model while still maintaining sample quality. We consider as our target distribution the tempered reward distribution $\pi_0(x_0) \propto p_0^{\text{pre}}(x_0)R(x_0)^{1/\beta}$ with $\beta = 0.25$.

For DDPP-LB we used 1,500 warmup steps for $\log \mathcal{Z}_{\pi_t}(x_t)$, a learning rate of $1e{-}4$ and a batch size of 16. We employ a replay buffer with a max length of 10,000 and sample training batches uniformly from the buffer. The buffer is filled every 50 training steps with a batch sampled on-policy from the current fine-tuned model, while every 250 steps a batch from the SFT training dataset is added to the buffer. We use EMA with a decay rate of $\epsilon = 0.9999$, a learning rate of $1e{-}4$, and train without LoRA. DDPP-LB was trained using 64 inference steps for simulation. DDPP-IS employed the same hyperparameter settings as DDPP-LB except that it dispelled with learning $\log \mathcal{Z}_{\pi_t}(x_t)$ and instead estimated it with $K = 16$ Monte Carlo samples from the one-step pre-trained posterior $p_t^{\text{pre}}(x_0|x_t)$.

As RTB cannot fit all timesteps of a trajectory into memory during the backwards pass, we detached 55% of timesteps where each trajectory consisted of 32 timesteps. RTB was trained with LoRA enabled, a LoRA rank of 16, and a learning rate of $5e{-}5$. Due to memory constraints the batch size was set to 4. SVDD was run by using $n = 10$ particles per inference timestep. best of $N$ (with $N = 10$) sampling was performed by taking 10 samples from the SFT model and selecting the sample with highest likelihood under the reward model.

---

[2] https://huggingface.co/kuleshov-group/mdlm-owt
[3] https://huggingface.co/openai-community/gpt2-large
[4] https://huggingface.co/nicholasKluge/ToxicityModel

Table 6: Curated samples from DDPP-LB on Tinystories.

| Generated Text |
| --- |
| Once upon a time, there was a little girl named Lily. One day, Lily went to the park with her mom. She loved to run fast and laugh. Lily saw a funny butterfly and ran it too. She fell fast and hurt a tree. Lily's knee hurt and she cried. But her mommy kissed her cheeks and said, "Be careful when you run, Lily." |
| Once upon a time, there was an elderly wolf. He lived in a big den against the woods. One day, the wolf felt very tired and wanted a place where there was a big tree to eat on. So, he went to sleep all day. But, while he was waking up, he saw a big, icy creature. The quickly jumped up, but his legs were too weak. The creature took the elderly wolf away. And that is how winter ended. |
| Once upon a time, there was a little bird. His wings were weak and he fell down. The bird wanted his wing to restore him. So, he flapped his wings with his weak heart. |
| Once upon a time, there was a little boy named Timmy. He loved going to the woods with his family. One day, Timmy's friend Johnny came to the woods to play. Johnny was excited to go outside and play.

They found a big tree with words said "I reverse," and Timmy would play on its branches. He said, "reverse!" and pushed the tree. Then, they ran and laughed.

But then, they heard a loud noise coming from the bushes and scared them. It was a big, mean bear! Timmy tried to reverse and run away, but he wasn't fast enough. The bear chased him and caught him up with its sharp hands.

Timmy was very scared and never went back to the woods again. |
| Once upon a time, on a calm blue sea, there was a small boat. The boat had sailors. They lived happily in the day water. One day, the water was very hot. So, they all decided to soak up and have a picnic.

But, by the time, the sailors started to play a game. They swam around and counted, ", two, three soon!" and all the sailors kept playing. They water flowers and trees, and everyone laughed.

But then, a clumsy heavy sailor hurt his head on a rock. "Ouch!" he cried. His friends helped him up and said, "Be careful next time!" The sailor felt better and they all laughed. They knew they could play again and have fun on a calm day soon. |

### D.4.2 AMAZON REVIEWS

Table 7: Curated samples from DDPP-LB on Amazon review generation task.

| Generated Text |
| --- |
| Cheap. Poor fit. The pants return immediately and the fabric was like a burlap sack bag-Wanted it for a gift-It's crap. Broke in one day. Customer service never responded.
Very cheap! |
| Everything was horrible. |
| It's the worst one I've ever bought.. you have to keep it in your bag for my daughter and they, seriously feel embarrassed if you ever got it it falls out and it ripped.. returning this. That said hated this bag till I see it!!!!!! (Cnaven at the neckline and it is super too short. Color is off white. Maybe I have to fix if I want ironing
What a waste of valu money |
| Such poor material!!! It was like a plastic. Way too small so I returned.It was smaller than the size listed |
| I don't feel this product has any quality. My sunglasses was delivered broken. |
| Cheap piece of garbage. Got it for my niece for Halloween and it broke inó one time |

We again begin by first performing supervised fine-tuning from the base MDLM model, but this time on the fashion split of the Amazon Reviews dataset (Hou et al., 2024) restricted to reviews consisting of at most 512 tokens, resulting in an SFT dataset of size. We perform SFT using Adam with $\beta_1, \beta_2 = \{0.9, 0.999\}$ and a learning rate of $4e-3$ and EMA with decay parameter $0.99$. The SFT model was trained for 85,000 training steps on 4 NVIDIA A100 GPUs. As for the tinystories task fine-tuning checkpoints were selected based upon the iteration with best average reward when sampling a new training batch from the model. All methods employ a learning rate schedule with a linear warmup for the first 2,500 training steps and keep the noise schedule provided by the pre-trained MDLM model – a log-linear schedule with $\sigma_{min} = 1e-4$ and $\sigma_{max} = 20$. All evaluations were performed by taking 1000 samples for each method across three seeds.

The reward function for this task was a BERT model consisting of 167 million parameters fine-tuned on Amazon customer reviews[5] to predict a review's star-rating. We then set the reward $R(x_0) = p(a = 1|x_0)$, the likelihood under the pre-trained classifier that the generated sample is a one-star review. We consider as our target distribution the tempered reward distribution $\pi_0(x_0) \propto p_0^{\text{pre}}(x_0)R(x_0)^{1/\beta}$ with $\beta = 0.5$.

For DDPP-LB we used 1,500 warmup steps for $\log \mathcal{Z}_{\pi_t}(x_t)$, a learning rate of $1e-4$ and a batch size of 16. We employ a replay buffer with a max length of 10,000 and sample training batches uniformly from the buffer. The buffer is filled every 5 training steps with a batch sampled on-policy from the current fine-tuned model, while every 250 steps a batch from the SFT training dataset is added to the buffer. We use EMA with a decay rate of $\epsilon = 0.9999$, a learning rate of $1e-4$, and train without LoRA. DDPP-LB was trained using 64 inference steps for simulation. DDPP-IS employed the same hyperparameter settings as DDPP-LB besides not learning $\log \mathcal{Z}_{\pi_t}(x_t)$ and instead estimating it with $K = 16$ Monte Carlo samples from the one-step pre-trained posterior $p_t^{\text{pre}}(x_0|x_t)$.

As RTB cannot fit all timesteps of a trajectory into memory during the backwards pass, we detached 78.5% of timesteps where each trajectory consisted of 64 timesteps. RTB was trained with LoRA enabled, a LoRA rank of 16, and a learning rate of $5e-5$. Due to memory constraints the batch size was set to 4. SVDD was run by using $n = 10$ particles per inference timestep. best of $N$ (with $N = 10$) sampling was performed by taking 10 samples from the SFT model and selecting the sample with highest likelihood under the reward model.

We provide a set of curated samples for the Amazon task in Table 7.

---

[5]https://huggingface.co/LiYuan/amazon-review-sentiment-analysis

# E ADDITIONAL EXPERIMENTAL RESULTS

## E.1 AUTOREGRESSIVE BASELINE

Table 8: Results for Tinystories with Twisted SMC autoregressive baseline. Because DDPP and Twisted SMC use different architectures and number of parameters, we report the reward and generative perplexity before and after fine-tuning and bold the method which provides the best percent change from the base model.

| | $\log R(x_0)$ pre | $\log R(x_0)$ post | % change in $\log R(x_0)$ ↑ | Gen PPL pre | Gen PPL post | % change in Gen PPL ↓ |
|---|---|---|---|---|---|---|
| Twisted SMC | $40.52 \pm 0.10$ | $94.56 \pm 0.85$ | $133.4 \pm 1.60$ | $8.52 \pm 0.05$ | $10.25 \pm 0.51$ | $\mathbf{20.31 \pm 5.96}$ |
| DDPP-LB | $54.94 \pm 0.76$ | $205.76 \pm 3.88$ | $\mathbf{278.0 \pm 15.2}$ | $16.66 \pm 0.20$ | $19.6 \pm 0.69$ | $\mathbf{18.38 \pm 4.05}$ |

In order to compare the performance of DDPP against autoregressive methods, we evaluated Twisted SMC (Zhao et al., 2024b) on the Tinystories task and compared its performance to DDPP. Unfortunately, the base model used by Twisted SMC was of a different architecture and model size than we used. In particular, as the base autoregressive model, we used the GPT-Neo architectured model trained in the original Tinystories paper (Eldan and Li, 2023) which uses 68 million parameters while our base model used a diffusion transformer (Peebles and Xie, 2023) with 170 million parameters. To ensure the fairest possible comparison, despite the difference in model parameters of the pre-trained models, we compare Twisted SMC and DDPP fine-tuning in terms of the percent change in the average reward and generative perplexity of the finetuned samples. Results are presented in Table 8.

We observe that DDPP improves reward more than the autoregressive baseline while incurring a comparable but minor performance drop in generative perplexity to Twisted SMC. Our results here contextualize that DDPP can better negotiate the tradeoff between optimizing reward and sample quality than twisted SMC on autoregressive models. Finally, we note that Twisted SMC cannot be easily performed for MDM's and as such DDPP remains a compelling choice for fine-tuning.

## E.2 COMPARING OVERALL COMPUTATION TIME OF DDPP VS INFERENCE BASED METHODS

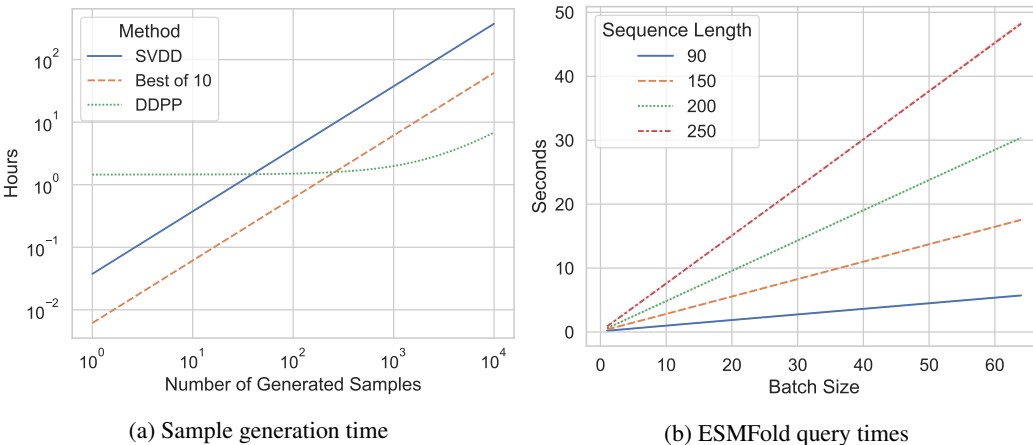

(a) Sample generation time          (b) ESMFold query times

Figure 6: (a) We plot the number of hours required to generate a particular number of samples for each method (including DDPP training time) on the 90 sequence length designable $\beta$-sheet protein task. We see that although inference time methods may be preferable if generating only a few samples, DDPP quickly offers faster sampling as the number of samples grows. (b) We demonstrate how the ESMFold based reward function cannot be parallelized on a single GPU and that the computational overhead becomes even more pronounced as sequence length increases.

To further analyze the computation time tradeoff of inference time methods compared to DDPP we examine their computational overhead on the task of generating designable $\beta$-sheet protein of sequence length 90. We expect that the computational tradeoff of inference time method vs fine-tuning method to favor fine-tuning based methods as the number of generated samples scales and as the computation required to evaluate the reward function increases. We examine the designable $\beta$-sheet task as its reward function is especially onerous to compute as it involves folding a protein with ESMFold. This process does not parallelize well on even on an A100 80 GB GPU as show in Figure 6b where we see that reward computation time scales linearly with batch size. Moreover, the

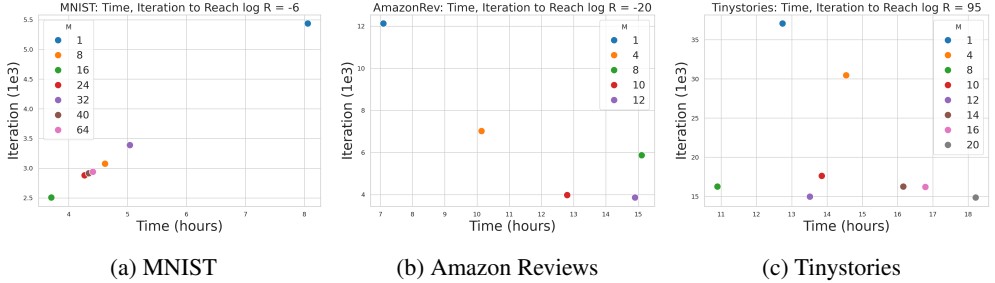

(a) MNIST        (b) Amazon Reviews        (c) Tinystories

Figure 7: Iterations and Time required to reach a threshold value of $\log R(x_0)$ for varying number of MC samples $M$ in DDPP-IS. Threshold values are computed using training curves smoothed with a rolling average, to better capture training trends.

reward function computation becomes even more expensive if scale sequence length as one might do in many real world tasks due to an $O(N^3)$ (where $N$ denotes sequence length) operation involving pairwise residue interactions.

To evaluate computational efficiency we compare the overall computation time on a single A100 80 GB GPU required to generate different numbers of sequence length 90 samples from a given method, as shown in Figure 6a. To compute the time for DDPP to generate samples, we first measure the training time to convergence and add this to the inference time required to sample from the fine-tuned model. For best of 10 and SVDD we simply generate the specified number of samples and record elapsed computation time. We observe in Figure 6a that while SVDD and best of 10 are fast if one needs to generate only a few samples, they quickly become significantly more expensive as the number of samples needed increases. In particular, to generate only 1000 samples DDPP, combining both its training and inference time, requires only 1.99 hours while best of 10 and SVDD require 6.15 and 37.5 hours, respectively. This means that for generating even this relatively small set of 1000 samples, amortized sampling using DDPP is up to **18x** faster than comparable inference time methods while generating higher quality samples. Moreover, this inference time gap only increases as the number of generated samples increases and would also become more severe were sequence length to increase, as evidenced by Figure 6b.

Of course, the utility of amortized sampling methods for reducing overall computation time is dependent on the number of samples required and the computational overhead of the reward function. If only a few samples are needed and the reward function is cheap, it is advisable to use an inference time method such as best of $N$ or SVDD to generate samples. However, if a large number of samples must be generated or the reward is expensive amortized sampling approaches like DDPP are preferred. Indeed, this is one of the ultimate motivations of using RLHF algorithms in autoregressive models instead of methods like best of $N$ – the overall computation required to fine-tune a pre-trained model and subsequently sample from the fine-tuned model is much cheaper than generating $N$ samples and selecting the best one many times over.

### E.3 ANALYSIS OF IMPACT OF THE NUMBER OF MC SAMPLES $M$ IN DDPP-IS

We include an ablation comparing the impact of the number of MC samples $M$ used in DDPP-IS, on the datasets: MNIST, Amazon reviews, and Tinystories. The training steps and process times at which the model achieves a certain reward threshold, for different $M$, are plotted in Figure 7.

We observe that a larger number of samples generally improves the reward at a faster rate per iteration (gradient step), while each iteration generally takes more time, based on how expensive the reward function is to evaluate. Each gradient step involves a single call to the denoising model, and $M$ calls to the reward function. Therefore, when increasing the number of samples $M$, it results in a trade-off with fewer calls to the denoiser, while having more calls to the reward function, to achieve the same reward. In addition, when using more samples, it results in a loss with lower variance, which can benefit training.

For the MNIST task in Figure 7a we see that a larger number of samples achieves the reward threshold in fewer iterations, and in less time. In this task the reward function is simple (a small classifier), so a larger number of samples doesn't add too much of a time cost per iteration, and a larger number of samples is preferred. On the other hand, for the Amazon reviews task in Figure 7b this manifests

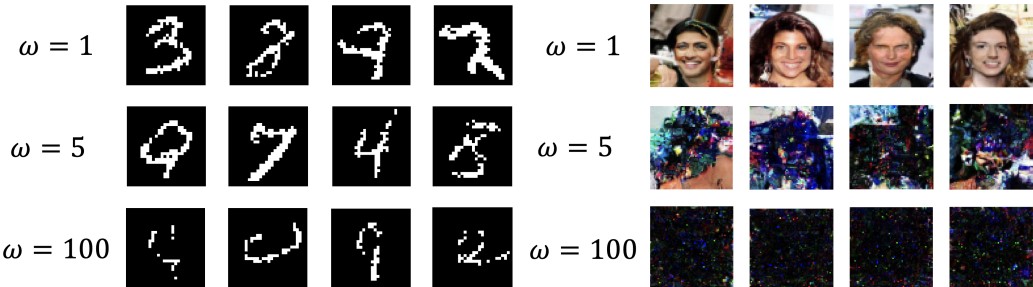

Figure 8: Discrete guidance samples for MNIST (left) and CelebA (right). As guidance scale increases sample quality decreases, especially for CelebA, while reward increases (see Table 2 and Figure 3). We view the generation of these high reward, low quality samples as $\omega$ increases as a form of reward hacking.

in the trend that as $M$ increases, the number of iterations to reach the reward threshold generally decreases while the overall time increases. For this task, the reward model is expensive to evaluate, so a smaller number of samples is more time-efficient. Finally, for tasks such as Tinystories, shown in Figure 7c, where the reward function is more expensive than MNIST but less so than Amazon reviews the interplay is more complicated. Increasing the number of Monte Carlo samples to $M = 8$ improves the variance properties of the loss curve, which leads to an improvement in both iterations and overall time. As $M$ increases beyond this, the number of iterations to convergence decreases slightly at the cost of more overall computation time. A suggestion informed by these experiments is that, for a given time budget, $M$ should be treated as a hyperparameter for tuning, with ranges over lower values for tasks with more expensive reward functions.

### E.4 ABLATION ON PROTEIN EXPERIMENTS SEQUENCE LENGTH

Table 9: Ablation on different protein lengths using DDPP-LB. DDPP-LB still generates high quality proteins as sequence length increases.

| Sequence length | $\beta$-sheet % ↑ | pLDDT ↑ | pTM ↑ | $\log R(x_0)$ ↑ |
|---|---|---|---|---|
| 90 | $0.44 \pm 0.04$ | $0.90 \pm 0.03$ | $0.81 \pm 0.03$ | $3.70 \pm 0.19$ |
| 150 | $0.71 \pm 0.04$ | $0.74 \pm 0.01$ | $0.66 \pm 0.04$ | $4.56 \pm 0.28$ |
| 200 | $0.56 \pm 0.09$ | $0.77 \pm 0.13$ | $0.75 \pm 0.11$ | $4.36 \pm 0.32$ |
| 250 | $0.64 \pm 0.01$ | $0.91 \pm 0.05$ | $0.89 \pm 0.02$ | $4.78 \pm 0.09$ |

We investigate further the performance of DDPP on the designable $\beta$-sheet task as the sequence length scales. To this end, we repeat our $\beta$-sheet experiment for DDPP-LB over additional protein sequence lengths of 150, 200, and 250 (we note that the length 90 we used in our original experiments was selected due to constraints regarding wet lab experimental protocol). We maintain all experimental settings, but for each of the different sequence lengths we perform a grid search over the reward temperature parameter $\beta$ and the learning rate. We selected a learning rate of 1e-6 for each additional sequence length, while for reward temperature we selected a setting of $\beta = 0.0625$ for sequence lengths 150 and 250 and maintained the original reward temperature of $\beta = 0.125$ for the sequence length 200 task. We report our results in Table 9, where we see that DDPP-LB can still generate proteins according to the target distribution even as protein sequence length increases. In fact, DDPP-LB seems to generate sequences with higher reward as we increased sequence length, an observation which follows our intuition that the DPLM base model should is better at generating slightly longer protein sequences as miniproteins of short lengths like 90 are relatively rare in the base model's training set compared to slightly longer proteins.

### E.5 DISCRETE GUIDANCE EXPERIMENTS

To help compare DDPP's performance to inference-based methods we compare against discrete guidance as proposed in Nisonoff et al. (2024). Discrete guidance requires both a differentiable reward as well as a reward which may be evaluated for partially masked (noisy) states. Unfortunately, our text tasks require a retokenization step as the reward models use a different tokenization than does

the pre-trained MDLM model we employ. As tokenization is a non-differentiable operation we are unable to evaluate discrete guidance on our text tasks. Further, since discrete guidance requires the reward function be evaluated on noisy states we are also prevented from evaluating it on our protein task. This is because our protein reward function uses ESMFold, a complicated protein folding model, which is only defined on full protein sequences. However, our image tasks fulfill both criteria and as such we evaluate discrete guidance on the MNIST and CelebA tasks.

As our initial image classifiers (used as reward functions) are not defined on partially masked states we trained noisy versions of them for use in discrete guidance. To train the noisy reward functions we follow the recommendations of Nisonoff et al. (2024) by training on the same dataset as the original classifiers and noising the sampled datapoints according to the same forward process as the fine-tuned diffusion model. The noisy classifiers performed nearly as well as the original, non-noisy classifiers with a test set accuracy for MNIST degrading from 99% for the non-noisy classifier to 98% for the noisy classifier and from 96% for the non-noisy to 95% for the noisy classifier on CelebA. Final reward evaluations are performed by evaluating samples generated using discrete guidance on the noisy reward models with the original, non-noisy rewards. BPD values for this baseline are computed by evaluating the base model's ELBO on images sampled using guidance (due to lacking an analogous ELBO formula for guidance-based sampling). Evaluation protocol follows that used for DDPP and other baselines, described in more detail in Appendix D.1.2 for MNIST and Appendix D.3 for CelebA. Results are computed across three seeds for each guidance scale.

Guidance results are shown in Table 2 and Figure 3. We evaluate on rewards with the same temperature $\beta$ as the other baselines ($\beta = 5$ for MNIST and $\beta = 10$ for CelebA). We have an additional multiplier in the guidance scale $\omega$. For settings corresponding to those of DDPP ($\omega = 1$ for MNIST and $\omega = 1$ for CelebA) and other baselines we observe that discrete guidance either improves mean reward by a small amount (or decreases it) compared to the base model and does not approach the performance of DDPP. We scale the guidance scale by 5x and 100x over the original guidance scale and, as expected, observe an increase in mean reward as the guidance scale increases, even becoming competitive with the mean reward of DDPP on MNIST. However, this improvement in reward coincides with a crippling decrease in sample quality as indicated by the increase in BPD values. In Figure 8 we show guided samples for both MNIST and CelebA with increasing guidance strengths, where we see that as guidance strength increases an instance of reward hacking occurs, with the guided samples achieving high reward under the classifier while being of low sample quality.

