# OpenReview forum: "Steering Masked Discrete Diffusion Models via Discrete Denoising Posterior Prediction"
_ICLR.cc/2025/Conference — ICLR 2025 Poster_

### Official Review · Reviewer_oyrv · 2024-11-02

**Soundness:** 2
**Presentation:** 3
**Contribution:** 2
**Rating:** 5
**Confidence:** 5

**Summary:**

The paper introduces Discrete Denoising Posterior Prediction (DDPP), a framework for guiding Masked Discrete Diffusion Models (MDMs) toward a reward model. DDPP offers three scalable, simulation-free objectives—DDPP-IS, DDPP-LB, and DDPP-KL—suited for fine-tuning MDMs in tasks like image modeling, protein design, and text generation.

**Strengths:**

1. The paper presents DDPP, a method to align Masked Diffusion Models (MDMs) with reward functions directly at inference, skipping RL and costly sampling typical for autoregressive models.

2. DDPP is backed by theories.  Additionally, its three variants—DDPP-IS, DDPP-LB, and DDPP-KL— are designed for both differentiable and non-differentiable rewards. These enable efficient MDM fine-tuning with strong performance across tasks, which is impressive.

3. The writing flow of this work is somehow clear. I find the presentation smooth and easy to follow.

**Weaknesses:**

1. First of all, while the presentation was good, I found the paper’s theoretical parts were not very easy to follow. I would suggest simplifying theory sections and providing illustrations to improve readability. It is still not crystal clear to me what are the motivations and the differences for making such 3 variants. For instance, for DDPP-KL, which involves a reverse KL divergence objective, the authors break down its computational pathway further, which might help readers better grasp the core mechanisms quickly.

2. Since the study centers on MDMs, it remains unclear if DDPP can generalize to other types of discrete diffusion models or if the results depend heavily on the MDM-specific setup.

3. The authors claimed that DDPP enabled efficient MDM fine-tuning across many domains. There's still limited discussion of the computational costs relative to baseline approaches. For example, I assume SVDD is computationally friendly as it is purely inference time. Furthermore, it can balance memory and time for lighter computation demands. Thus, in terms of computation efficiency, I would assume DDPP is not superior to SVDD. Would the authors comment on this?

**Questions:**

1. Could DDPP be adapted to other types of discrete generative models beyond MDMs such as autoregressive or flow-based discrete models?

2. Relevant to weakness, how does the computational cost of DDPP compare to other fine-tuning/inference-time approaches, especially in larger-scale applications?

3. Protein design applications can involve sequences of varying lengths, which might affect DDPP’s performance. How does the method scale with longer or structurally complex sequences? I suppose an ablation study on sequence length is critical.

4. The paper introduces three variants of DDPP (DDPP-IS, DDPP-LB, and DDPP-KL) but does not make it clear which one is preferred in what scenarios. Could the authors comment on selecting the right variant given specific reward functions or computational constraints?

---

> ### Author Response · Authors · 2024-11-21
> **Rebuttal (1/2)**
>
> We would like to thank the reviewer for their time and feedback on our work. We are heartened to hear that the reviewer feels that DDPP’s empirical performance across a diverse set of tasks was “impressive”. We answer each of the reviewer’s concerns and questions in turn below.
>
> ## On the differences between the DDPP objectives
>
> > It is still not crystal clear to me what are the motivations and the differences for making such 3 variants
>
> > The paper introduces three variants of DDPP (DDPP-IS, DDPP-LB, and DDPP-KL) but does not make it clear which one is preferred in what scenarios. Could the authors comment on selecting the right variant given specific reward functions or computational constraints?
>
> We appreciate that in the paper it may not have been clear which DDPP variant is preferred in each scenario. We cover the motivations for each DDPP variant and when to use them in detail in our global response and kindly invite the reviewer to consult it there.  We briefly summarize our comments in the global response here. We recommend DDPP-KL in low to middle dimensionality cases when reward function is differentiable. For non-differentiable and especially high dimensional cases we advise the use of DDPP-LB due to its cheap approximation of the partition function and our empirical results suggesting its superiority. Finally, if the reward function is especially cheap to evaluate DDPP-IS may be a competitive option as we see that, especially with larger numbers of Monte Carlo samples, the method converges quite quickly despite requiring more queries to the reward function.
>
> ## Is DDPP applicable to other discrete generative models?
> > Since the study centers on MDMs, it remains unclear if DDPP can generalize to other types of discrete diffusion models or if the results depend heavily on the MDM-specific setup.
>
> > Could DDPP be adapted to other types of discrete generative models beyond MDMs such as autoregressive or flow-based discrete models?
>
> We thank the reviewer for the question and agree that we did not make this as clear as possible in the manuscript. While our paper focuses on masked diffusion models, our theory applies equally to other sorts of discrete diffusion models with any specification of forward process, as well as the related and recent discrete flow matching methods [1]. DDPP-LB and DDPP-IS rely on the satisfaction of the relative trajectory balance constraint in a way that is equally applicable for any forward process, while DDPP-KL relies on a reverse KL formulation which is generalizable in the same way. In fact, we had initially planned to also include experiments using discrete flow matching but could not as open-sourced pre-trained models were not available for use. We focused the exposition in our paper on masked diffusion models as they are currently state of the art for discrete diffusion models and of most current relevance for ML practitioners.
>
> The reviewer also brings up an interesting question of whether DDPP is applicable to autoregressive generative models.  The ultimate goal of the different DDPP objectives is to ensure the relative trajectory balance constraints [2] are satisfied.  While in DDPP we must contend with the difficulty of computing the likelihood of a point under the pretrained diffusion model (which is very difficult, especially in discrete spaces), for autoregressive models where likelihood evaluation is simple the relative trajectory balance constraint simplifies and becomes
>
> $$Z p^{post}(x) = p^{pre}(x) R(x)$$
>
> Since likelihoods are simple to compute for autoregressive models this can immediately be converted into a log-ratio objective
>
> $$ L(x) = (\log Z + \log p^{post}(x) - \log p^{pre}(x) - \log R(x))^2 $$
>
> This is, in fact, a variant of the objective function used in the recent paper “Amortizing Intractable Inference in Large Language Models” [3]. While [3] employs a variant of DDPP-LB, DDPP-IS should also be applicable there (though expensive due to the required simulation). The performance in autoregressive models of DDPP-IS and whether DDPP-KL is still applicable there are interesting future avenues for potential research.

---

> > ### Author Response · Authors · 2024-11-21
> > **Rebuttal (2/2)**
> >
> > ## Computational overhead vs inference time methods
> > > The authors claimed that DDPP enabled efficient MDM fine-tuning across many domains. There's still limited discussion of the computational costs relative to baseline approaches. For example, I assume SVDD is computationally friendly as it is purely inference time. Furthermore, it can balance memory and time for lighter computation demands. Thus, in terms of computation efficiency, I would assume DDPP is not superior to SVDD.
> >
> > We thank the reviewer for the important question and kindly ask them to consult our global response which discusses in detail the difference in overall runtime between DDPP and SVDD.  In summary, the runtime of generating many samples with SVDD is heavily dependent on the computational complexity of a particular task’s reward function. For example, in the protein task setting where the reward function involves folding a generated sequence with ESMFold we found that **generating 1000 samples with SVDD took 37.5 hours while the process of fine-tuning DDPP and subsequently using it for amortized sampling took only 1.99 hours**. We have also added a more detailed discussion of this point in Appendix E.2 in the updated PDF.
> >
> > ## Ablation on sequence length for protein experiments
> > > Protein design applications can involve sequences of varying lengths, which might affect DDPP’s performance. How does the method scale with longer or structurally complex sequences? I suppose an ablation study on sequence length is critical.
> >
> > We thank the reviewer for bringing up this interesting point.  We have provided in the updated PDF an ablation for DDPP-LB on the $\beta$-sheet task in Appendix E.4 where we measure our performance when generating sequences of length 90 (the length originally in the paper), 150, 200, and 250. We find that DDPP-LB’s ability to generate designable sequences with significant amounts of $\beta$-sheets in their secondary structure remains intact across different sequence lengths.
> >
> > ## Closing comments
> > We thank the reviewer again for their time and valuable feedback. We hope that our rebuttal addresses their questions and concerns and that the updated PDF with additional experiments and key clarifications requested by the reviewer allows them to endorse this paper more wholeheartedly and potentially upgrade their score. We are happy to address any further comments and concerns the reviewer might have and we thank the reviewer again for their time.
> >
> >
> > ## References
> > [1] Gat, Itai, et al. "Discrete flow matching." arXiv preprint arXiv:2407.15595 (2024)
> >
> > [2] Venkatraman, Siddarth, et al. "Amortizing intractable inference in diffusion models for vision, language, and control." NeurIPS (2024)
> >
> > [3] Hu, Edward J., et al. "Amortizing intractable inference in large language models." ICLR (2024)

---

> > > ### Author Response · Authors · 2024-11-24
> > > **Kindly awaiting more feedback**
> > >
> > > Dear reviewer,
> > >
> > > We are very appreciative of your time and constructive comments. As the end of the rebuttal period is fast approaching we would like to have the opportunity to answer any lingering questions or doubts that may remain. We would like to note that in our rebuttal we followed your great suggestions and included new ablations on computational time overhead and protein sequence length. We also tried to highlight in both our global response and the rebuttal response the differences between the different DDPP objectives, their motivations, and use cases.
> > >
> > > We would be happy to engage in any further discussion on these points or any other salient points that the reviewer finds important, please let us know! We thank the reviewer again for their time and if the reviewer finds our rebuttal and new experimental findings satisfactory we also would appreciate it if the reviewer could potentially consider revising their assessment of our paper.

---

> ### Comment · Reviewer_oyrv · 2024-11-24
>
> I thank the authors for the answers. However, upon further and careful review of this work and the line of relevant works in the literature, I found concrete concerns with this work.
>
> Firstly, regarding the novelty of this work,  technically, I found it is just a variance of path consistency learning in the discrete setting (gflownet papers often used these loss functions). Thus, it becomes unclear (1) why this algorithm is special in the discrete setting as compared to standard continuous setting and (1) what the technical difficulty is for making this extension.
>
> Second, the experimental part is incomplete. This work lacks lots of baselines/discussions that should be compared with, in terms of fine-tuning-based alignments, there is no comparison with classifier free guidance, DDPO/DPOK, and direct reward backpropagation which are all prevailing approaches for guided generations. Comparisons with the most common inference-time guidance methods are also missing, such as classifier guidance in discrete diffusion [1]. In a continuous setting, it is common that new alignment/guidance methods will take classifier guidance as a standard baseline to compare with. In a discrete setting, it is also probably considered essential.
>
> [1] https://arxiv.org/abs/2406.01572

---

> > ### Author Response · Authors · 2024-11-28
> > **Response (1/3)**
> >
> > We thank the reviewer for their continued participation during this rebuttal period. We now address the new questions posed by the reviewer.
> >
> > ## Novelty of DDPP vs. GFlowNets
> >
> > We appreciate the reviewer’s concern that DDPP and GFlowNets share similarities. We begin by first highlighting that we sought to explain the chief difference between DDPP and the most relevant GFlowNet approach in Relative Trajectory Balance (RTB) on lines 211-215 in our manuscript.
> >
> > Both DDPP and RTB (a GFlowNet objective) attempt to match a detailed balance condition. We highlight that the notion of satisfying a detailed balance criterion is a very general problem setup and the chief novelty of any approach rests on **algorithmic components to satisfying detailed balance**. In our DDPP, we construct a *simulation-free* objective in equation 8 to satisfy the detailed balance constraint. This is in contrast to the prevalent approach of training GFlowNets with a trajectory balance loss (path-consistency learning) [1,2] and stochastic optimal control based models under KL divergence [3]. We argue that using simulation-free training is one of the key hallmarks of diffusion models which enables scalability and unlike continuous space models and RTB, DDPP enjoys the benefits of still being simulation-free in discrete space.
> >
> > A great question asked by the reviewer is why the discrete setting in discrete diffusion models  is different for our method. We answer this question by noting that DDPP-IS and DDPP-LB heavily exploit the fact that the pretrained model is another discrete diffusion model. More precisely, DDPP leverages the denoiser parameterization of an MDM – i.e. $\hat{x}_0 = \mu{\theta}(x_t, t)$ which predicts a clean sample $x_0$ from a partially masked sample $x_t$. This specific parameterization, as well as the structure of the absorbing state diffusion process, are uniquely beneficial in discrete state space and exploited by DDPP.
> >
> > First, as shown in Appendix C.3, we explicitly use the MDM’s parameterization in order to derive a cheap Monte Carlo estimator of the subtrajectory KL divergence in equation 41 and make use of this throughout our manuscript. In particular, a continuous space diffusion model one must sample from the reverse process in order to estimate the KL divergence. However, for an MDM as all tokens are equally likely to be unmasked at any specific time $t$, all trajectories (or stated more simply, all orders of unmasking masked tokens in $x_t$ to $x_0$) are uniformly likely. This allows us to estimate the subtrajectory KL without requiring simulation to sample a trajectory $\tau(x_{0:t})$ by instead uniformly sampling an unmasking ordering. We exploit this formalism, allowed specifically by the MDM parameterization, throughout DDPP.
> >
> > Second, we hypothesize that, since the MDM denoiser directly maps to clean datapoints at all token positions (i.e., if a token is unmasked at time t, it will retain that token for subsequent inference time-steps), instead of predicting a blurry, averaged estimate of x0, in continuous space diffusion models, the estimation of $p(x_0|x_t)$ is different in discrete space than continuous. We believe this allows for better estimation in the difference between log likelihoods of pretrained and fine-tuned models in discrete vs continuous settings, as well as better estimation of the partition function in DDPP-IS, particularly for the single-step method (eq 11). We believe the exploitation of this trick further allows efficient and scalable training in comparison to a traditional GFlowNet objective like RTB. Note that in Table 1 we highlight that RTB requires T forward pass calls to compute a loss objective compared to only 1 in DDPP-LB.
> >
> > We further highlight that our objective DDPP-KL is **fundamentally different** from any GFlowNet approach since we do not need to estimate the tricky log partition function. Moreover DDPP is grounded in an annealed reverse KL objective applied to Masked Diffusion Models—which to the best of our knowledge is a novel objective and contribution.
> >
> > Finally, we argue that our DDPP steering **was deployed to create wet-lab validated proteins with more $\beta$-sheets**. Such an experimental result we argue is a novel finding as *no prior paper on GFlowNet/path learning steering approach has gone beyond in-silico metrics*. We hope that the reviewer agrees that, although our wet-lab experiments are preliminary, this bolsters the empirical caliber and novelty of DDPP.
> >
> > As a result of the points outlined, we would like to politely pushback on the reviewers assertion on the lack of novelty of DDPP compared to GFlowNets.

---

> > > ### Author Response · Authors · 2024-11-28
> > > **Response (2/3)**
> > >
> > > ## Requested Baselines
> > >
> > > We acknowledge the reviewer’s comment on including additional baselines to improve the totality of the empirical findings of our paper. Given the limited amount of time before the end of
> > > PDF update period during this rebuttal process we focus on the reviewer’s most important suggestion of comparing with the Discrete Guidance approach of Nisonoff et. al 2024.
> > >
> > > We thank the reviewer for suggesting the discrete guidance baseline which we have now included in the updated draft for the binarized MNIST and CelebA tasks. These updates are highlighted in Table 2, Figure 3, and Appendix E.5 and are reproduced below for convenience.
> > >
> > > ### MNIST
> > > | Method                   | $\log R(x_0)$ | BPD |
> > > | ------------------------ | ------------- | -------------------------------------------------- |
> > > | Guidance (scale = 1.0)   | -25.24 ± 0.26 |         0.171 ± 0.001*                 |
> > > | Guidance (scale = 5.0)  | -23.21 ± 0.21 |         0.174 ± 0.001*                   |
> > > | Guidance (scale = 100.0) | -9.32 ± 0.24  |         0.147 ± 0.001*                    |
> > > | DDPP-IS                  | -5.14 ± 1.24  | 0.130 ± 0.000                                      |
> > > | DDPP-LB                  | -5.68 ± 0.34  | 0.128 ± 0.000                                      |
> > > | DDPP-KL                  | -3.13 ± 0.06  | 0.129 ± 0.000                                      |
> > > \* The BPD values for guidance were evaluated by the base model, on generated samples
> > >
> > >
> > > ### CelebA
> > > | Method                   | $\log R(x_0)$    | BPD |
> > > | ------------------------ | ---------------- | -------------------------------------------------- |
> > > | Guidance (scale = 1.0)   | -91.10 ± 1.63    |             3.20 ± 0.01*                  |
> > > | Guidance (scale = 5.0)   | -62.75 ± 0.25    |            5.39 ± 0.01*                      |
> > > | Guidance (scale = 100.0) | -41.51 ± 0.02    |            5.15 ± 0.00*                    |
> > > | DDPP-LB                  | -6.94 ± 1.39 | 2.62 ± 0.15     |
> > > \* The BPD values for guidance were evaluated by the base model, on generated samples
> > >
> > > We observe that despite trying multiple different guidance strengths it is hard to optimize the reward effectively in comparison to DDPP. Critically, we observe that enforcing a high guidance strength improves the reward moderately but at the cost of crippling sample quality (as reported in Appendix E.5). These findings bolster our claim that DDPP is better at optimizing reward without a significant degradation in sample quality as observed in other baselines including Discrete Guidance.
> > >
> > > ## Discrete Guidance needs a differentiable reward
> > >
> > > We wish to note that Discrete Guidance as formulated in Nisonoff et. al 2024 requires a reward function to be a.) differentiable and b.) provide an output a scalar reward on partially masked sequences $x_t$. The former condition limits its application to settings where a differentiable reward is available—in contrast to DDPP-IS and DDPP-LB. The latter condition means some rewards used in our current experimental setups are inapplicable as they solely operator on $x_0$. In particular, we note that in our protein experiments we used a protein folding model, ESMFold, as the reward which makes it impossible to score partially masked sequences as required by discrete guidance. Moreover, the former condition rules out discrete guidance's use on text experiments. Our text rewards rely on pre-trained classifiers which use a different tokenization than that of the pre-trained discrete diffusion models we employ. As such, part of computing the reward involves retokenizing the samples from the tokenization of the discrete diffusion model to that of the pre-trained classifier. Unfortunately, retokenization is a non-differentiable operation, preventing us from applying discrete guidance to the Tinystories or Amazon reviews tasks.
> > >
> > > The noisy reward model training for image tasks is outlined in Appendix E.5 and in short we try to closely match the performance of the original reward for both CelebA and MNIST datasets. We observe that our noisy reward models are similar in performance to the non-noisy reward models, with the noisy MNIST reward model reaching a test set accuracy of 98% vs the non-noisy version which attained 99%. Similarly, the noisy CelebA reward model attains a 95% test set accuracy while the original, non-noisy reward model obtained 96%.

---

> > > > ### Author Response · Authors · 2024-11-28
> > > > **Response (3/3)**
> > > >
> > > > ## DPOK and Direct Backprop Baselines
> > > >
> > > > We acknowledge the reviewer’s request for the inclusion of DPOK and the direct backprop baselines. Given the tight time constraints we focused on including the discrete guidance baseline into the updated draft and we hope to have results on DPOK before the end of the rebuttal discussion period. Regarding the inclusion of a direct backprop baseline, to the best of our knowledge this is not a method that has been widely used in **discrete diffusion** models, as opposed to their continuous counterparts. In particular, in our literature review we were able to find a contemporaneous submission to ICLR in DRAKES [4]. We wish to politely note that as per ICLR guidelines (https://iclr.cc/Conferences/2025/FAQ) comparing against contemporaneous works is not required and is not grounds for rejection.
> > > >
> > > > However, if we have missed any work that is based on direct backprop for **discrete diffusion** and not contemporary with our work we would be happy to include it before the end of the rebuttal period. Please do let us know if this is the case!
> > > >
> > > > ## Closing comment
> > > >
> > > > We thank the reviewer for their feedback, especially the suggestion of additional baselines that aided us in improving our paper. We hope that with this response we have alleviated the main concerns shared by the reviewer and we invite the reviewer to consider a fresh assessment of our paper with this response in context.
> > > >
> > > > ## References
> > > >
> > > > [1] Malkin, Nikolay, et al. "Trajectory balance: Improved credit assignment in gflownets." Advances in Neural Information Processing Systems 35 (2022): 5955-5967.
> > > >
> > > > [2] Venkatraman, Siddarth, et al. "Amortizing intractable inference in diffusion models for vision, language, and control." arXiv preprint arXiv:2405.20971 (2024).
> > > >
> > > > [3] Zhang, Qinsheng, and Yongxin Chen. "Path integral sampler: a stochastic control approach for sampling." arXiv preprint arXiv:2111.15141 (2021).
> > > >
> > > > [4] Wang, Chenyu, et al. "Fine-tuning discrete diffusion models via reward optimization with applications to dna and protein design." arXiv preprint arXiv:2410.13643 (2024).

---

> ### Comment · Reviewer_oyrv · 2024-12-02
>
> I appreciate the authors' responses to address some of the concerns. However, several key points remain unresolved after careful investigation of the baselines included in this paper. Moreover, some claims you've made in the rebuttal are evidently incorrect.
>
> Firstly, the authors argue against the inclusion of certain baselines due to technical constraints. However, methods like PPO can be directly applied to discrete diffusion models without requiring significant modifications or "twisting." Additionally, crucial comparisons with standard approaches such as classifier-free guidance (conditioning on high values) and classifier guidance in other domains are notably absent. The lack of these comparisons makes it difficult to show the effectiveness of DDPP relative to existing methods,
>
> Moreover, omitting comparisons with sequential Monte Carlo (SMC) methods is another significant gap. These methods are relevant for inference in diffusion processes, and their inclusion would provide a more comprehensive understanding of DDPP's efficiency and performance. Other reviewers have flagged this part as a common concern.
>
> The authors' discussion of SVDD performance is similarly incomplete. After investigating this baseline, I found it is basically a nested-SMC whose performance depends heavily on the duplication number. However, this paper does not even include a detailed analysis or discussion of this dependency. The lack of mention or exploration of this aspect weakens the empirical evaluation, which is problematic: have the baseline implementations in this work achieved their best performances?
>
> Furthermore, the authors state that certain rewards used in their experiments are not differentiable, which limits comparisons with baselines like discrete guidance. This claim is imprecise as far as I am concerned. Many protein forward/inverse folding models, including AlphaFold, are differentiable (as highlighted in sources like (https://onlinelibrary.wiley.com/doi/full/10.1002/pro.4653). This raises questions about why such differentiable rewards were not incorporated.
>
> In summary, too many essential baselines (SMC-based methods, classifier guidance, PPO, classifier-free guidance) are absent in the current draft. While the rebuttal partially addresses some of these omissions (e.g., including a basic SMC), several critical comparisons remain missing, leaving the evaluation incomplete and unconvincing.

---

> > ### Author Response · Authors · 2024-12-03
> > **Response (1/3)**
> >
> > We appreciate the reviewer for their time, dedication, and continued participation in this rebuttal period. We now answer the new questions posed by the reviewer as well as clarifying their lingering doubts.
> > We answer these in turn below.
> >
> > ## PPO Baseline (DPOK)
> > >Firstly, the authors argue against the inclusion of certain baselines due to technical constraints. However, methods like PPO can be directly applied to discrete diffusion models without requiring significant modifications or "twisting."
> >
> > We thank the reviewer for the comments. We wish to reiterate that our previous rebuttal comment was to highlight that we prioritized the inclusion of discrete guidance in the PDF update deadline of this rebuttal period rather than not including a PPO baseline. As the reviewer correctly points out PPO based approaches can be applied to discrete diffusion models, and in this comment we follow the reviewers suggestion in their previous comment and include DPOK [1] as a demonstrative PPO based baseline. We highlight that DPOK is well-suited for diffusion models as DPOK adapts PPO by approximating the intractable KL constraint in typical RLHF based finetuning. We however note that DPOK was originally designed for continuous diffusion models and here we adapt them—as the reviewer suggests—for discrete data as “methods like PPO can be directly applied to discrete diffusion models without requiring significant modifications”.
> >
> > Specifically, we evaluate DPOK for MNIST, CelebA, and text experiments and summarize the results below for convenience:
> >
> > ### MNIST
> >
> > | Algorithm  | $\log R(x_0) \uparrow$ | FLD $\downarrow$     | BPD $\downarrow$      |
> > | ---------- | ---------------------- | -------------------- | --------------------- |
> > | Base Model | -26.90 $\pm$ ---       | 33.89 $\pm$ ---      | 0.130 $\pm$ ---       |
> > | SVDD       | **-0.03 $\pm$ 0.01**   | 34.19 $\pm$ 0.95     | ---                   |
> > | Guidance (scale 1) | -25.24 $\pm$ 0.26      | 34.67 $\pm$ 0.67     | 0.171 $\pm$ 0.001     |
> > | Guidance (scale 5) | -23.21 $\pm$ 0.21      | 37.33 $\pm$ 0.87     | 0.174 $\pm$ 0.001     |
> > | Guidance (scale 100) | -9.32 $\pm$ 0.24  | 72.19 $\pm$ 0.43 | 0.147 $\pm$ 0.001    |
> > | RTB                | -18.66 $\pm$ 2.45      | 45.97 $\pm$ 0.89     | 0.128 $\pm$ 0.000     |
> > | DPOK               | -3.39 $\pm$ 0.15       | **29.65 $\pm$ 1.05** | **0.126 $\pm$ 0.001** |
> > | DDPP-IS            | -5.14 $\pm$ 1.24       | 33.11 $\pm$ 0.71     | 0.130 $\pm$ 0.000     |
> > | DDPP-LB | -5.68 $\pm$ 0.34 | 33.76 $\pm$ 0.90 | 0.128 $\pm$ 0.000 |
> > | DDPP-KL | -3.13 $\pm$ 0.06 | 31.75 $\pm$ 0.51 | 0.129 $\pm$ 0.000 |
> >
> > ### CelebA
> >
> > | Algorithm            | $\log R(x_0) \uparrow$ | BPD $\downarrow$    |
> > | -------------------- | ---------------------- | ------------------- |
> > | Base                 | -57.31 $\pm$ ---       | 2.67 $\pm$ ---      |
> > | SVDD                 | -13.27 $\pm$ 12.38     | ---                 |
> > | Guidance (scale 1)   | -91.10 $\pm$ 1.63      | 3.20 $\pm$ 0.01     |
> > | Guidance (scale 5)   | -62.75 $\pm$ 0.25      | 5.39 $\pm$ 0.01     |
> > | Guidance (scale 100) | -41.51 $\pm$ 0.02      | 5.15 $\pm$ 0.00     |
> > | RTB                  | -60.28 $\pm$ 1.74      | **2.04 $\pm$ 0.00** |
> > | DPOK                 | -53.49                       |      2.03               |
> > | DDPP-LB              | **-6.94 $\pm$ 1.39**   | 2.62 $\pm$ 0.15     |
> >
> >
> > ### Text
> > | Algorithm  | Tinystories $\log R(x_0) \uparrow$ | Tinystories Gen PPL $\downarrow$ | Amazon $\log R(x_0) \uparrow$ | Amazon Gen PPL $\downarrow$ |
> > | ---------- | ---------------------------------- | -------------------------------- | ----------------------------- | --------------------------- |
> > | Best of 10 | 93.25 $\pm$ 0.17                   | 15.94 $\pm$ 0.03                 | -103.05 $\pm$ 0.25            | 124.45 $\pm$ 1.02           |
> > | SVDD       | 146.95 $\pm$ 1.08                  | 20.35 $\pm$ 0.03                 | -27.48 $\pm$ 10.91            | 165.86 $\pm$ 1.22           |
> > | RTB        | 107.83 $\pm$ 3.08                  | **18.53 $\pm$ 0.55**             | -35.22 $\pm$ 16.03            | 160.54 $\pm$ 12.19          |
> > | DPOK                                 |          53.35                    | 25.65  |      -1.77                              |   123.45   |
> > | DDPP-IS    | 163.45 $\pm$ 7.06                  | 20.15 $\pm$ 0.30                 | 105.16 $\pm$ 2.41             | **152.85 $\pm$ 1.64**       |
> > | DDPP-LB    | **205.76 $\pm$ 3.88**              | **19.60 $\pm$ 0.69**             | **152.08 $\pm$ 34.01**        | 167.25 $\pm$ 27.33          |
> >
> > We find that on the relatively low dimensional MNIST task (784 dimensions) DPOK generates samples with slightly worse reward but slightly better naturalness (as measured by FLD and BPD) than DDPP.  However, on the larger CelebA, tinystories, and Amazon reviews tasks we found that DPOK faced significantly more difficulty.

---

> > > ### Author Response · Authors · 2024-12-03
> > > **Response (2/3)**
> > >
> > > We saw no improvement over the base model for tinystories, while for CelebA we observed a degradation in generated reward, though samples begin to exhibit a tendency towards blonde colors (the attribute emphasized by the reward function) despite a loss in overall sample quality. For Amazon we saw an improvement in overall reward and saw that DPOK retained reasonable sequence naturalness as evidenced by its Gen PPL results. However, its mean generated reward was still significantly worse than DDPP’s, indicating its difficulty to optimize a reward in sparse settings. Our findings suggest that DPOK is inferior in more complex settings of CelebA and text than DDPP, a point that we will add in our updated manuscript.
> > >
> > > Given the challenges faced by DPOK on more complex discrete settings we opted against including it for our protein experiment, which arguably has the most complex reward. Due to time constraints we were only able to run one seed for CelebA and text tasks, though we will add more seeds in the updated manuscript.
> > >
> > > To fit a single sample into 80GB of GPU memory while training we applied LoRA with rank 4 to drastically reduce the number of trainable parameters, used a batch size of 1 (with 8 gradient accumulation steps), used the gradient detaching trick used for RTB to fit DPOK in memory, and reduced trajectory lengths. For CelebA we detach all but one step and use trajectories of length 48, while for tinystories and Amazon we detached 50% of inference steps with trajectories of length 16.
> > > ## Guidance Baselines
> > >
> > > > Additionally, crucial comparisons with standard approaches such as classifier-free guidance (conditioning on high values)
> > >
> > > We’d like to note that classifier-free guidance requires the training of a conditional diffusion model (either conditioned on target labels, or reward values), which is a substantial deviation from the problem setup tackled in this work: where an unconditional pretrained model is available, and we wish to fine-tune it to generate higher-reward samples, similar to the setup of RLHF in LLMs. As we state explicitly, our problem setting is that we wish to sample from $\pi_0(x_0) \propto p^{pre}(x_0) R(x_0)$, which assumes **a pretrained unconditional MDM**. We do not assume access to a conditional model as well, which would enable CFG as the reviewer requests. Indeed, in typical guidance setups both unconditional and conditional models are trained simultaneously. The current steering task we consider is framed as sampling from a desired target $\pi_0$, which does not afford us the luxury of also having a conditional model. As a result, we very respectfully do not view this baseline as an appropriate baseline.
> > >
> > > >  classifier guidance in other domains are notably absent … Furthermore, the authors state that certain rewards used in their experiments are not differentiable, which limits comparisons with baselines like discrete guidance. However, many folding models, including AlphaFold, are differentiable (as highlighted in sources like (https://onlinelibrary.wiley.com/doi/full/10.1002/pro.4653). This raises questions about why such differentiable rewards were not incorporated.
> > >
> > > We appreciate the reviewer’s concern, however, classifier guidance requires the differentiability of the reward function. We previously compared classifier guidance on the rewards where this was the case (MNIST and CelebA), but this is not possible for the other domains (text and proteins) as discussed in part 2 of our previous response to the reviewer. For clarity and further specificity, we elaborate on these points below.
> > >
> > > For text experiments we note that the non-differentiable part is the tokenization of text and not the reward which prevents the use of discrete classifier guidance. This also means that DDPP-KL is not applicable in this setting. We understand that this subtle point may not have been sufficiently clear in the draft and we will highlight this detail in our updated manuscript.
> > >
> > > For our main protein experiment, our reward is also not differentiable. In particular, our reward contains three terms with zero gradient almost everywhere. As detailed in Section D.2.1 of our manuscript, we use the proportion of residues predicted to form $\beta$-sheets, the sequence entropy, and the exposed hydrophobic residues. None of these scores has a meaningful gradient with respect to the input, as they are all based on counting the fraction of residues fitting a specific criteria.

---

> > > > ### Author Response · Authors · 2024-12-03
> > > > **Response (3/3)**
> > > >
> > > > > This raises questions about why such differentiable rewards were not incorporated.
> > > >
> > > > For our protein experiments we aimed to create a meaningful reward that was in our expert opinion most likely to correlate with the successful design of predominantly beta-sheet proteins in the wet lab. Some of these aims are inherently not differentiable. We were successful in this endeavor as shown in our biological results confirmed in a wet-lab in Figure 2. We leave the design of differentiable rewards that correlate with this wet lab task to future work.
> > > >
> > > > > Moreover, omitting comparisons with sequential Monte Carlo (SMC) methods is another significant gap. These methods are relevant for inference in diffusion processes, and their inclusion would provide a more comprehensive understanding of DDPP's efficiency and performance. Other reviewers have flagged this part as a common concern.
> > > >
> > > > We acknowledge the reviewer’s question but would like to politely push back on their assertion. We have in fact included an SMC baseline with twisted SMC for autoregressive models for the text domain, in appendix E.1, and reproduced in the global rebuttal (part 3). In addition, as pointed out by the reviewer, SVDD is essentially an SMC method, which we do compare against. We note that in the original SVDD paper, this method outperforms simple SMC baselines in all their settings and as a result our inclusion of SVDD provides a more robust baseline than a simple inclusion of a naive SMC baseline. Finally, we wish to highlight that the use of SMC based approaches like SVDD incurs significant inference overhead which in certain domains is a computation concern where it is expensive to query the reward. For instance, this is the case for our protein experiments and as a result it is preferable to find an amortized sampler like DDPP in contrast to SVDD which takes 37.5 hours to generate 1000 samples compared to 1.99 hours for training and generation in DDPP (see global rebuttal part 2 and Appendix E.2).
> > > >
> > > >
> > > > > The authors' discussion of SVDD performance is similarly incomplete. … Are the implementations of baselines in this work achieving the best performances?
> > > >
> > > > We thank the reviewer for the comment. We begin by noting that SVDD is a contemporaneous submission to ICLR and as per ICLR guidelines does not require comparison. Despite this, we believe it is an important method in the literature and we sought to go above and beyond and compare against it as it is robustly better than vanilla SMC based approaches.
> > > >
> > > > We first note that we used the publicly available code provided in SVDD whose hyperparameters we adapt to our settings. Moreover, as discussed in the SVDD paper (section 7.2, Figure 3) [2], more particles $M$ result in higher performance (such as reward), at the cost of a linear growth in inference time. Our experimental setups strived to increase the reward as a function of particles without incurring an astronomical computational overhead. Specifically, we observe inference using SVDD being a computational bottleneck for reward functions which are expensive to evaluate. This inference time is a bottleneck to adding more particles for SVDD. For instance, on the protein task - with $M=10$ particles, on a single 80GB A100 GPU, SVDD takes 37.5 hours to generate 1000 samples from the target distribution. In comparison, on the same hardware DDPP took only 1.99 hours total to train and generate the 1000 samples, so that **DDPP took 18x less time than SVDD in order to generate better samples**. A similar situation holds for CelebA where SVDD takes around 56 A100 GPU hours to generate 600 samples for evaluation with $M=10$ particles, in contrast to DDPP which takes 12 A100 GPU hours for training, and an additional 5 hours for sample generation. The global rebuttal (part 2), and our appendix E.2 include more details on the time required for SVDD compared to our method. In light of our findings, we argue that DDPP is significantly more efficient than SVDD and is suited for tasks where querying the reward is expensive. We thank the reviewer again for their question and we will update the paper with these details.
> > > >
> > > > ## References
> > > >
> > > > [1] Fan, Y., Watkins, O., Du, Y., Liu, H., Ryu, M., Boutilier, C., ... & Lee, K. (2024). Reinforcement learning for fine-tuning text-to-image diffusion models. Advances in Neural Information Processing Systems, 36.
> > > >
> > > > [2] Li, X., Zhao, Y., Wang, C., Scalia, G., Eraslan, G., Nair, S., ... & Uehara, M. (2024). Derivative-free guidance in continuous and discrete diffusion models with soft value-based decoding. arXiv preprint arXiv:2408.08252.

---

### Official Review · Reviewer_2bfH · 2024-11-03

**Soundness:** 3
**Presentation:** 2
**Contribution:** 3
**Rating:** 6
**Confidence:** 3

**Summary:**

This paper addresses the task of controlled generation of a pretrained masked diffusion model (MDM) $p^{\rm pre}(x)\approx p _0(x)$ towards an altered target distribution $\pi _0(x)\propto p _0(x)R(x)$, where $R(x)$ is a reward function. The approach involves fine-tuning the MDM to align with the new target distribution. The paper introduces a simulation-free framework called "discrete denoising posterior prediction" (DDPP) to train the amortized MDM. The framework has shown promising empirical results for various tasks in image, text, and proteins.

**Strengths:**

The paper addresses the important problem of controlled generation, which has significant applications as discussed in the paper. The proposed DDPP framework, consisting of three different versions, is a novel approach that efficiently tackles this problem without simulating the reverse diffusion process. The empirical results demonstrate the superiority of the DDPP method over several baseline methods, including best-of-$N$, SVDD, and RTB. This highlights the effectiveness and potential of the proposed framework in the context of discrete diffusion models.

The paper is well-written and clearly organized, and provides sufficient details to understand and replicate the experiments. However, further improvement can be made in the clarity of the mathematical explanations.

**Weaknesses:**

The methodologies introduced in the paper are somewhat confusing to me, and I am unable to fully grasp the underlying intuition behind their design. Please check the question part for details.

The writing of the paper, particularly the mathematical explanations, could benefit from improvement. I have provided comments and suggestions in the part regarding these issues.

**Questions:**

1. Questions for the framework.

- Could the author(s) provide further insight into the rationale behind choosing an $L^2$-based loss in learning $q _\theta({\bf x} _0,...,{\bf x} _{t-1}|{\bf x} _t,\hat{\bf x} _0)$ in DDPP-IS/LB? The use of an $L^2$-based loss comparing the difference of log-densities is not commonly seen in the literature of learning probability distributions (e.g., variational inference), and it requires learning or estimating the normalizing constant ${\cal Z} _{\pi _t}({\bf x} _t)$, which involves computational overhead. Furthermore, what is the probability of ${\bf x} _{0:t}$ in the losses in equations (7,8,11,43)? Although this can be any trajectory, it would be beneficial to clarify your choice for the experiments.

- In DDPP-KL, the author(s) employ the reverse KL loss instead of forward KL loss. However, this requires the differentiability of reward function $R(x)$, and needs the Reinmax trick to estimate the discrete gradients. It would be valuable if the author(s) could elaborate on why the forward KL loss is not suitable in this context.

2. As discussed in Ou et al. (2024) and later Zheng et al. (2024), the time component in MDMs does not play a significant role. An MDM $\mu _\theta({\bf x},t)\in(\Delta^d)^n$ is equivalent to an any-order autoregressive model, i.e., it predicts the one-dimensional conditional probability distributions of the masked positions given the observed ones in a partially observed sequence. If we formulate the MDM without time component, how would your fine-tuning framework be adapted?

3. Questions regarding the experiments.

- In the MNIST experiment, what would be the possible reason why SVDD having a higher reward value $\log R({\bf x} _0)$ compared with DDPP? Additionally, why is there no BPD metric for the samples generated by SVDD?

- In the CelebA experiment, it seems that the quality of samples from DDPP (Figure 3) is worse than those from the pretrained model (Figure 5).

4. Minor comments

- In lines 89 and 97, it seems that the notation $\cal V$ should be $\cal X$ instead.

- In line 98, the RHS of the equation $p({\bf X}={\bf x})$ is a bit confusing. Does it imply that each position is mutually independent? Clarification would be helpful.

- In line 105, it might be more appropriate to say "drop $t$ and write $i$" instead. Since the paper uses a discrete time setting, replacing $\tau({\bf x} _{0:1})={\bf x} _1\to\cdots\to{\bf x} _t\to{\bf x} _0$ with $\tau({\bf x} _{0:T})={\bf x} _T\to\cdots\to{\bf x} _t\to{\bf x} _0$ would be more appropriate.

- In section 2.1, when dealing with multiple dimensions, $\mu _\theta({\bf x},t)$ should be in $(\Delta^d)^n$, which predicts the probabilities $\Pr({\bf X} _0^i=\cdot|{\bf X} _t={\bf x})$.

- In line 260, it would be better to move the training objective (Eq. 11) to this location for better readability.

- Furthermore, I noticed a few typos or errors in the paper. For example, in line 323, it states "In settings, where the reward model is differentiable", and in line 430, it says "We provide full a deeper description of evaluation metrics and experimental setup". I strongly recommend that the author(s) carefully review and polish the writing in this paper during the rebuttal period to ensure clarity and accuracy.

I would be happy to raise the score if the author(s) could address my concerns.

**References**

Ou et al. Your Absorbing Discrete Diffusion Secretly Models the Conditional Distributions of Clean Data. Arxiv 2406.03736.

Zheng et al. Masked diffusion models are secretly time-agnostic masked models and exploit inaccurate categorical sampling. ArXiv 2409.02908.

---

> ### Author Response · Authors · 2024-11-21
> **Rebuttal (1/3)**
>
> We would like to thank the reviewer for the time and effort they spent on reviewing our work. We are appreciative of the fact that the reviewer found DDPP to be a “novel approach” for fine-tuning to sample from the target product distribution supported by better empirical performance. We now address the concerns raised by the reviewer.
>
> ## Clarifications about the DDPP framework
> > Questions for the framework.Could the author(s) provide further insight into the rationale behind choosing an L2-based loss in learning qθ(x0,...,xt−1|xt,x^0)in DDPP-IS/LB? The use of an L2-based loss comparing the difference of log-densities is not commonly seen in the literature of learning probability distributions (e.g., variational inference), and it requires learning or estimating the normalizing constant Zπt(xt), which involves computational overhead. Furthermore, what is the probability of x0:t in the losses in equations (7,8,11,43)? Although this can be any trajectory, it would be beneficial to clarify your choice for the experiments.
>
> The reviewer is correct that historically, the typical way of comparing two probability distributions has been through the lens of KL divergence (Variational Inference and Equilibrium Propagation, as the reviewer points out), and more generally other notions of divergence (f-divergences). However, multiple recent works show that using an $L_2$ based loss to compare the difference of log-densities does provide a valid measure of training signal, as is highlighted in [1-5]. In particular, it has also been shown that for certain choices of the sampling density under which the expectation is taken, this method can give the same expected gradients as KL divergence [3]. While it does require learning or estimating the partition function, it has already been shown in related work that it has the potential for better mode coverage and more accurate sampling. Such approaches of estimating the log partition function from multiple trajectories as well as other tricks for training such methods have been explored previously in [4]. We hope that this clarifies the reviewer’s concerns regarding the $L_2$ objective between the two log probabilities as a valid training signal for learning the right probabilities at convergence.  Finally, with regards to the sampling distribution of $x_{0:t}$, in practice we sampled trajectories from a replay buffer filled with on-policy samples from the current fine-tuned model.
>
>
> > In DDPP-KL, the author(s) employ the reverse KL loss instead of the forward KL loss. However, this requires the differentiability of reward function R(x), and needs the Reinmax trick to estimate the discrete gradients. It would be valuable if the author(s) could elaborate on why the forward KL loss is not suitable in this context.
>
> We agree with the reviewer's point that in using the reverse KL objective we require the reward $R(x)$ to be differentiable. It is tempting to consider the forward KL divergence which is commonly done in regular generative modeling but this unfortunately **fundamentally not possible** because this would require an empirical sampling from the true posterior $\pi_0(x_0)$ we are trying to match
>
> $$
> KL(\pi_0(x_0) || p^{pre}_0(x_0) R(x_0) ) = \mathbb{E}[ \log \pi(x’) - \log  p^{\text{pre}}_0(x’) R(x’) ]
> $$
>
> where the expectation should be taken as $\mathbb{E}_{x \sim \pi_0}$ (we could not get OpenReview to render the subscript of the expectation in the equation above properly). As we see here because we are in the fine-tuning/steering setting we do not have access to data samples from $\pi_0$ which prevents us from sampling in from $\pi_0$ in the expectation for the forward KL above. If we did have samples then this would reduce to any other generative modeling problem and would be significantly easier to solve.
>
> We understand that this subtle point may not have been sufficiently clear in the description of DDPP-KL and we have added a sentence in the updated draft (lines 326-327) to reduce any further doubt on why the forward KL is an unsuitable objective for the fine-tuning setting as we considered in our problem definition in Eqn 1.

---

> > ### Author Response · Authors · 2024-11-21
> > **Rebuttal (2/3)**
> >
> > ## On time parameterization
> > > As discussed in Ou et al. (2024) and later Zheng et al. (2024), the time component in MDMs does not play a significant role. … If we formulate the MDM without time component, how would your fine-tuning framework be adapted?
> >
> > The question raised by the reviewer is quite interesting. As they mention, the time component is not needed in MDMs as the number of masked tokens effectively corresponds to the time in the diffusion process. As such, the theory of our DDPP framework translates directly to using the number of masked tokens in the sequence to denote time, and when putting this into practice only requires one to remove the time input to the fine-tuned denoisers. In fact most of our experiments actually use networks which do not take time as input and instead use the number of masked tokens to infer it. In particular, for all experimental tasks besides proteins we used the model architecture of MDLM [10] which itself *does not* use a time variable as input to the network and relies on the number of masked tokens to infer time.
> >
> > ## Reasons for SVDD having higher reward on MNIST
> > > In the MNIST experiment, what would be the possible reason why SVDD having a higher reward value log⁡R(x0) compared with DDPP?
> >
> > The reviewer is correct that for MNIST, SVDD obtains a higher reward than DDPP. A likely reason for this is due to the simple nature of the reward function. Since the reward function assigns a large value for any even digit, the pre-trained diffusion model has a 50% chance of generating high reward outputs to begin with. SVDD is an inference time procedure which generates $N$ particles at each step of the pre-trained diffusion process (here $N=10$), and picks one based on estimated future reward. For this experiment, it’s likely that these particles have high reward. We point out that other reward functions, such as CelebA, and the text tasks (Tinystories and Amazon reviews) assign a higher value to behavior that is more rare under the prior model, so this is less likely to happen. We’d also like to note that for MNIST, in other metrics which more holistically assess sample quality, such as FLD, DDPP outperformed SVDD. This suggests DDPP achieves a better trade-off between sample quality and reward.
> >
> > > Additionally, why is there no BPD metric for the samples generated by SVDD?
> >
> > We’d like to clarify that BPD is computed on a pre-existing test set (for instance a set of even MNIST test digits), rather than on samples generated by any method. BPD isn’t reported for SVDD since it requires evaluating (a bound on) the likelihood for the model on this test set. For DDPP and the pre-trained model, this is done by computing the ELBO loss. Since SVDD is an inference time procedure, the model is unchanged, and instead, samples are steered through reweighting. This means that an analogous likelihood bound is not straightforward to compute.
> >
> > ## Degradation of sample quality after fine-tuning on CelebA
> > > In the CelebA experiment, it seems that the quality of samples from DDPP (Figure 3) is worse than those from the pretrained model (Figure 5).
> >
> > The reviewer astutely observes that the quality of samples post-fine-tuning with DDPP slightly degrades compared to the base model. We note that this phenomenon is well-known in the literature as many papers that do fine-tuning observe a quality-diversity tradeoff between optimizing for the reward vs. the base model [7]. More precisely, this fact has been proven theoretically in [8], where fine-tuning by filtering/using a reward model can amplify biases and also reduce overall sample quality. Intuitively, this occurs because the reward model is simply a classifier that ignores sample quality and only cares about the scoring samples that achieve a high-class probability. This intuition is further supported by the fact that all our considered baselines show a drop in sample quality—i.e. SVDD and RTB, the latter being particularly poor in quality. Thus we conclude that all fine-tuning methods suffer—to various extents—a degradation in quality compared to the base model. Importantly, we note that among the baselines DDPP samples have the highest reward while obtaining better visual fidelity than both SVDD and RTB—which suggests that it better negotiates this Pareto front.

---

> > > ### Author Response · Authors · 2024-11-21
> > > **Rebuttal (3/3)**
> > >
> > > ## On the minor comments
> > > We thank the reviewer for pointing out these errors and helping improve our manuscript. We have corrected these errors and highlighted them in the updated PDF in the color purple.
> > >
> > > > In line 98, the RHS of the equation p(X=x) is a bit confusing. Does it imply that each position is mutually independent? Clarification would be helpful.
> > >
> > > Thank you for your clarifying question. In discrete diffusion/flow matching approaches, each dimension is modeled independently in the forward process. The notation indicates the realization of a value $x$ taken by the random variable $X$. Note that this notation is standard in the literature e.g. discrete flow matching [9], and we drop the $X=x$ to reduce notational clutter.
> > >
> > > ## Closing comments
> > > We deeply appreciate the reviewer’s time and effort in reviewing and helping to improve our paper.  We hope that our additional experiments along with the questions answered in our rebuttal help to remove any remaining doubt the reviewer has regarding our manuscript. If the reviewer feels our manuscript and rebuttal merits it, we would be delighted if they considered raising our score as they suggested they would be open to doing. Finally, if the reviewer has any further questions or concerns we would be more than happy to address them through the end of the discussion period.
> > >
> > >
> > > References
> > >
> > > [1] Bengio, Emmanuel, et al. "Flow network based generative models for non-iterative diverse candidate generation." Advances in Neural Information Processing Systems 34 (2021): 27381-27394.
> > >
> > > [2] Lahlou, Salem, et al. "A theory of continuous generative flow networks." International Conference on Machine Learning. PMLR, 2023.
> > >
> > > [3] Malkin, Nikolay, et al. "GFlowNets and variational inference." arXiv preprint arXiv:2210.00580 (2022).
> > >
> > > [4] Sendera, Marcin, et al. "Improved off-policy training of diffusion samplers." The Thirty-Eighth Annual Conference on Neural Information Processing Systems. ACM, 2024.
> > >
> > > [5] Venkatraman, Siddarth, et al. "Amortizing intractable inference in diffusion models for vision, language, and control." arXiv preprint arXiv:2405.20971 (2024).
> > >
> > > [6] Hu, Edward J., et al. "Amortizing intractable inference in large language models." arXiv preprint arXiv:2310.04363 (2023).
> > >
> > > [7] Domingo-Enrich, C., Drozdzal, M., Karrer, B., & Chen, R. T. (2024). Adjoint matching: Fine-tuning flow and diffusion generative models with memoryless stochastic optimal control. arXiv preprint arXiv:2409.08861.
> > >
> > > [8] Ferbach, Damien, et al. "Self-consuming generative models with curated data provably optimize human preferences." arXiv preprint arXiv:2407.09499 (2024).
> > >
> > > [9] Gat, Itai, et al. "Discrete flow matching." arXiv preprint arXiv:2407.15595 (2024).
> > >
> > > [10] Sahoo, Subham Sekhar, et al. "Simple and Effective Masked Diffusion Language Models." arXiv preprint arXiv:2406.07524 (2024)

---

> > > > ### Author Response · Authors · 2024-11-24
> > > > **Kindly awaiting more feedback**
> > > >
> > > > Dear reviewer,
> > > >
> > > > We are very grateful for your thorough review of our paper which allowed us to provide additional clarifications in the rebuttal on the important raised points. We hope our rebuttal and the global response (including the additional experiments we included there) have allowed the reviewer to clear any remaining doubts about our paper, and if not we would love to engage further in the remaining time before the rebuttal period closes. Please note our rebuttal strived to answer all questions on the details and extensibility of our method, as well as to clarify all experimental results.
> > > >
> > > > We again appreciate the reviewer's time and would love to answer any further questions. We would also kindly request the reviewer to potentially consider updating their score as they suggested they may be open to if our rebuttal and global response have succeeded in addressing all the great points raised in the review.

---

> > > > > ### Comment · Reviewer_2bfH · 2024-11-24
> > > > >
> > > > > I would like to thank the authors for your detailed and professional response to all of my concerns, especially for pointing out related literature in using $L^2$-based loss, which I was not aware of. I suggest the authors discuss this motivation for considering $L^2$-based loss while introducing your methodology in Sec. 3. Also, thanks for pointing out the inapplicability of forward KL loss, which seems to be a naive question from hindsight. For the question of time-parameterization, I hope that the authors could briefly mention this small adaptation (i.e., using the number of masked tokens as a representation of time) in the paper. Also, I appreciate your explaination in both theoretical and empirical perspectives to the problem of higher SVDD reward and the degradation of sample quality after fine-tuning.
> > > > >
> > > > > Thank you again for your insightful response. I do not have further question on the paper, and as promised, I will raise my rating.

---

### Official Review · Reviewer_jeLb · 2024-11-04

**Soundness:** 3
**Presentation:** 3
**Contribution:** 3
**Rating:** 6
**Confidence:** 4

**Summary:**

The paper construct transition distributions of discrete diffusion models for sampling from the composed distribution $\pi_0(x_0) = p(x_0) R(x) / Z$, sometimes referred to as "guided" generation with reward function $R(x)$. Unlike in continuous space diffusions, the transition distribution  $p(x_{t-1} | x_t)$ corresponding to $\pi_0(x_0)$ cannot be approximated using a Taylor expansion around $x_t$. The method proposed by the authors instead trains a second model $q(x_0 | x_t)$ to match the effective transition distribution of $\pi_0(x_0)$. The primary challenge of matching the transition distribution of $\pi_0(x_0)$ is estimating the log partition function $\log Z$, and the authors propose three approaches to overcoming this challenge: (1) approximating $\log Z$ through importance sampling (2) parameterizing $\log Z$ with a neural network (3) foregoing approximation of $\log Z$ by matching the reverse KL, which instead requires back-propagation through stochastic discrete sampling. Each of these variants has slightly different requirements for inference/train time compute, or differentiability of $R$. The authors evaluate their proposed variants on a few toy tasks as well as on conditional image generation and protein design. Overall they find that the proposed method (DDPP) achieves better reward values while maintaining plausible samples.

**Strengths:**

The paper provides a nice overview of the problem of discrete guidance and several sensible solutions to it. I appreciate that the authors implemented several variants and compared them to each other, as it offers more signal to the community about promising directions. The significance of the work is substantial because discrete diffusion models are increasingly popular in both text applications, where guidance can equate to controllability/alignment, and for scientific modalities, where guidance is used for solving inverse problems. I personally don't find the originality to be high, as the method largely adopts pre-existing tools to solve the problems that arise in implementing each variant of their method, but I don't think this relatively low originality outweighs the quality of the work's presentation and analysis. To this end, I found the evaluation of the method mostly satisfactory, as it spanned multiple modalities and even included some wet lab experiments.

**Weaknesses:**

As I stated briefly above, I don't see the paper as making any foundational contributions that are broadly applicable beyond guidance of discrete diffusion models or any observations that are deeply surprising in nature. The paper is driven by a narrow and practical goal of finding a good method for approximating $\log Z$ (or foregoing its approximation) when training a secondary model $q$ to approximate the transition distribution of $\pi_0 (x_0)$. I personally don't find these contributions lackluster, as guidance of discrete diffusions is an open problem with practical impact. In terms of evaluation, the results seem fairly reasonable and there are some helpful baseline comparisons, though it's a little challenging to conclude much from the wet lab experiments when the only non-control methods are DDPP, so it's unclear how much heavy lifting the posterior inference method is doing versus other facets of sampling and filtering.

One notable limitation of the proposed method is that it requires training a secondary model in the first place. Once trained, this secondary model has relatively fast sampling, and therefore the training amortizes depending on how much inference is performed. In many scientific applications, inference can be relatively limited and throughput is often more important than latency, so in some cases maybe this tradeoff is not entirely worth it depending on the cost and complexity of setting up the post-training procedure, but I don't think these considerations disqualify the approach in general. It's also worth noting that best-of-N sampling seems to perform quite well in many experiment for relatively small N, even approaching the performance of DDPP in some cases, and this method can be deployed essentially out of the box, with only inference time overhead (which might be relatively minor).

**Questions:**

1. Why was no comparison made to other methods that post-train diffusion models with preference data, e.g. Diffusion-DPO [1]? Is there a fundamental limitation of these methods in the discrete setting or would porting these methods to a discrete setting just end up looking similar to DDPP-KL?
2. Was any variance reduction method required to make DDPP-KL work in practice?
3. Why is it sufficient that DDPP-LB provides a lower bound on the log partition? As far as I can tell the loss functions can incentive minimizing the value of log partition, in which case minimizing a lower bound is not analogous to minimizing the bounded quantity.
4. In the language examples, DDPP appears to perform much better than the provided baselines. Do you have any sense for how these conditional samples might compare to samples from a similar sized autoregressive model used in tandem with "steering" methods for autoregressive models (e.g. PPLM, fudge, twisted SMC). These models a probably capable of generating samples with higher likelihoods and many of these methods are "plug-and-play", meaning they don't require training a secondary model. Therefore, in practical settings, they might still be preferred over this method.

[1] Diffusion Model Alignment Using Direct Preference Optimization. Wallace et al. (2023)

---

> ### Author Response · Authors · 2024-11-21
> **Rebuttal (1/2)**
>
> We thank the reviewer for their time and detailed feedback on our paper that allowed us to provide a strengthened updated PDF draft as part of this rebuttal. We are pleased to hear that the reviewer finds that “the significance of the work is substantial” and that the evaluation of our method is “satisfactory” given that it spanned multiple modalities and included wet-lab validation. We now address the key points in the review, while additional experiments are included in the global response and rebuttal PDF.
>
> ## On the tradeoff of training a secondary model vs inference-based methods
> > One notable limitation of the proposed method is that it requires training a secondary model in the first place. Once trained, this secondary model has relatively fast sampling, and therefore the training amortizes depending on how much inference is performed. … It's also worth noting that best-of-N sampling seems to perform quite well in many experiments for relatively small N … and this method can be deployed essentially out of the box, with only inference time overhead (which might be relatively minor).
>
> We acknowledge the reviewer’s healthy skepticism regarding the utility of using best of $N$ as opposed to training a secondary model. We start by highlighting that fine-tuning for discrete models in general (e.g. RLHF) considers training another model and as such we believe that this amounts to standard practice in the community. As outlined, in the global response, the computational complexity of steering MDMs can be broken down into 3 aspects with training, inference, and querying the reward model being the main bottlenecks. As a result, in scientific domains, while the inference cost (a forward pass) might be cheap, querying the reward model is quite expensive, e.g. a protein folding model, and we wish to minimize this cost. In this case, best-of-$N$ is undesirable because we need to query the reward model $N$ times to get 1 purposeful sample, while DDPP amortizes this cost through training and inference doesn’t require querying the reward model. Quantitatively we address the reviewer's concern directly by calculating the wall clock time for our protein experiment and find that DDPP is still significantly faster than best of $N$ sampling, and even more so compared to SVDD. We hope this new result allays any reasonable concern the reviewer might have had.
>
> ## On the use of preference-based data baselines
> > Why was no comparison made to other methods that post-train diffusion models with preference data, e.g. Diffusion-DPO [1]? Is there a fundamental limitation of these methods in the discrete setting or would porting these methods to a discrete setting just end up looking similar to DDPP-KL?
>
> The reviewer brings up an interesting point regarding the use of Diffusion-DPO as a baseline. While we agree that Diffusion-DPO could be straightforwardly extended to discrete diffusion models we have two main reasons that we did not use it as a baseline. First, and most importantly, Diffusion-DPO requires a corpus of preference data which was, for all of our tasks, not readily available.  We are not aware of a pre-existing preference dataset for any of the tasks we used in our experiments, though would be happy to learn of their existence if the reviewer has any datasets in mind.  Second, there is mounting evidence that Diffusion-DPO is not particularly performant in continuous space.  This is because Diffusion-DPO requires computing the KL divergence between the fine-tuned and reference diffusion models which cannot be computed exactly because diffusion models provide a (loose) bound on the ELBO. Moreover, the probability flow ODE variant requires expensive simulation of the entire trajectory for each KL evaluation—making it computationally too expensive in practice. These concerns remain present in the discrete setting and unfortunately, DPO/RLHF methods provide a suboptimal mechanism for steering MDMs which is further evidenced by our chosen baseline SVDD which attempts to solve the RLHF problem for MDMs and performs generally worse than DDPP.
>
> ## Variance reduction for DDPP-KL
> > Was any variance reduction method required to make DDPP-KL work in practice?
>
> We thank the reviewer for the insightful question. For the MNIST experiments we used 8 samples to compute the KL term in DDPP-KL, while for the grid experiments we used a single sample. Other variance reduction techniques (such as baseline subtraction) were investigated and found to be unnecessary.

---

> > ### Author Response · Authors · 2024-11-21
> > **Rebuttal (2/2)**
> >
> > ## Autoregressive baselines
> > > In the language examples, DDPP appears to perform much better than the provided baselines. Do you have any sense for how these conditional samples might compare to samples from a similar sized autoregressive model used in tandem with "steering" methods for autoregressive models (e.g. PPLM, fudge, twisted SMC).
> >
> > We invite the reviewer to kindly refer to our global response which includes a new experiment comparing the performance of DDPP to Twisted SMC with an autoregressive model.  We believe that while autoregressive methods may still offer superior generative perplexity and sample quality, methods for fine-tuning discrete diffusion models such as DDPP will become more competitive as the machinery for pre-training large discrete diffusion models continues to improve and better rival the performance of autoregressive LLMs.
> >
> > ## Clarifying the DDPP-LB partition function lower bound
> >
> > > Why is it sufficient that DDPP-LB provides a lower bound on the log partition? As far as I can tell the loss functions can incentive minimizing the value of log partition, in which case minimizing a lower bound is not analogous to minimizing the bounded quantity.
> >
> > We acknowledge the reviewer's comment regarding DDPP-LB and it being a lower bound to DDPP-IS. We believe that there may be a cause for a small confusion which we now attempt to address. We do not claim that DDPP-LB lower-bounding the log partition function is sufficient for successful or optimal training; instead, we claim that by training the model with the DDPP-LB objective, our estimate for the log partition function by default always lower-bounds the log partition function, where the bound is tight at optimality. There is actually no incentive in the loss function to minimize the value of the log partition, in fact, the true value remains fixed and is defined as $Z=\int p^{pre}_0(x_0) R(x_0) dx_0$. In this sense, our lower bound should approach the true partition function as training progresses. Thus, we are not minimizing the lower-bound of any quantity, but optimizing an objective function which at optimality by definition says that $p(x) \propto p^{pre}_0(x) R(x)$, and thus allows us to sample from the right (product) distribution.
> >
> > ## Wet lab experiments control details
> > > In terms of evaluation, the results seem fairly reasonable and there are some helpful baseline comparisons, though it's a little challenging to conclude much from the wet lab experiments when the only non-control methods are DDPP, so it's unclear how much heavy lifting the posterior inference method is doing versus other facets of sampling and filtering.
> >
> > We appreciate the reviewer's concern but we would like to politely push back as the experimental setup of our wet-lab experiments may not have been sufficiently clear. While our evaluation does not include proteins generated by another baseline method due to the cost of protein synthesis, it does include direct comparisons between our designed proteins and four well-characterized controls: two de novo proteins (Top7 and OR485, designed by Rosetta(-derived) algorithms) and two natural proteins (ubiquitin and barstar). These controls were carefully selected because they represent strong benchmarks with experimentally solved structures and well-documented properties. Furthermore, Top7 and ubiquitin are predominantly alpha-helical, while OR485 and barstar are primarily beta-sheet proteins, making them structurally analogous to our designed proteins. We conclude by noting that our experimental findings, while reasonable, are demonstrative *as these proteins did not exist in the wild—a result we find particularly exciting for an ML-based conference in ICLR.*
> >
> > ## Closing comments
> > We hope that our responses were sufficient in clarifying all the great questions asked by the reviewer. We thank the reviewer again for their time, and we very politely encourage the reviewer to consider updating their score if they deem that our responses in this rebuttal along with the new experiments merit it. We are also more than happy to answer any further questions that arise.

---

> > > ### Author Response · Authors · 2024-11-24
> > > **Kindly awaiting more feedback**
> > >
> > > Dear reviewer,
> > >
> > > We are very appreciative of your time and constructive comments.  As the end of the rebuttal period is fast approaching we would like to have the opportunity to answer any lingering questions or doubts that may remain. We would like to note that in our rebuttal we followed your great suggestions and included new experiments with autoregressive baselines, as well as added further ablations showing the computation time benefit of our DDPP approach.
> > >
> > > We would be happy to engage in any further discussion on these points or any other salient points that the reviewer finds important, please let us know! We thank the reviewer again for their time and if the reviewer finds our rebuttal and new experimental findings satisfactory we also would appreciate it if the reviewer could potentially consider revising their assessment of our paper.

---

> > > > ### Comment · Reviewer_jeLb · 2024-11-25
> > > > **Rebuttal acknowledgement**
> > > >
> > > > I have read the rebuttal and appreciate the additional experiments. I hope they make it into the final paper, along with the helpful clarifications. I still support acceptance and I will maintain my score.

---

### Official Review · Reviewer_nvQG · 2024-11-04

**Soundness:** 3
**Presentation:** 3
**Contribution:** 4
**Rating:** 8
**Confidence:** 3

**Summary:**

This is a very interesting paper that design a new framework DDPP to sample from a target Bayesian posterior by steering pretrained MDMs. The author derived 3 primary objectives within the DDPP: DDPP-IS, DDPP-LB, DDPP-KL, providing different methods to handle the intractable partition function in the Bayesian posterior. This framework demonstrated strong performance across diverse tasks. In synthetic tasks, DDPP generated target-aligned samples with higher fidelity than baselines. For image modeling on the CelebA dataset, it produced attribute-specific images with competitive fidelity. In protein sequence modeling, DDPP generated high-quality sequences optimized for specific structural features, validated by wet-lab testing. In text generation, DDPP achieved alignment with sentiment and toxicity targets, producing fluent and reward-optimized outputs. Overall, DDPP consistently outperformed baselines, effectively steering model outputs to meet specific reward criteria across varied domains.

**Strengths:**

1. The authors introduce the DDPP framework for steering MNDMs  by treating the task as sampling from a Bayesian posterior, which is an innovative approach for discrete generative models. They derive three scalable, simulation-free objectives that make fine-tuning computationally efficient, even for large pre-trained models.
2. Extensive experiments were conducted across diverse tasks, including image modeling, text generation, and protein sequence design. DDPP demonstrated strong performance across all domains, highlighting its adaptability to various data types and applications.
3. The paper includes wet-lab validation for protein sequence tasks, confirming the practical applicability of the model in real-world protein design. This physical validation of AI-generated outputs adds quite a lot credibility.
4. The paper is clearly written and well-structured. The presentation is overall good.

**Weaknesses:**

1. Including autoregressive models as baselines could be useful, as they are widely used and effective in many generative tasks.
2. Since DDPP-IS involves MCMC, it would be helpful to analyze the computational cost of each method, covering both training and inference phases.

**Questions:**

N/A

---

> ### Author Response · Authors · 2024-11-21
> **Rebuttal**
>
> We would like to thank the reviewer for their time, feedback, and positive evaluation of our work.  We are heartened to hear that the reviewer feels that our paper was well written, that DDPP is “very interesting”, and that our empirical evaluation was compelling. We now address the key comments in the review.
>
> ## Autoregressive baselines
>
> We appreciate the reviewer’s great suggestion that the inclusion of fine-tuning of autoregressive baselines would provide a more holistic picture of DDPP’s utility given that it is primarily for masked diffusion models. Inspired by this comment we evaluate an autoregressive baseline in the form of Twisted SMC [1] on the Tinystories task and compare its performance to that of DDPP. We have added this result to the manuscript in Appendix E.1 and include a more detailed discussion of this in the global response, we note here that DDPP offers superior reward-optimizing performance compared to the autoregressive baseline while exhibiting comparable degradation in generative perplexity to Twisted SMC on autoregressive models.
>
> We note that this is not an apples-to-apples comparison because the base pre-trained models are different sizes.  Specifically, we wish to reiterate that applying Twisted SMC to the Masked Diffusion Models is at present not trivial—and perhaps not possible in its current form. Despite this, our new added autoregressive baseline allows us to contextualize DDPP for MDMs in that it is quite effective in negotiating the reward optimization vs. sample quality tradeoff that is inherent to all fine-tuning approaches, including Twisted SMC.
>
> ## Quantifying the computational overhead of DDPP-IS
>
> We acknowledge the reviewer's point regarding the need for further quantification of the computational overhead of the methods (especially DDPP-IS) in addition to what is already outlined in Table 1 of the paper. Though we briefly touch on this question in our global response, we dive deeper into the question here. We first clarify by noting that DDPP-IS involves drawing $M$ samples from the denoiser during each training iteration and evaluating the reward function on these. This involves 1 call to the denoiser (to obtain logits), and $M$ calls to the reward function. The complexity per training iteration is therefore larger by a factor of $M$ in the case where the latter operation dominates compute. We note that expensive Markov chains are not required in this method since it is tractable to sample from the denoiser. The inference time cost of all DDPP methods is identical to the cost of the pre-trained model. Namely, for $T$ diffusion steps, we require $T$ calls to the denoiser. We emphasize that the different DDPP methods differ only in the way they train the model, but have identical inference time algorithms.
>
> To better quantify the impact of the number of samples $M$ in DDPP-IS, we now include the results of ablations done on the MNIST, Amazon Reviews, and Tinystories tasks in Appendix E.3. A larger number of samples generally improves the reward at a faster rate over gradient updates, while each iteration generally takes more time, based on how expensive the reward function is to evaluate. The general suggestion from these experiments is that a smaller $M$ is preferable for tasks with more expensive reward functions, subject to more careful hyperparameter tuning.
>
> ## Closing comment
>
> We thank the reviewer for their time and constructive feedback. We hope that our responses were sufficient in clarifying all the great questions asked by the reviewer. We are more than happy to address any further questions that arise during this discussion phase.
>
> ## References
> [1] Zhao, Stephen, et al. "Probabilistic inference in language models via twisted sequential monte carlo."  ICML (2024)

---

> > ### Comment · Reviewer_nvQG · 2024-12-03
> >
> > Thanks for your response. I appreciate it. I will keep my score.

---

### Author Response · Authors · 2024-11-21
**Global Rebuttal (1/4)**

We would like to thank all reviewers for their time, thorough reviews, and valuable comments.  We are encouraged that the reviewers found our DDPP approach to be “innovative” with “strong performance” (nvQG), that the “significance of the work is substantial” (jeLB), that “the paper is well-written and clearly organized” (2bfH, nvQG, oyrv), and that it shows “strong performance across tasks” (oyrv, 2bfH). We were particularly pleased to hear that reviewers appreciated DDPP’s performance in the real world wet lab validation of DDPP’s generated samples (nvQG, jeLB).  We first summarize the main changes made in the updated draft before addressing concerns shared across multiple reviewers, grouped by theme.

## Summary of changes to the paper and additional experiments
We briefly summarize the changes made to the paper for rebuttal and their locations in the paper, including the additional experiments added for the rebuttal. All changes or additions to the paper have their text colored purple in the updated PDF.

- We added an autoregressive baseline for the Tinystories task and included a discussion of the results in Appendix E.1 (nvQG, jeLB).

- We include additional experiments and plots regarding the computational overhead of DDPP vs inference time methods in Appendix E.2 where we observed that on overall computation time DDPP was much faster than in inference time methods, especially as the number of generated samples scaled (jeLB, oyrv).

- We added an ablation on the impact of the number of samples in DDPP-IS on performance and computation time in Appendix E.3 where we observe that more IS samples result in a higher reward for fewer gradient updates, at the cost of more time per update (particularly dominated by the cost of evaluating the reward function) (nvQG).

- We added an ablation on the impact of varying protein sequence length in Appendix E.4 where we saw that DDPP maintains good performance even as sequence length is increased (oyrv).

- We corrected the various typos and made notational improvements according to suggestions from all reviewers.

We now discuss the experiments requested by multiple reviewers in more detail.

---

> ### Author Response · Authors · 2024-11-21
> **Global Rebuttal (2/4)**
>
> ## Compute time comparisons between DDPP and inference time methods (jeLB, oyrv)
> Reviewer comments:
> > (jeLB) It's also worth noting that best-of-N sampling seems to perform quite well in many experiment for relatively small N, even approaching the performance of DDPP in some cases, and this method can be deployed essentially out of the box, with only inference time overhead (which might be relatively minor).
>
> > (oyrv) The authors claimed that DDPP enabled efficient MDM fine-tuning across many domains … I assume SVDD is computationally friendly as it is purely inference time. … Thus, in terms of computation efficiency, I would assume DDPP is not superior to SVDD. Would the authors comment on this?
>
>
> We acknowledge the reviewers’ comments that an ML practitioner applying DDPP benefits from the knowledge of its computational complexity—and thus its practicality—and this aspect may not have been sufficiently clear in our original manuscript. We address this by highlighting the computational cost for fine-tuning in the settings considered can be broken down along 3 axes: 1.) Upfront cost of learning to sample DDPP or RLHF 2.) The inference time complexity of drawing samples from target distribution 3.) The cost of querying the task-dependent reward function. We further note Table 1 in the main paper serves to explain the computational cost of each method in terms of a.) Number of Model calls / inf. step b.) Number of Model calls / train steps, and c.) whether the method is simulation-free during fine-tuning.
>
> With this in mind, it is evident that a non-negotiable cost is associated with querying the reward function incurred regardless of the approach taken, e.g. SVDD, DDPP, etc…. For instance, our protein experiments represent a real-world setting where the reward function is costly to evaluate. More precisely, the reward function first folds a protein sequence with ESMFold, a costly operation involving passing the sequence through an LLM before doing a number of iterations of self-conditioned recycling through a structure module, before subsequently evaluating the properties of the predicted structure. Unfortunately, even while batching multiple sequences together the computation time required to fold sequences with ESMFold scales linearly, taking 5.72 seconds to evaluate a batch of 64 length 90 sequences.  Adding additional difficulty, latency increases substantially as sequence length increases, with a batch of 64 length 250 protein sequences taking 48.11 seconds to fold. Moreover, this is with setting ESMFold settings to maximally decrease query time at the cost of folding accuracy. We quantify the computational overhead of ESMFold queries in terms of sequence length and batch size in the newly added Figure 6b in Appendix E.2.
>
> Given this, even with the relatively cheap setting of generating length 90 protein sequences we find that on a single 80GB A100 GPU Best of 10 and SVDD take 6.20 and 37.5 hours to generate 1000 samples from the target distribution, respectively.  On the other hand, on the same hardware training DDPP took only 1.99 hours total to train and generate the 1000 samples, so that **DDPP took 18x less time than SVDD and 3x less time than Best of 10 in order to generate better samples**.  Moreover, this trend becomes even worse as the number of samples scales or the reward function becomes more expensive to query, which we quantify in Figure 6a in Appendix E.2. In general, the tradeoff between inference time and fine-tuning based methods for sampling from the target distribution is indeed one of the main motivations for RLHF-type methods in autoregressive generative modeling, where the overall computational overhead of fine-tuning a pre-trained model and sampling only from the fine-tuned model can become drastically cheaper than inference time methods such as best of N when the number of samples required is large enough.
>
> We thank the reviewers for their excellent question and refer them to Appendix E.2 in the updated manuscript for figures and an extended discussion of this point.

---

> > ### Author Response · Authors · 2024-11-21
> > **Global Rebuttal (3/4)**
> >
> > ## Autoregressive baseline (nvQG, jeLB)
> >
> > Reviewer comments:
> >
> > > (nvQG) Including autoregressive models as baselines could be useful, as they are widely used and effective in many generative tasks.
> >
> > > (jeLB) In the language examples, DDPP appears to perform much better than the provided baselines. Do you have any sense for how these conditional samples might compare to samples from a similar sized autoregressive model used in tandem with "steering" methods for autoregressive models (e.g. PPLM, fudge, twisted SMC) … In practical settings, they might still be preferred over this method.
> >
> > Multiple reviewers (nvQG, jeLB) were interested to see how DDPP might compare to autoregressive baselines. To this end, we evaluated Twisted SMC [1] on the Tinystories task and compared its performance to DDPP. Unfortunately, the base model used by Twisted SMC was of a different architecture and model size than we used. In particular, as the base autoregressive model, we used the GPT-Neo architectured model trained in the original Tinystories paper [2] which uses 68 million parameters while our base model used a diffusion transformer [3] with 170 million parameters. To ensure the fairest possible comparison, despite the difference in model parameters of the pre-trained models, we compare Twisted SMC and DDPP fine-tuning in terms of the percent change in the average reward and generative perplexity of the finetuned samples. These results are included in Appendix E.1 and reproduced below for convenience.
> >
> > | Method      | $\log R(x_0)$ before fine-tuning | $\log R(x_0)$ after fine-tuning | % Change in $\log R(x_0) \uparrow$ | Gen PPL before fine-tuning | Gen PPL after fine-tuning | % Change in Gen PPL $\downarrow$ |
> > | ----------- | -------------------------------- | ------------------------------- | ---------------------------------- | -------------------------- | ------------------------- | -------------------------------- |
> > | Twisted SMC | 40.52 $\pm$ 0.10                 | 94.56 $\pm$ 0.85                | 133.4 $\pm$ 1.6                    | 8.52 $\pm$ 0.05            | 10.25 $\pm$ 0.51          | **20.31 $\pm$ 5.96**             |
> > | DDPP        | 54.94 $\pm$ 0.76                 | 205.76 $\pm$ 3.88               | **278.0 $\pm$ 15.2**               | 16.66 $\pm$ 0.20           | 19.6 $\pm$ 0.69           | **18.38 $\pm$ 4.05**             |
> >
> > We observe that DDPP improves reward more than the autoregressive baseline while incurring a comparable but minor performance drop in generative perplexity to Twisted SMC. Our results here contextualize that DDPP can better negotiate the tradeoff between optimizing reward and sample quality than twisted SMC on autoregressive models. Finally, we note that Twisted SMC cannot be easily performed for MDM’s and as such DDPP remains a compelling choice for fine-tuning.

---

> > > ### Author Response · Authors · 2024-11-21
> > > **Global Rebuttal (4/4)**
> > >
> > > ## On when to use each of DDPP-KL, DDPP-IS and DDPP-LB (oyrv, nvQG)
> > > > (oyrv) It is still not crystal clear to me what are the motivations and the differences for making such 3 variants
> > >
> > > > (nvQG) Since DDPP-IS involves MCMC, it would be helpful to analyze the computational cost of each method, covering both training and inference phases.
> > >
> > > We acknowledge that in the paper it may not have been sufficiently clear which DDPP variant is preferred in each scenario, which we now clarify. In probabilistic inference and sampling tasks perhaps the most significant difficulty faced when training models is dealing with the intractable partition function. Unlike in the case where data is available, because we cannot directly sample from the target distribution most sampling objectives see the partition function appear in one way or another and determining how to deal with its presence is perhaps the main factor in the ultimate performance of a method.
> > >
> > > Because of the difficulty in estimating the partition function we proposed three different objectives each with its own way of dealing with the partition function. If the task of interest has a differentiable reward function DDPP-KL may be used.  DDPP-KL, through the use of a reparameterization trick, dispenses with the need to estimate the partition function entirely. If the reward function is instead non-differentiable we provide two other objectives: DDPP-IS and DDPP-LB. When the reward function for a task is particularly cheap, DDPP-IS --- which uses Monte Carlo samples to compute an importance sampling estimate of the partition function --- is a good choice. This is especially true given that the denoising posterior which we are learning is actually the optimal variance reducing proposal distribution for importance sampling on the partition function.  On the other hand, if the reward function is too costly to evaluate frequently as needed by the DDPP-IS Monte Carlo estimator one may make use of DDPP-LB.  By parameterizing a predictor of the partition function directly, DDPP-LB allows for a particularly cheap method of estimating the partition function. Moreover, as we show in Proposition 1, the partition function produced by DDPP-LB lower bounds the true partition function value with the bound becoming tight when the denoising posterior function has been learned perfectly.
> > >
> > > Overall we make the following recommendations for selecting which DDPP variant to use. First, if the reward function is differentiable and the task is of low to mid dimensionality we recommend the use of DDPP-KL. We advise caution with DDPP-KL for particularly high dimensional tasks as the Reinmax gradient estimator may become less accurate as dimensionality increases. Additionally, DDPP-KL requires more memory than the other methods, due to the reward differentiation, and a higher dimensionality may make this intractable. If the reward function is non-differentiable we typically saw better performance with DDPP-LB and in general recommend its use. However, if the reward function is computationally cheap enough to take many samples for the Monte Carlo estimator DDPP-IS may offer superior performance.  To aid in the selection of the number of Monte Carlo samples $M$ in DDPP-IS we add an additional ablation in Appendix E.3 where we observe that larger values of $M$ result in faster convergence in terms of number of gradient updates at the cost of more expensive updates per gradient step. To this end, we advise a practitioner to make the selection of $M$ on a task-by-task basis.
> > >
> > > ## Closing comment
> > > We would like to thank all reviewers again for their valuable time and effort in reviewing our manuscript and are especially grateful for the improvements to our paper that they led to. We hope that, if the reviewers believe our responses and overall manuscript merit it, they will consider raising our score to reflect those improvements.
> > >
> > > ## References
> > > [1] Zhao, Stephen, et al. "Probabilistic inference in language models via twisted sequential monte carlo."  ICML (2024)
> > >
> > > [2] Eldan, Ronen, and Yuanzhi Li. "Tinystories: How small can language models be and still speak coherent english?." arXiv preprint arXiv:2305.07759 (2023)
> > >
> > > [3] Peebles, William, and Saining Xie. "Scalable diffusion models with transformers." Proceedings of the IEEE/CVF International Conference on Computer Vision. 2023

---

### Meta-Review · Area_Chair_mGfK · 2024-12-19

**Metareview:**

This paper considers the posterior sampling problem from a masked discrete diffusion model. The authors propose a novel method for MDM posterior sampling termed discrete denoting posterior prediction (DDPP). The DDPP method is based on fine-tuning the prior model using the reward function. Posterior/preference data is not needed for training. All reviewers agree the paper contains interesting results and the paper is well-written. Several reviewers request comparison with more baselines. The authors have added experiment results accordingly. Some reviewers raise question on the technical results and presentation. The authors have added clarifications to address these comments. Overall, this paper makes a nice contribution to the topic of discrete diffusion model.

**Additional Comments On Reviewer Discussion:**

All reviewers agree the paper contains interesting results and the paper is well-written. Several reviewers request comparison with more baselines. The authors have added experiment results accordingly. Some reviewers raise question on the technical results and presentation. The authors have added clarifications to address these comments.

---

### Decision · Program_Chairs · 2025-01-22

Accept (Poster)